



# Observations of speciated isoprene nitrates in Beijing: implications for isoprene chemistry

Claire E. Reeves[1], Graham P. Mills[1], Lisa K. Whalley[2], W. Joe F. Acton[3], William J. Bloss[4], Leigh R. Crilley[4,5], Sue Grimmond[6], Dwayne E. Heard[7], C. Nicholas Hewitt[3], James R. Hopkins[8], Simone Kotthaus[6,9], Louisa J. Kramer[4], Roderic L. Jones[10], James D. Lee[8], Yanhui Liu[1], Bin Ouyang[10], Eloise Slater[7], Freya Squires[11], Xinming Wang[12], Robert Woodward-Massey[13], and Chunxiang Ye[13]

[1]Centre for Ocean and Atmospheric Sciences, School of Environmental Sciences, University of East Anglia, UK
[2]National Centre for Atmospheric Science, School of Chemistry, University of Leeds, UK
[3]Lancaster Environment Centre, Lancaster University, Lancaster, UK
[4]School of Geography, Earth and Environmental Sciences, the University of Birmingham, Birmingham, B15 2TT, UK
[5]now at Department of Chemistry, York University, Toronto, Canada.
[6]Department of Meteorology, University of Reading, Reading, UK
[7]School of Chemistry, University of Leeds, UK
[8]National Centre for Atmospheric Science, Wolfson Atmospheric Chemistry Laboratories, Department of Chemistry, University of York, UK
[9]Institut Pierre Simon Laplace, Ecole Polytechnique, France
[10]Department of Chemistry, University of Cambridge, UK
[11]Wolfson Atmospheric Chemistry Laboratories, Department of Chemistry, University of York, UK
[12]Guangzhou Institute of Geochemistry, Chinese Academy of Sciences, Guangzhou, China
[13]Beijing Innovation Center for Engineering Science and Advanced Technology, State Key Joint Laboratory for Environmental Simulation and Pollution Control, Center for Environment and Health, College of Environmental Sciences and Engineering, Peking University, Beijing, 100871, China

*Correspondence to*: Claire E. Reeves (c.reeves@uea.ac.uk)

**Abstract.** Isoprene is the most important biogenic volatile organic compound in the atmosphere. Its calculated impact on ozone ($O_3$) is critically dependent on the model isoprene oxidation chemical scheme, in particular the way the isoprene-derived nitrates (IN) are treated. By combining gas chromatography with mass spectrometry, we have developed a system capable of separating, and unambiguously measuring, individual IN isomers. In this paper we report measurements from its first field deployment, which took place in Beijing as part of the Atmospheric Pollution and Human Health in a Chinese Megacity (APHH-Beijing) programme, along with box model simulations using the Master Chemical Mechanism (MCM) (v.3.3.1) to assess the key processes affecting the production and loss of the IN.

Seven individual isoprene nitrates were identified and quantified during the summer campaign: two β-isoprene hydroxy nitrates (IHN); four δ isoprene carbonyl nitrates (ICN); and propanone nitrate. Whilst we had previously demonstrated that the system can measure the four δ-IHN, we found no evidence of them in Beijing.





The two β-IHN mixing ratios are well correlated with an $R^2$ value of 0.85. The mean for their ratio ((1-OH, 2-ONO$_2$)-IHN : (4-OH, 3-ONO$_2$)-IHN) is 3.4 and exhibits no clear diel cycle (the numbers in the names indicate the carbon (C) atom in the isoprene chain to which the radical is added). Examining this in a box model demonstrates its sensitivity to nitric oxide (NO), with lower NO mixing ratios favouring (1-OH, 2-ONO$_2$)-IHN over (4-OH, 3-ONO$_2$)-IHN. This is largely a reflection

of the modelled ratios of their respective precursor peroxy radicals which, at NO mixing ratios of less than 1 part per billion (ppb), increase substantially with decreasing NO. Interestingly, this ratio in the peroxy radicals still exceeds the kinetic ratio (i.e. their initial ratio based on the yields of the adducts from OH addition to isoprene and the rates of reaction of the adducts with oxygen (O$_2$)) even at NO mixing ratios as high as 100 ppb. The relationship of the observed β-IHN ratio with NO is much weaker than modelled, partly due to far fewer data points, but it agrees with the model simulation in so far as there

tend to be larger ratios at sub 1 ppb amounts of NO.

Of the δ-ICN, the two *trans* (E) isomers are observed to have the highest mixing ratios and the mean isomer ratio (E-(4-ONO$_2$, 1-CO)-ICN to E-(1-ONO$_2$, 4-CO)-ICN)) is 1.4, which is considerably lower than the expected ratio of 6 for addition of NO$_3$ in the C1 and C4 carbon positions in the isoprene chain. The MCM produces far more δ-ICN than observed,

particularly at night and it also simulates an increase in the daytime δ-ICN that greatly exceeds that seen in the observations. Interestingly, the modelled source of δ-ICN is predominantly during the daytime, due to the presence in Beijing of appreciable daytime amounts of NO$_3$ along with isoprene. The modelled ratios of δ-ICN to propanone nitrate are very different to the observed.

This study demonstrates the value of speciated IN measurements to test our understanding of the isoprene degradation

chemistry. Our interpretation is limited by the uncertainties in our measurements and relatively small data set, but highlights areas of the isoprene chemistry that warrant further study, in particular the NO$_3$ initiated isoprene degradation chemistry.

# 1 Introduction

Isoprene is the most important biogenic volatile organic compound (BVOC) in the atmosphere, with its emissions accounting for around 500 Tg yr$^{-1}$, about half of the global biogenic non-methane VOC emissions (Guenther et al., 2012). It

is emitted by vegetation primarily during the daytime as a function of temperature and solar radiation and is readily oxidised by the hydroxyl (OH) and nitrate (NO$_3$) radicals and ozone (O$_3$). Through its degradation chemistry, isoprene impacts O$_3$ and the formation of secondary organic aerosols (SOA), which together impact the oxidising capacity of the atmosphere and radiative forcing. Global and regional model studies show that the calculated impact of isoprene on O$_3$ is critically dependent on the model isoprene oxidation chemical scheme, in particular the way the isoprene-derived nitrates (IN) are treated (e.g.

Emmerson and Evans, 2009; Fiore et al., 2005; Squire et al., 2015; von Kuhlman et al., 2004; Wu et al., 2007). Much of the



uncertainty in this chemistry is related to the yield and fate of IN, in particular whether NO$_X$ (nitrogen oxides) and radicals, which are tied up in the nitrates, are later recycled or lost from the atmosphere.

First generation IN are formed following oxidation of isoprene by either OH or NO$_3$ (Wennberg et al., 2018) (Fig. 1). The
OH reaction dominates accounting for around 85 % of the reactive fate of isoprene largely due to concurrent daytime presence of isoprene and OH. On oxidation by OH, peroxy radicals are formed which when they react with nitric oxide (NO) can lead to the formation of hydroxy nitrates (IHN), with a yield of around 4-15 % (e.g. Chen et al., 1998; Chuong and Stevens, 2002). These are dominated by β-IHN, but some δ-IHN are also formed. Although NO$_3$ is mostly present at night, isoprene oxidation by NO$_3$ can be important, particularly in the early evening (Brown et al., 2009) and, due to the larger
organic nitrate yield of ~65-80 % (Kwan et al., 2012; Perring et al., 2009b, Rollins et al., 2009; Schwantes et al., 2015), can be responsible for a considerable proportion (~40-50 %) of the IN (Horowitz et al., 2007; von Kuhlmann et al., 2004; Paulot et al., 2012; Xie et al., 2013). Depending on the fate of the peroxy radicals formed following NO$_3$ addition, a variety of IN can be produced: isoprene hydroperoxy nitrates (IPN); isoprene dinitrates (IDN); isoprene carbonyl nitrates (ICN); as well as IHN.


The fate of first generation IN is poorly understood and until recently understanding was based on theoretical calculations, with most observational constraints based on measurements of either groups of nitrates as totals, or degradation products that come from more than one reaction and precursor species (Giacopelli et al., 2005; Paulot et al., 2009; Rollins et al., 2009). Much advancement in recent years has been made through new laboratory studies following the synthesis of some of the IN
(Jacobs et al., 2014; Lee et al., 2014; Lockwood, et al., 2010; Teng et al., 2017; Xiong et al., 2016), but these are still limited to specific IN isomers (six IHN and one ICN) and reaction rates for others are based on extrapolation and structural activity relationships. The IN are lost via reaction with OH (lifetimes ~3-10 h at 0.04 parts per trillion (ppt) of OH), O$_3$ (lifetimes ~10 -1000 h at 40 ppb of O$_3$) and NO$_3$ (lifetimes ~100 -400 h at 1 ppt of NO$_3$) (lifetimes calculated for 298K and 993 hPa based on rate recommendations in Wennberg et al., (2018)) and by photolysis (Xiong et al., 2016; Müller et al., 2014) and
deposition (Nguyen et al., 2015).

One of the key issues is whether NO$_X$ and radicals are returned to the system or whether they remain tied up in second generation nitrates. In the case of the IHN and ICN reactions with OH can lead to the formation of carbonyls and release of NO$_2$ or the formation of shorter chained nitrates such as methyl vinyl ketone nitrate, methacrolein nitrate, propanone nitrate
(acetone nitrate) and ethanal nitrate, with the ratio between these two pathways differing for specific IN isomers (Wennberg et al., 2018). Critically the β-IHN and δ-IHN have different lifetimes and return different fractions of NO$_2$ (Paulot et al., 2012). As the yields of the IHN from the β and δ peroxy radicals are fairly similar (Teng et al., 2017), a key factor in the amount of NO$_X$ recycled is the relative yields of the β and δ peroxy radicals from OH oxidation of isoprene.



Over the last decade or so there has been a considerable effort to improve understanding of isoprene oxidation mechanisms, with the latest understanding detailed in Wennberg et al. (2018) (hereafter referred to as W2018). More details of the chemistry of IN are given in Sect. 1 of the Supplementary Information.

The implications of this chemistry have been subject to several model studies, although it should be noted that several of these studies predate many of the recent advancements in understanding of the detailed isoprene nitrate chemistry outlined above. However, the key findings are that the yield of the IN and the rate at which $NO_X$ is recycled are important in determining the overall impact of isoprene on $O_3$ and the distribution of $O_3$ production.

Wu et al. (2007) found that increasing the yield of the IHN from 4 % to 12 % led to a significant decrease in the global production rate of $O_3$. They also found that the production rate of $O_3$ saturated above a certain threshold of VOC emissions due to $NO_X$ being efficiently lost from the atmosphere via the IN. Fiore et al. (2005) calculated that increasing the yield of IHN from 8 % to 12 %, and assuming $NO_X$ is permanently lost via the IHN, led to decreases in surface $O_3$ concentrations by 4-12 ppb in the South Eastern U.S. In a regional model of the same area Xie et al. (2013) found that uncertainties in the IN chemistry can impact $O_3$ production by 10 % and OH concentrations by 6 %, the main uncertainty being in the efficiency of $NO_X$ recycling rather than IN yield or dry deposition rate. This is consistent with a similar conclusion regarding the importance of the removal, recycling and export of $NO_X$ by IN on tropical $O_3$ (Paulot et al., 2012). Emmerson and Evans (2009), in a comparison of chemistry schemes, showed that even the sign (positive or negative) of the impact of isoprene on $O_3$ varied between schemes. They identified the treatment of IN as the dominant cause of these discrepancies. Similarly Squire et al. (2015) assessed how four different reduced isoprene schemes in a global model respond to climate, isoprene emission, anthropogenic emission and land-use change. They noted the importance of the yield of the IN and how this also affected the distribution of $O_3$ production, with more produced locally and less further away when the IN yields are lower.

The results of the model studies need to be evaluated against field data of IN. Some of the early field measurements used gas chromatography (GC) to separate the IHN (Werner et al., 1999; Giacopelli et al., 2005; Grossenbacher et al., 2001; 2004), but without synthesised samples of the IN, the identity of the specific isomers could not be confirmed. Several studies (e.g. Horowitz et al., 2007; Perring et al., 2009a; Mao et al., 2013; Xie et al., 2013; Zare et al., 2018) have used thermal dissociation laser-induced fluorescence spectroscopy (TD-LIF) measurements of the sum of alkyl and multifunctional nitrates (∑ANs) as observational constraints, but whilst this includes IN, it also includes other organic nitrates, and there is no separation of individual IN. Field studies using chemical ionisation mass spectrometry (CIMS) (e.g. Beaver et al., 2012; Xiong et al., 2015; Fisher et al., 2016; Lee et al., 2016; Lee et al., 2018) were able to provide further insight through partial separation of different types of IN, including first generation and second generation IN, but were not able to distinguish between different isomers (e.g. of IHN).



By combining GC with mass spectrometry (MS), we (Bew et al., 2016; Mills et al., 2016) and Vasquez et al. (2018) have
developed systems capable of separating, and unambiguously measuring, individual IN isomers in the field. In our case we
used negative ion (NI) MS whilst Vasquez et al. (2018) used CIMS.

In this study we deploy our system in the field for the first time, allowing us to quantify the concentrations of several of the
individual IHN along with some of the ICN and propanone nitrate during a major field campaign in Beijing as described in
Sect. 3. In Sect. 4 we present the observed time series of the measured IN, examine their ratios and diel patterns. We use a
simple box model to examine the ratio of the β-IHN isomers (Sect. 5) and a more detailed chemical box model to investigate
how their ratio changes with NO and to assess the key processes affecting the production and loss of all the IN measured
(Sect. 6).

## 145  2 Nomenclature

In this paper when naming the IN we have followed the nomenclature described by Wennberg et al. (2018) who state: 'we
assign numbers to the carbons of isoprene as follows: carbons 1−4 comprise the conjugated butadiene backbone, with the
methyl substituent (carbon 5) connected to carbon 2 of the backbone. Throughout the text, we refer to these carbons as "C#"
without subscripts (e.g., "C2"); subscripted numbers (e.g., "$C_2$") are used instead to refer to the total number of carbon atoms
in a molecule. In the names of functionalized isoprene oxidation products, we drop the "C" when describing substituent
positions; for example, (1-OH, 2-$ONO_2$)-isoprene hydroxy nitrate (IHN) has a hydroxy group at C1 and a nitrooxy group at
C2.' This is different to the way we named the IN in Mills et al. (2016) in which the IHN naming followed that of
Lockwood et al. (2010) and the ICN were named similarly to the equivalent IHN where the oxygen atom and nitrate are in
the same position in the molecule, except they have "–al" as a suffix. We also referred to acetone nitrate as NOA in Mills et
al. (2016) whereas here we refer to it as propanone nitrate.

## 3 Field campaigns and instrumentation

## 160  3.1 Field campaigns



The GC-NI-MS system was deployed in Beijing as part of the Atmospheric Pollution and Human Health in a Chinese Megacity (APHH-Beijing) programme (Shi et al., 2019) during two campaigns at the Institute of Atmospheric Physics (IAP), Chinese Academy of Sciences. IAP is located at 39.97° N, 116.38° E in a residential area between the 3rd and 4th

North ring roads of Beijing. The site contained small trees and grass, with roads 150 m away. The first campaign was in winter (10th November to 10th December 2016) and the second in the summer (21st May to 22nd June 2017). For reasons discussed below we shall focus on the summer campaign

### 3.2 Isoprene nitrate measurements


This was the first deployment of a GC-NI-MS system in the field to measure speciated isoprene nitrates. Air was drawn at 10 L min$^{-1}$ down a 2.5 m heated inlet (3/8" PFA and 45 °C) mounted on the roof (a height of approximately 3 m above the ground) of a mobile laboratory. During the summer campaign three different instrument setups were employed: (1) From the start of the measurements to 31 May, samples of 500 ml were taken off the inlet line down a 0.3 m length of 0.53 mm ID

MxT-200 transfer line held at 50 °C and preconcentrated on a Tenax adsorption trap at 35 °C and 50 ml min$^{-1}$, and injected onto the column via a metal six port Valco valve by heating to 150 °C (Mills et al. 2016). A 30.5 m, 0.32 mm (internal diameter (ID)) combination column was used which was comprised of 28 m of Rtx-200 followed by 2.5 m of Rtx-1701 column. The GC oven was temperature profiled from 40 °C to 200 °C, with a constant column flow of 4.5 ml min$^{-1}$ of helium; (2) Between 10th June and 16th June, the system was operated without a trap but instead direct injection of a 3 ml

sample through a plastic Valco Cheminert valve connected to a short 0.32 mm ID combination column (2.5 m of Rtx-200 joined to 0.5 m of Rtx-1701). The GC oven was temperature programmed from 10 °C to 200 °C and cooled with carbon dioxide ($CO_2$). A constant flow of 6.5 ml min$^{-1}$ of helium was used as the carrier gas; (3) From 18th June to the end, the system again used the 30 m column and Tenax trapping as described above but the metal valve was replaced with the Cheminert valve that was used for the direct injections.


Of the compounds reported here, all but those of (1-OH, 2-$ONO_2$)-IHN and E-(1-$ONO_2$, 4-CO)-ICN were confirmed by injection of known isomers (Mills et al. 2016) post campaign. (1-OH, 2-$ONO_2$)-IHN was identified based on its expected elution just before (4-OH, 3-$ONO_2$)-IHN (Nguyen et al., 2014) and the similarity of the observed ions to those of (4-OH, 3-$ONO_2$)-IHN. The E-(1-$ONO_2$, 4-CO)-ICN peak was identified by its relative elution position compared to the other ICN

(Schwantes et al., 2015), its expected retention time estimated from the relative retention times of known δ-IHN on this system and their aldehydic equivalents, and the similarity of observed ions to the other ICN.

During several comparisons of samples measured immediately before and after the valve was changed from metal to plastic and vice versa (1 h between samples), it was evident that the (4-OH, 3-$ONO_2$)-IHN and the ICN were lost to varying degrees



on the metal valve as suggested by Crounse (J. D. Crounse, personal communication 2016), while simple alkyl nitrates were not. To account for this, all data obtained with the metal valve were scaled by the ratio of peak areas from the samples on either side of the valve changes to give results equivalent to those obtained when using the Cheminert valve.

Calibrations for (4-OH, 3-ONO$_2$)-IHN and propanone nitrate were derived from the relative sensitivity of the compound to
that of n-butyl nitrate (Mills et al., 2016) corrected for the relative ion abundances of the specific measurement ions used for each compound (m/z 71 and 73, respectively). M/z 73 is a relatively minor ion for propanone nitrate, but we were unable to use a more major ion due to interferences from other compounds. N-butyl nitrate calibrations were performed every few days by attaching the transfer line to the standard in place of the inlet. We were unable to measure the relative sensitivities of (1-OH, 2-ONO$_2$)-IHN and the ICNs to n-butyl nitrate directly. To obtain an estimate, we have assumed that the ICN and
propanone nitrate all have the same total ion yields compared to those of n-butyl nitrate and scaled this relative total ion yield by the fraction of the ion yield that the measurement ion represents. Similarly we have assumed that the total ion yields of (1-OH, 2-ONO$_2$)-IHN and (4-OH, 3-ONO$_2$)-IHN are the same (and thus the n-butyl nitrate m/z 71: total IN ion ratio) and scaled this to reflect the proportion of the total ions that m/z 101 represents for (1-OH, 2-ONO$_2$)-IHN.

**3.3 Other measurements**

A large suite of meteorological and chemical measurements was made during the campaigns (Shi et al., 2019). Here we describe briefly only those used in this paper. Isoprene was measured using a dual channel GC with a flame ionisation detector (DC-GC-FID) (Hopkins et al. (2011). A Thermo Environmental Instruments (TEI) 49i UV absorption analyser was
used to measure O$_3$. Measurements of OH, HO$_2$ and RO$_2$ were obtained using the fluorescence assay by gas expansion (FAGE) technique equipped with a scavenger inlet for OH, with the OH chem method used to obtain the background OH signal (Whalley et al., 2010; Whalley et al., 2018; Woodward-Massey et al., 2019). NO was measured using a TEI 42i, NO$_2$ by a Teledyne cavity attenuated phase shift (CAPS) instrument and HONO by long path absorption photometer (LOPAP) (Crilley et al., 2016). NO$_3$ and glyoxal were measured using broadband cavity enhanced absorption spectroscopy (Kennedy
et al., 2011). SO$_2$ was measured by TEI 43i and CO by a sensor box (Smith et al., 2017). A proton transfer reaction-time of flight-mass spectrometer (PTR-ToF-MS) was used to measure multi-functional aromatics and monoterpenes, whilst HCHO was measured by LIF (Cryer, 2016). The mixed layer height was determined from the attenuated backscatter measured with a Vaisala CL31 ceilometer (Kotthaus and Grimmond (2018), and photolysis rates from spectral radiometer measurements (Bohn et al., 2016).




# 4 Field observations

## 4.1 Overview of air quality conditions

During the winter campaign air quality was very poor with several haze events and average concentrations of $PM_{2.5}$ of ~90 $\mu g\ m^{-3}$, and average mixing ratios of $NO_2$ of 40 ppb, CO of 1300 ppb and $O_3$ of ~8 ppb (Shi et al., 2019). During the summer campaign air quality was also poor, but the mix of pollutants differed, with higher amounts of $O_3$ and lower amounts of $PM_{2.5}$, $NO_2$ and CO. i.e. average concentrations of $PM_{2.5}$ of ~30 $\mu g\ m^{-3}$ and average mixing ratios of $NO_2$ of 15 ppb, CO of 450 ppb and $O_3$ of ~45 ppb (Shi et al., 2019).


## 4.2 Isoprene nitrate observations

### 4.2.1 Time series

The winter campaign was the first field deployment of the GC-NI-MS system for measuring isoprene nitrates. With the very poor air quality the air was loaded with nitrated species which led to many unidentifiable peaks in the chromatograms and, with low temperature and sunlight, biogenic emissions of isoprene were low and no IN were identified, however the campaign allowed useful tests of the field operation procedures. We shall therefore limit our presentation of results to the summer campaign.


Seven individual isoprene nitrates were identified and quantified during the summer campaign (Fig. 2): two β-IHN ((1-OH, 2-$ONO_2$)-IHN, (4-OH, 3-$ONO_2$)-IHN); four ICN (E-(1-$ONO_2$, 4-CO)-ICN, Z-(1-$ONO_2$, 4-CO)-ICN, E-(4-$ONO_2$, 1-CO)-ICN, Z-(4-$ONO_2$, 1-CO)-ICN); and propanone nitrate.

Whilst we had previously demonstrated that the system can measure the four δ-IHN (Mills et al., 2016), we found no evidence of them in Beijing. This may not be so surprising given that Xiong et al. (2015) calculated the sum of the δ-IHN to have made up only a few percent of the total IHN during Southern Oxidant and Aerosol Study (SOAS) in the United States in 2013. Jenkin et al. (2015), using the Master Chemical Mechanism (MCMv3.3.1) (http:/mcm.york.ac.uk), calculated that the molar faction of IHN yield that would be made up of δ-IHN would increase with increasing NO such that for NO mixing 255 ratios of 5-40 ppb typical of peak daytime values in Beijing this would be around 5-15% but less than 5% for mean daytime





NO values of ~2.5 ppb. Note the isomer distribution will also be affected by loss processes and the δ-IHN have shorter lifetimes than the β-IHN (W2018).

Figures 3 and 4 show the time series of the IN and other relevant chemical species during the summer campaign. All measured IN follow similar patterns with elevated mixing ratios for the first five days, followed by a period of five days of lower values before rising again (Fig. 3). There were then breaks in data and a period of seven days whilst the GC-NI-MS was run in direct injection mode enabling the measurement of (1-OH, 2-ONO$_2$)-IHN along with (4-OH, 3-ONO$_2$)-IHN. During this period these two β-IHN followed similar patterns. There was then a final period when the trap was reinstalled
when the other IN again broadly followed similar patterns.

The general trend in IN mixing ratios does not appear to be related to a similar trend in isoprene mixing ratios (Fig. 4). The isoprene time series exhibits a number of spikes in mixing ratios, several of which occurred at night. These are likely from very local sources, probably anthropogenic, injected into a shallow nocturnal mixed layer. The highest mixing ratios of IN
appear not to be related to polluted periods, but rather coincide with low NO mixing ratios when the NO$_2$:NO ratios were high (Fig. 4), which suggests that the air was more photochemically aged.

For the few days with measurements of both (1-OH, 2-ONO$_2$)-IHN and (4-OH, 3-ONO$_2$)-IHN available, the sum of the β-IHN show daily maxima of around 40-120 ppt with night-time values of around 10-30 ppt (Fig. 5). This is broadly similar to
the sum of IHN reported for the SOAS (Xiong et al., 2015; Schwantes et al., 2016). Likewise, our observed sum of the δ-ICN also exhibited night-time peaks of around 30-70 ppt which are broadly similar to those reported for SOAS (Schwantes et al., 2016). Our measurements of propanone nitrate often exceeded 50 ppt making them generally higher than those observed in SOAS, which rarely exceeded 40 ppt (Schwantes et al., 2016). There is a large uncertainty in our calibration for propanone nitrate since we were only able to use a minor ion for detection during the field campaign (Sect. 3.2).

### 4.2.2 IN ratios

The two β-IHN are well correlated with an R$^2$ value of 0.85. The mean for the ratio (1-OH, 2-ONO$_2$)-IHN : (4-OH, 3-ONO$_2$)-IHN is 3.4 (standard deviation of 1.7) and exhibits no clear diel cycle (Fig. 5). This compares with the average
daytime ratios of ~2.6 and ~1.4 obtained by Vasquez et al. (2018) in Michigan during the PROPHET campaign and at Pasadena California, respectively. Xiong et al. (2015) calculate a ratio ranging from 2.6 to 6.0 based on the conditions experienced in SOAS. Jenkin et al. (2015), using the MCMv3.3.1, calculated that the ratio of (1-OH, 2-ONO$_2$)-IHN to (4-OH, 3-ONO$_2$)-IHN decreases with increasing NO and is around 2 for NO mixing ratios typical of the peak daytime values



observed in Beijing. It should be noted that the ratio we obtain from our measurements is not based on an independent
calibration for (1-OH, 2-ONO$_2$)-IHN, but based on the assumption that the analytical system has the same sensitivity to (1-OH, 2-ONO$_2$)-IHN as it does to (4-OH, 3-ONO$_2$)-IHN. Nevertheless, the ratio we obtain is broadly consistent with the studies described above and it is examined in more detail with models in Sect. 5 and Sect. 6.

Of the δ-ICN, the two *trans* (E) isomers have the highest mixing ratios with E-(1-ONO$_2$, 4-CO)-ICN being the most
abundant (Fig. 3). Focusing on the last four days (three nights) if the summer campaign (Fig. 6), when we have most confidence in the data (i.e. when the plastic valve was used (Sect. 3.2), we see that the ratios of the ICN showed similar values each day. There are some diel variations in these ratios, particularly those involving E-(1-ONO$_2$, 4-CO)-ICN, which shall be discussed further below.

The mean observed ICN C1:C4 isomer ratio is 1.4 (i.e. E/Z-(1-ONO$_2$, 4-CO)-ICN : E/Z-(4-ONO$_2$, 1-CO)-ICN). This value is considerably lower than would be expected based solely on the addition of NO$_3$ to isoprene occurring in the C1 and C4 positions in a ratio of 6 (C1:C4) (W2018). Our observed ratios are more comparable to the C1:C4 isomer ratio of 2.8 (74:26) reported in Schwantes et al. (2016) for their environmental chamber. It should be noted that in their experiment the ICN mostly came from RO$_2$ + RO$_2$ reactions (see Sect. S1.2) because the NO and NO$_3$ concentrations were low, whereas we had
relatively higher NOx concentrations in Beijing. Turning to the E:Z ratios, we observed the E-ICN isomers to dominate over the Z-ICN isomers. The (1-ONO$_2$, 4-CO)-ICN isomers exhibit a mean E:Z ratio of 5, whilst the (4-ONO$_2$, 1-CO)-ICN isomers exhibit a mean E:Z ratio of 11, giving an overall mean E:Z ratio of 7. These values are far greater than the *trans*:*cis* ratio of 1 presumed by W2018 for the reaction of NO$_3$ addition to isoprene, based on the OH addition to C1 of isoprene calculated by Peeters et al. (2009). However, it should be noted that the peroxy radicals formed from the reaction of the
adducts with O$_2$ may be in a different ratio as these reactions are reversible, similar to those for peroxy radicals formed following OH addition to isoprene, as discussed in Sect. S1.1.

### 4.2.3 Diel patterns

In this section we examine the diel patterns of the IN and other trace gases by considering the medians for each hour of the day (Fig. 7). The isoprene mixing ratios exhibit a typical diel pattern with the highest values (~1 ppb) around midday, which are maintained through the afternoon before declining in the evening to near zero values at night. Some of the variability in the values is caused by the high spikes shown in Fig. 4. O$_3$ mixing ratios build up gradually through the daytime, peaking mid to late afternoon and then slowly declining to minimum values around sunrise. Remarkably the median diel peak value
of O$_3$ was very high at around 100 ppb, demonstrating the considerable amount of photochemical pollution during the campaign. OH concentrations also exhibited a typical diel cycle peaking around midday at just below 1 x 10$^7$ molecules cm$^-$





[3]. Evidence was found of low, but appreciable concentrations of OH at night, along with peroxy radicals, signifying the presence of nocturnal radical chemistry. NO peaked just after sunrise at around 7 ppb.

The β-IHN, as illustrated by (4-OH, 3-ONO$_2$)-IHN, exhibit diel patterns (Fig. 7) that are consistent with formation from OH oxidation of isoprene (Sect.S1.2). They peak around midday and these levels are maintained until around sunset when they decline to reach minimum values just after sunrise. This pattern is broadly similar to that observed during SOAS for total IHN with a daytime peak of around 70 ppt and a minimum around sunset of around 10 ppt (Xiong et al., 2015). However, the SOAS IHN peaked earlier in the day at 10:00 Central Daylight Time, i.e. prior to the daytime maxima in OH and

isoprene. Xiong et al. (2015) attribute this to competition between the different peroxy radical (ISOPOO) reactions, with the relative importance switching from reaction with NO to reaction with HO$_2$. Whilst NO mixing ratios peaked in the morning in Beijing, similar to those in SOAS, they are of greater magnitude and remain so into the afternoon (e.g. the median value only goes below 1 ppb in the mid-afternoon (Fig. 7)), thus favouring IHN production for longer. Xiong et al. (2015) also suggest that mixing down of IHN from the residual layer may contribute to the morning increase in IHN mixing ratios. We

will explore the diel pattern in (4-OH, 3-ONO$_2$)-IHN in more detail with a simple model in Sect. 5.

Conversely, the δ-ICN, as illustrated by E-(4-ONO$_2$, 1-CO)-ICN, exhibit nocturnal peaks, with maximum values in the early night and minimum values during the daytime (Fig. 7), which is consistent with formation from NO$_3$ addition to isoprene (Sect. S1.2) in the evening and a lifetime of the order of a few hours or less. The median mixing ratio increases in the

evening at a rate of 3.0 ppt h$^{-1}$ from around 2.6 to 7.6 ppt in 2 hours. We calculated the concentrations of NO$_3$ required to produce such an increase by considering its loss via reactions with OH, O$_3$ and NO$_3$, and its production from NO$_3$ addition to isoprene with an assumed yield of ~5 %. We use bimolecular rate coefficients given in W2018 and number densities of the reactants from the observations (Fig. 7). This calculation assumes that the rate limiting step in the production of (4-ONO$_2$, 1-CO)-ICN is the reaction of isoprene with NO$_3$ and that deposition is negligible. The 5 % yield is based on Schwantes et al.

(2015) chamber experiments results, whereby we assume a 20 % yield of ICN from NO$_3$ addition to isoprene, of which 25 % is E-(4-ONO$_2$, 1-CO)-ICN. We calculate that ~2.5 ppt of NO$_3$ is required to produce the observed evening increase in E-(4-ONO$_2$, 1-CO)-ICN, which is consistent with the observed NO$_3$ mixing ratios (median values rise from 2 ppt to 5 ppt from 18:00 to 20:00) (Fig. 7).

Interestingly, ~1-2 ppt of E-(4-ONO$_2$, 1-CO)-ICN persists during the daytime. We performed a similar calculation to the above, but this time assuming steady state and a photolysis rate based on Xiong et al. (2016) (i.e. a value of 4.6 x 10$^{-4}$ s$^{-1}$ for a solar zenith angle of 0° and then adjusting for latitude, time of year and time of day). We find that around 1-2 ppt of NO$_3$ is required to produce the observed E-(4-ONO$_2$, 1-CO)-ICN. Whilst we observe this amount during the afternoon, median values in the morning are ~0.1-0.3 ppt (Fig. 7). This might suggest mixing down of ICN into the mixed layer in the morning

but would require considerable production of ICN in the residual layer during the previous evening/night.





As noted above, the ratios of E-(1-ONO$_2$, 4-CO)-ICN with both E-(4-ONO$_2$, 1-CO)-ICN and Z-(1-ONO$_2$, 4-CO)-ICN exhibit diel patterns (Fig. 6). The ratios are higher at night and lower in the daytime. The evening ratios of E-(1-ONO$_2$, 4-CO)-ICN to E-(4-ONO$_2$, 1-CO)-ICN are driven by the preferential addition of NO$_3$ to the C1 position as discussed above in Sect. 4.2.2.

The decrease in this ratio during the morning could be explained if the lifetime of E-(1-ONO$_2$, 4-CO)-ICN were shorter than for the other isomers. However, the rate coefficients for reaction with OH recommended by W2018 are about 20 % slower for (1-ONO$_2$, 4-CO)-ICN than for (4-ONO$_2$, 1-CO)-ICN. Photolysis is expected to be the largest daytime sink, but Xiong et al (2016) only determined this for E-(4-ONO$_2$, 1-CO)-ICN. It is not clear why there is such a strong diel pattern in the ratio of the E and Z isomers of (1-ONO$_2$, 4-CO)-ICN, but during the daytime when the mixing ratios are close to our detection

limit the ratios will be very sensitive to uncertainties in the measurements.

Propanone nitrate shows no clear diel cycle. The pattern can change from day to day, sometimes peaking during the daytime and sometimes at night-time. This is illustrated in Fig. 8 which shows the temporal variation of propanone nitrate, along with (4-OH, 3-ONO$_2$)-IHN and (4-ONO$_2$, 1-CO)-ICN for the last five days of the campaign. Propanone nitrate peaks on the night

of the 19/06/2017 coincident with the ICN suggesting formation from NO$_3$ oxidation of isoprene (Sect. S1.2). This is followed by two peaks, one in the night of the 20-21/06/2017, again coincident with the ICN, and then a second peak during the daytime of the 21/06/2017 when (4-OH, 3-ONO$_2$)-IHN maximises. Schwantes et al. (2016) noted that on some days during the SOAS campaign propanone nitrate increased after sunrise following the presence of ICN the night before, while on other occasions night-time ICN was not followed by increases in propanone nitrate or propanone nitrate appeared during

the day when ICN had not been present the night before. They suggest that as well as photooxidation of the ICN, boundary layer dynamics may have played a role as propanone nitrate may have been formed aloft in a residual layer at night and was then mixed down to the surface in the morning. We investigate the night-time and daytime sources of propanone nitrate during the Beijing campaign further in Sect. 6.5 through the use of a model.

**5 Simple Box Modelling of the β-IHN**

A simple model was constructed to calculate the diurnal cycle of the β-IHN. The production of the β-IHN used the equation of Xiong et al. (2015):

$$P = k_{\text{ISOP+OH}}[\text{OH}][\text{ISOP}] \times \varphi \times \gamma \times \alpha \qquad (1)$$





where $k_{\text{ISOP+OH}}$ is the rate constant for the reaction of isoprene with OH, [OH] and [ISOP] are the number densities of OH and isoprene. φ is the yield of ISOPOO following OH addition to isoprene that is available to react with NO, HO$_2$ and RO$_2$. I.e., it takes account of the redistribution of the ISOPOO isomers as a result of the reversibility of the reactions of the OH

adducts with O$_2$ and formation of hydroperoxyaldeyhyde (HPALD) following Z-δ-ISOPOO isomerisation (see Sect. S1.1). Xiong et al. (2015) calculates a value for φ of around 0.8 for β-ISOPOO when their lifetimes are around 1 to 35 s so we adopt that value for φ. As we consider the β-IHN separately we then need to assume a distribution of the β-ISOPOO so we assume this is the same as the kinetic ratio of the β-ISOPOO yields recommended by W2018 (65:35) giving values of φ of 0.52 and 0.28 for (1-OH, 2-OO)-ISOPOO and (4-OH, 3-OO)-ISOPOO, respectively, which given their short lifetimes in the

high NO$_X$ environment is reasonable. γ is the fraction of ISOPOO that reacts with NO, as opposed to reacting with HO$_2$, RO$_2$ or NO$_3$. Isomerisation is assumed to be negligible for β-ISOPOO loss. Due to the high NO$_X$ environment, γ was calculated to be very close to 1 throughout the whole day, ranging from 1.00 in the morning for both β-ISOPOO to 0.95 in the later afternoon / early evening for (4-OH, 3-OO)-ISOPOO and 0.97 for (1-OH, 2-OO)-ISOPOO. The higher value for (1-OH, 2-OO)-ISOPOO is due to its reaction with RO$_2$ being about 3.5 times slower than for (4-OH, 3-OO)-ISOPOO. α is the

branching ratio of the reaction of ISOPOO with NO that forms the IHN, which is set at 0.104 for both β-IHN based on the MCMv3.3.1 (Jenkin et al., 2015). Reaction rate constants are taken from the MCMv3.3.1 (Jenkin et al., 2015) and the number densities are the hour of day medians from the observations.

To calculate the diel variation in the mixing ratios of the β-IHN, we used the equation for the solution of the chemical
continuity equation:

$$C_{(t+\Delta t)} = \frac{P}{k\prime} + \left( C_t - \frac{P}{k\prime} \right) e^{-k\prime \Delta t} \tag{2}$$

$C_t$ is the concentration of each IHN at time $t$, $P$ is the production calculated using Eq. (1), $k'$ is the loss rate constant for each
IHN and $\Delta t$ the time step. $C_{t+\Delta t}$ was calculated for each hour using median concentrations of the reactants for that hour.

For the loss rate constants of the β-IHN, we consider the reactions with OH, O$_3$ and NO$_3$, taking the recommended rate constants from W2018 and use the observed median number densities of the reactants for the hour of day. We calculate the photolysis rates as a function of solar zenith angle using the parameterisation in the MCM, which assumes the rates for (1-
OH, 2-ONO$_2$)-IHN and (4-OH, 3-ONO$_2$)-IHN are equivalent to those of tert-butyl nitrate and iso-propyl nitrate, respectively. Based on the observed decrease in (4-OH, 3-ONO$_2$)-IHN between midnight and 6 in the morning, we calculate a first order loss of 4 x 10$^{-5}$ s$^{-1}$, which we take to be the dry deposition loss rate constant, and which we apply to the whole diel cycle for both β-IHN.



Average diel concentrations of all the chemical reactants were taken for the whole campaign period when the IN were measured. We considered two scenarios for initialising the concentrations of (4-OH, 3-ONO$_2$)-IHN: firstly using the average midnight concentrations for the whole campaign; secondly using the average midnight concentrations from just the days when (1-OH, 2-ONO$_2$)-IHN were measured. We ran the model for 2 days and by the second day the model had reached a steady state and the results were almost exactly the same for the two different methods of initialisation – i.e. the results were

independent of the initial concentrations. We therefore present the model results from the second day of simulation.

Figure 9 shows that the calculated production rates of (1-OH, 2-ONO$_2$)-IHN are about twice those of (4-OH, 3-ONO$_2$)-IHN. Since α and γ are either the same or very similar for both β-ISOPOO, this results from the ratio of the values used for φ (i.e. 1.85). Peak values of 1.26 x 10$^6$ and 0.68 x 10$^6$ cm$^{-3}$ s$^{-1}$ for (1-OH, 2-ONO$_2$)-IHN and (4-OH, 3-ONO$_2$)-IHN, respectively,

occur at midday when OH and isoprene concentrations are greatest, giving a total β-IHN production rate of 1.94 x 10$^6$ cm$^{-3}$ s$^{-1}$. This compares to a rate of around 0.6 x 10$^6$ cm$^{-3}$ s$^{-1}$ for IHN for the SOAS campaign (Xiong et al., 2015). During SOAS, isoprene concentrations were approximately 4 times higher than during the APHH summer campaign, but this was countered by the OH concentrations being around 4 times lower, and the NO mixing ratios being approximately 8 times lower, the latter leading to gamma being around 4 times lower.


The modelled mixing ratios of the β-IHN are far higher than the observed values (Fig. 9) particularly during the daytime when they are approximately 6 times higher for (4-OH, 3-ONO$_2$)-IHN and 4 times higher for (1-OH, 2-ONO$_2$)-IHN. The observed values (solid dots, Fig. 9) are from 10$^{th}$ to 16$^{th}$ June 2017 only, the time when (1-OH, 2-ONO$_2$)-IHN was measured, and so it should be noted that that three of the hourly bins contain just one value, with the rest having between three and

eight values. The observed mixing ratios of (4-OH, 3-ONO$_2$)-IHN are slightly higher for this period than for the whole campaign (crosses, Fig. 9), but the model still vastly overestimates the daytime values. One of the causes of the overestimation is likely to be dilution, in particular through entrainment of air aloft into the mixed layer, especially in the morning. Our simple model takes no account of this.

To counter the impact of the mixing we compared the modelled ratio of (1-OH, 2-ONO$_2$)-IHN to (4-OH, 3-ONO$_2$)-IHN with that observed (Fig. 10). There is a lot of scatter in the observed values largely because of the limited number of measurements. The observed ratios are also based on the assumption that our analytical system has the same sensitivity to both β-IHN. The model tends to give lower values than the observed ratio (mean values for the model and observed are 2.5 and 3.4, respectively). The ratio is a function of the relative production and loss rates of the two β-IHN. For the modelled

production the relative difference is driven by the values used for φ (ratio of 1.85 based on the kinetic ratio of the yields of the two β-ISOPOO) as we assume the same values for α and, in the high NO$_X$ environment, calculate very similar values for γ. As for the relative loss rates, (1-OH, 2-ONO$_2$)-IHN has a longer lifetime than (4-OH, 3-ONO$_2$)-IHN primarily due to its slower rate of reaction with OH (Fig. 12). The modelled ratio of (1-OH, 2-ONO$_2$)-IHN to (4-OH, 3-ONO$_2$)-IHN decreases

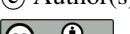



slightly just after sunrise when the production of the β-IHN rapidly increase (Fig. 9) despite the value for φ being 1.85 times

greater for (1-OH, 2-ONO$_2$)-IHN. This appears to be due to there being a greater amount of (1-OH, 2-ONO$_2$)-IHN existing overnight such that its relative increase is initially smaller than that of (4-OH, 3-ONO$_2$)-IHN. During the rest of the day the modelled ratio gradually increases due to the value for φ being greater for (1-OH, 2-ONO$_2$)-IHN and its loss rate being slower.

One way of getting the model to give a better fit to the observed (4-OH, 3-ONO$_2$)-IHN to (1-OH, 2-ONO$_2$)-IHN ratio would be to increase the ratio of the yields of the β-ISOPOO from 1.85 to ~ 2.5 (Fig. 10). Alternatively, we can consider the reactions of the β-IHN with OH which are their main loss processes. The rate coefficient for (1-OH, 2-ONO$_2$)-IHN we have used is that recommended by W2018 which is 0.71 times that for (4-OH, 3-ONO$_2$)-IHN. The MCM, however, uses a lower rate coefficient for (1-OH, 2-ONO$_2$)-IHN which is 0.52 times that for (4-OH, 3-ONO$_2$)-IHN. Reducing the rate coefficient

for (1-OH, 2-ONO$_2$)-IHN by a factor of 0.72 in line with the ratio of the rate coefficients used by the MCM leads to modelled ratios of the β-IHN much closer to the observed (Fig. 10). The loss of the β-IHN due to the assumed rates of photolysis are very small compared to those due to reaction with OH, so changes to them would do little to improve the comparison of the modelled ratios of the β-IHN with the observed, unless they were much faster, i.e. more like that derived by Xiong et al. (2016) for E-(4-ONO$_2$, 1-CO)-ICN. Adjusting the deposition rates to improve the modelled comparison with

the observed ratio of (1-OH, 2-ONO$_2$)-IHN to (4-OH, 3-ONO$_2$)-IHN would require increasing the rate for (4-OH, 3-ONO$_2$)-IHN relative to that for (1-OH, 2-ONO$_2$)-IHN, which is not what we would expect given the greater difficulty we have getting (1-OH, 2-ONO$_2$)-IHN through our analytical system indicating a greater loss of this isomer on to surfaces, and the fast rate of hydrolysis reported for (1-OH, 2-ONO$_2$)-IHN (W2018) (Sect. S1.3.5).

# 6 MCM Box modelling

## 6.1 MCM model set up

A zero dimensional box model, utilising a subset of the chemistry described within the Master Chemical Mechanism, MCMv3.3.1 (Jenkin et al., 2015), was used to calculate the concentration of the various isoprene nitrates for comparison

with those measured. The MCMv3.3.1 includes an update of the isoprene degradation chemistry to reflect findings of recent laboratory and theoretical studies.

The model was constrained by measured values of water vapour, temperature, pressure, NO, NO$_2$, NO$_3$, O$_3$, CO, SO$_2$, HONO and HCHO. Speciated VOC measurements of alcohols, alkanes, alkenes, dialkenes (including isoprene), multi-





functional aromatics, carbonyls and monoterpenes were included as further model constraints. The concentrations of $H_2$ and

$CH_4$ were held constant at 500 ppb and 1.8 ppm, respectively. The photolysis rates for $j(O^1D)$, $j(NO_2)$ and $j(HONO)$,

calculated from the measured actinic flux and published absorption cross sections and quantum yields, were included as

model inputs. Other photolysis frequencies used in the model were calculated. For UV-active species, such as HCHO and

$CH_3CHO$, photolysis rates were calculated by scaling to the ratio of clear-sky $j(O^1D)$ to observed $j(O^1D)$ to account for

clouds. For species able to photolyse further into the visible the ratio of clear-sky $j(NO_2)$ to observed $j(NO_2)$ was used. The

variation of the clear-sky photolysis rates ($j$) with solar zenith angle ($\chi$) was calculated within the model using the following

expression:

$$j = l \cos(\chi)^m \times e^{-n \sec(\chi)} \tag{3}$$


with the parameters $l$, $m$ and $n$ optimised for each photolysis frequency (see Table 2 in Saunders et al. (2003)).

The model was run for the entirety of the campaign (21st May 2017 – 25th June 2017) in overlapping 7 day segments, with

the model constraints updated every 15 minutes. By this method, a model time-series was produced which could be directly

compared with observations and, from which, diel averages were generated. Fluxes through each reaction were calculated for

every 15 minute period to allow an analysis of the production and loss terms of the chemical species.

The loss due to mixing of all non-constrained, model generated species, including the speciated isoprene nitrates, was

parametrised and evaluated by comparing the model-predicted glyoxal concentration with the observed glyoxal

concentration. Applying a loss rate proportional to the observationally-derived mixed layer height (Fig. 7), the model was

able to reproduce glyoxal observations reasonably well. As a result of this first order loss process, the partial lifetime of the

model generated species was ~2 h at night, then decreased rapidly to a lifetime of <30 min in the morning as the mixed layer

grew, effectively simulating ventilation of the model box. With the collapse of the mixed layer in the late afternoon the

model lifetime with respect to ventilation of glyoxal (and other model generated species) increased. However, the model has

a tendency to underestimate glyoxal concentrations between 4 pm and midnight. This underestimation suggests that either

the lifetime with respect to ventilation should be even longer or that the model is underestimating oxidation processes that

lead to glyoxal production at these times.

### 6.2 β-IHN


The MCM simulates (1-OH, 2-$ONO_2$)-IHN daytime peak mixing ratios similar to those observed for the few days with

measurements available (Figs. 11 and 12). The modelled night-time mixing ratios are lower than observed. Like glyoxal, this





underestimation suggests that either the lifetime with respect to ventilation should be longer than 2 hours or that the model is underestimating production of the (1-OH, 2-ONO$_2$)-IHN, the only modelled pathway being reaction of NO with (1-OH, 2-

OO)-ISOPOO. For (4-OH, 3-ONO$_2$)-IHN, the MCM tends to give larger mixing ratios than observed during the daytime but lower than observed from early evening to midnight (Figs. 11 and 12). Again, like glyoxal, this night-time underestimation suggests that that the lifetime with respect to ventilation might be longer than 2 hours. Whilst increasing the production of the (4-OH, 3-ONO$_2$)-IHN, the only modelled pathway being the reaction of NO with the (4-OH, 3-OO)-ISOPOO, would improve the comparison of the model with the night-time measurements it would worsen it during the daytime. The

production rates calculated for both β-IHN using the MCM (Fig. 12) are very similar to those calculated using the simple model (Fig. 9).

To limit the impact of mixing on the comparison between the model and observations, Fig. 10 compares the diel pattern of the ratios of (1-OH, 2-ONO$_2$)-IHN to (4-OH, 3-ONO$_2$)-IHN from the observations and as calculated by the MCM along with

those calculated using the simple model. Throughout much of the day the MCM simulates very similar ratios to that calculated by the simple model and are lower than the observed ratio.

There are four main factors that determine the ratio of the β-IHN: 1) the yields of their respective peroxy radicals (ISOPOO) following oxidation of isoprene by OH addition (represented by φ in the simple model); 2) the fraction of the respective

ISOPOO that reacts with NO (represented by γ in the simple model); 3) the relative loss rates of the β-IHN; and 4) the branching ratios for the formation of the IHN from the reaction of NO with the ISOPOO. It should be noted that deposition is not included in the MCM model but, as discussed for the simple model (Sect. 5), we would expect (1-OH, 2-ONO$_2$)-IHN to be lost more efficiently than (4-OH, 3-ONO$_2$)-IHN which would further reduce the agreement with the observed β-IHN ratios.


For the first two factors, the concentration of NO is largely the determining influence. NO is present in large amounts so that reaction with it is the dominant loss process for the ISOPOO. NO is the major factor determining the lifetime of the ISOPOO and therefore the extent of the redistribution of the ISOPOO from a kinetic ratio towards a thermodynamic equilibrium. The adducts formed from OH addition to a specific C in isoprene can form a β-ISOPOO and either a *trans* or

*cis* δ-ISOPOO. These reactions are reversible and occur at different rates which along with the rapid 1,6 H atom shift isomerisation of the Z-δ-ISOPOO means that the longer the lifetime of the ISOPOO the more the ratio of the β-ISOPOO shifts towards (1-OH, 2-OO)-ISOPOO. Consequently, at lower NO mixing ratios the ratio of φ-(1-OH, 2-ONO$_2$)-IHN to φ -(4-OH, 3-ONO$_2$)-IHN becomes larger. This is illustrated in Fig. 13, which shows the modelled ratio of the values of φ (each calculated as the net production of the specific ISOPOO from the reversible reactions of the OH-adduct with O$_2$ divided by

the loss of isoprene due to addition of OH). For mixing ratios of NO greater than ~2 ppb the ratio of the values of φ decreases approximately linearly from around 2 to about 1.7 at 100 ppb of NO. The values of φ that we used for the simple





model have a ratio of 1.85 and came from the using the kinetic yields recommended by W2018. The MCM uses slightly different kinetic yields giving a ratio of 1.58, which is the ratio of the values of $\varphi$ that we get if we switch off the reverse pathway of the $O_2$ reactions. This implies that even at 100 ppb of NO, the ratio of the yields of the (1-OH, 2-OO)-ISOPOO
to (4-OH, 3-OO)-ISOPOO is shifted to values slightly greater than the kinetic ratio. At NO mixing ratios less than ~2 ppb the ratio of the values of $\varphi$ increase greatly with decreasing NO, such that at a few 10s of ppt of NO the ratio is typically between 2.5 and 4.

The rates at which the ISOPOO are assumed to be lost via the reactions with NO, $HO_2$ and $NO_3$ are the same for both β-
ISOPOO. However, the rate of reaction for (1-OH, 2-OO)-ISOPOO with $RO_2$ and its rate of isomerisation are slower than for (4-OH, 3-OO)-ISOPOO. At lower NO mixing ratios, these reactions become relatively more important and so the value of $\gamma$ is lower for (4-OH, 3-OO)-ISOPOO than for (1-OH, 2-OO)-ISOPOO, therefore the ratio of $\gamma$-(1-OH, 2-OO)-ISOPOO to $\gamma$-(4-OH, 3-OO)-ISOPOO is larger (Fig. 14). This is further extenuated as the concentrations of $RO_2$ can also be much greater at the lower NO concentrations, particularly below 1 ppb of NO (Fig. 14), which leads to the ratio in the $\gamma$ values
being considerably greater than 1 at NO concentrations below a few 10s of ppt.

The net effect of these relationships is that the ratio of (1-OH, 2-OO)-ISOPOO to (4-OH, 3-OO)-ISOPOO increases with decreasing NO (Fig. 13), i.e. for NO mixing ratios greater than 2 ppb the ratio is around 1.7-2.0, but at NO mixing ratios less than 2 ppb the ratio increases up towards a value of around 4. The ratio of the rate of production of (1-OH, 2-$ONO_2$)-IHN to
(4-OH, 3-$ONO_2$)-IHN will have the same relationship with NO as the ratio of their precursor ISOPOO. The diel pattern of the median hourly NO mixing ratios (Fig. 7) illustrates that whilst NO mixing ratios were typically above 2 ppb between 06:00 and 12:00 local time, they were mostly below this value in the afternoon when production rates of the β-IHN were also high (Fig. 12). This explains some of the diel pattern in the ratio of the β-IHN shown in Fig. 10.

As for the loss processes of the β-IHN, the dominant one in the model is the mixing term which is set at the same rate for both β-IHN. Photolysis is assumed to be faster for (1-OH, 2-$ONO_2$)-IHN than for (4-OH, 3-$ONO_2$)-IHN in the MCM, but is only a minor loss process. However, (1-OH, 2-$ONO_2$)-IHN reacts with both OH and $O_3$ more slowly than does (4-OH, 3-$ONO_2$)-IHN and since the dominant chemical loss process for the β-IHN are by far their reactions with OH, the net effect of these loss processes is to increase the ratio of (1-OH, 2-$ONO_2$)-IHN to (4-OH, 3-$ONO_2$)-IHN above their production ratio.
The diel pattern in OH (Fig. 7) will tend to increase the ratio of 1-OH, 2-$ONO_2$)-IHN to (4-OH, 3-$ONO_2$)-IHN during the daytime (Fig. 10). Interestingly, the MCM simulation is closer to the simple model which uses the rate coefficients for the reactions of the β-IHN with OH recommended by W2018, rather than the simple model run in which they are set to those in the MCM (Fig. 10).



It should be noted that the MCM model underestimates the measured $RO_2$ mixing ratios. This will lead to underestimation of the ratio of γ-(1-OH, 2-OO)-ISOPOO to γ-(4-OH, 3-OO)-ISOPOO and consequently the ratio of (1-OH, 2-ONO$_2$)-IHN to (4-OH, 3-ONO$_2$)-IHN, primarily at mixing ratios of NO below ~2 ppb. This might explain some of the differences between the MCM modelled and observed β-IHN ratios and why the MCM ratios are lower than those of the simple model run that used the MCM rate coefficients for the reactions of OH with the β-IHN (the simple model was constrained by observed $RO_2$

concentrations).

We have also assumed that the branching ratios for the formation of the two β-IHN from the reaction of NO with the ISOPOO are the same. However, there are still considerable uncertainties in these branching ratios (Sect. S1.1).

Overall, this means that the ratio of (1-OH, 2-ONO$_2$)-IHN to (4-OH, 3-ONO$_2$)-IHN increases with decreasing NO mixing ratios (Fig. 13) (as also seen by Jenkin et al. (2015) in a box model using the MCM), and generally does not drop below the ratio of the β-ISOPOO, which sets a baseline value (Fig. 13). In the conditions modelled for Beijing, at NO mixing ratios above ~30 ppb it remains between 1.75 and 2.0. At NO mixing ratios between 1 ppb and ~30 ppb it does not fall below 1.95, is mostly around 2, but is sometimes up to 3. At NO mixing ratios below 1 ppb, it is typically between 2 and 3, but

sometimes up to 4. There are several cases at these low NO mixing ratios when the ratio of the β-IHN is below the ratio of the β-ISOPOO, but these occur at night when the production rates and the mixing ratios of the β-IHN are very small.

In comparison, the observed ratios of (1-OH, 2-ONO$_2$)-IHN to (4-OH, 3-ONO$_2$)-IHN show a much weaker relationship with NO (Fig. 15). This may be due to there being far fewer data points and uncertainties in the measurements. The observed

ratios tend to be higher, although at times they drop below the kinetic ratio for φ. Although there is clearly far more scatter there is a tendency for higher values of the observed ratio at NO mixing ratios of less than 1 ppb, as simulated by the model.

The field observations reported by Vasquez et al. (2018) are consistent with our results in that they obtained higher average daytime values for the ratio of (1-OH, 2-ONO$_2$)-IHN to (4-OH, 3-ONO$_2$)-IHN in the low NO$_X$ environment of the

PROPHET campaign compared to the high NO$_X$ environment in Pasadena (i.e. ratios of ~2.6 and ~1.4, respectively). On the other hand, the ratio of ~3.4 that we observed in the very high NO$_X$ environment of Beijing is higher than for the two US studies. On average our modelled results (both with the simple model and the MCM) are close to the ratios observed in PROPHET, despite Beijing often being a high NO$_X$ environment. We cannot rule out calibration differences affecting this comparison and like us Vasquez et al (2018) relied on relative calibrations estimates. Also, differences in the observed β-

IHN ratios may be due to the amount and reactivity of the peroxy radicals present in the different studies. However, the ratio of 1.4 observed for Pasadena is lower than the value of around 1.75 that we calculate for NO mixing ratios of 100 ppb, and furthermore, it is also lower than the kinetic φ ratios of 1.58 and 1.85 based on MCM and W2018 kinetic yields, respectively.


**6.3 δ-ICN**

The δ-ICN time series (Fig. 16) show that the MCM often produces far more δ-ICN than observed, particularly at night as illustrated by the diel patterns (Fig. 17). Note that an event in which the modelled mixing ratios of the δ-ICN reached nearly 10 ppb in the early hours of the $16^{th}$ June 2017 (Sect. 6.6) skewed the night-time 15 minute mean values. Removing this event from the modelled means gives night-time values that are much lower but still higher than the observed values. Moreover, the MCM model simulates an increase in the daytime δ-ICN that far exceeds that seen in the observations. We are unable to assess the ratios of the different δ-ICN isomers using the model as the MCM assumes all of the δ-ICN formed can be represented by a single species, (1-ONO$_2$, 4-CO)-ICN, called NC4CHO in the MCM.

The source of δ-ICN is via the addition of NO$_3$ to isoprene followed by addition of O$_2$. This produces δ-nitroxy peroxy radicals (INO$_2$) (NISOPO2 in the MCM) and, in the conditions simulated for Beijing, the major loss of NISOPO2 is reaction with NO to form NO$_2$ and a δ-nitroxy alkoxy radical (NISOPO in the MCM), which then reacts rapidly with O$_2$ to form the δ-ICN (NC4CHO). Other production pathways for NC4CHO exist in the MCM (reactions of OH with isoprene hydroperoxy nitrate and with isoprene dinitrate, and reaction of NISOPO2 with RO$_2$), but the reaction of NISOPO with NO is by far the dominant source of δ-ICN in our simulations. There are some nights when there are large sources of NISOPO2, but typically the production of NISOPO2 maximises in the mid-afternoon when isoprene concentrations are still high and median NO$_3$ mixing ratios were observed to be around 2 ppt (Fig. 7). Consequently, the production of δ-ICN in the model is mostly during the daytime, despite NO$_3$ usually being considered to be more important at night. Comparison of the modelled and observed mixing ratios of the δ-ICNs suggest that this source might be too fast even during the daytime, despite the model being constrained by observed concentrations of isoprene and NO$_3$.

Alternatively, the loss processes could be too slow. The dominant loss in the model is the mixing term, which is greatest during the daytime when the mixed layer is fully developed. The same loss process has been applied to all model generated species (Sect. 6.1). For glyoxal, (1-OH, 2-ONO$_2$)-IHN and (4-OH, 3-ONO$_2$)-IHN, the model tends to underestimate the observed concentrations from late afternoon onwards suggesting that the lifetime with respect to mixing should be longer at these times. Increasing the lifetime of all the model intermediates would lead to a further overestimation of δ-ICN. Alternatively, this potentially suggests that applying the same loss term to all model species is not necessarily appropriate. Overestimation of the dilution due to the growing mixed layer depth at around 7 am might explain why the modelled mixing ratios are lower than observed at that time.





The next most important loss processes for δ-ICN are simulated to be photolysis and reaction with OH, which are also both predominantly daytime losses. The net effect of the production and loss terms is that the modelled δ-ICN maximise during the night-time (Fig. 17).

The MCM uses a photolysis frequency for δ-ICN based on that measured for propanone nitrate, which is equivalent to $3.16 \times 10^{-4}$ $s^{-1}$ for a solar zenith angle of 0°. Xiong et al. (2016) determined a rate of $4.6 \times 10^{-4}$ $s^{-1}$ for (4-ONO$_2$, 1-CO)-ICN for a solar zenith angle of 0°. Reaction with OH constitutes a similar size loss for δ-ICN as photolysis in the model. Whilst these are both predominantly daytime sinks, increasing them would not only reduce the daytime increase in δ-ICN but would also reduce the amount of modelled δ-ICN that would persist into the night. The MCM treats all the δ-ICN as (1-ONO$_2$, 4-CO,)-

ICN and uses a rate coefficient for reaction with OH of $4.1 \times 10^{-11}$ $cm^3$ $s^{-1}$. However, W2018 suggests a lower rate coefficient for reaction of OH with (4-ONO$_2$, 1-CO)-ICN than for (1-ONO$_2$, 4-CO,)-ICN ($3.4 \times 10^{-11}$ $cm^3$ $s^{-1}$ versus $4.1 \times 10^{-11}$ $cm^3$ $s^{-1}$). Therefore, treating the two separately in the model would, overall, reduce the loss of δ-ICN with respect to OH, increasing the concentration.

Night-time losses of δ-ICN are reaction with O$_3$ and NO$_3$. The MCM uses a rate coefficient of $2.4 \times 10^{-17}$ $cm^3$ $s^{-1}$ for the reaction of δ-ICN with O$_3$, which is 5 times faster than the rate of $4.4 \times 10^{-18}$ $cm^3$ $s^{-1}$ recommended by W2018, giving a partial lifetime on the order of 12 hours for an O$_3$ mixing ratio of 40 ppb. On the other hand, the MCM uses a rate for the reaction of δ-ICN with NO$_3$ which is 10 times slower than the rate recommended by W2018, but even so the lifetime of δ-ICN with respect to reaction with NO$_3$ as estimated by W2018 is of the order of 4 days, so this loss pathway would have to

be much faster to reduce the modelled night-time δ-ICN to close to that observed.

**6.4 Propanone nitrate**

Figures 18 and 19 show the time series and diel patterns of the measured and modelled propanone nitrate. The observed

mixing ratios are generally higher than the modelled values, which may be due to uncertainties in the measurement calibration (Sect. 3.2). Looking at the diel patterns (Fig. 19) there are some similarities in that both the measured and modelled values show a bi-modal pattern with maxima at night and during the daytime. The modelled night-time maximum is much greater than its daytime maximum due to the event in the early hours of the 16$^{th}$ June 2017 during which the modelled mixing ratios of propanone nitrate exceeded 1.5 ppb (Sect. 6.6). Removing this event from the modelled means

gives a night-time maximum similar to the daytime one, which is consistent with the observed pattern. However, whilst the modelled 15 minute means approach zero at around 08:00 and 20:00 local time, the average observed hourly values remain reasonably high throughout the day (>30 ppt). Fig. 19 illustrates that the observed propanone nitrate mixing ratios are at times as low as a few ppt, but that higher values occurred at most times of the day, leading to a weak diel pattern. This may



be due to a lack of observations, along with daily variability, but looking at the results on individual days shows that the
observations often remain high whilst the modelled values drop to much lower values and generally there is little correlation
between the observed and modelled values.

Both the model and observed values indicate production of propanone nitrate during day and night. The main source of
propanone nitrate is via oxidation of isoprene, with routes via both OH and $NO_3$ addition to isoprene. Its primary source in
the MCM simulation is via the OH oxidation of NC4CHO (i.e. the δ-ICN) and this is reflected in them sharing many
similarities in their modelled time series (Fig. 18). As discussed in Sect. 6.3, the production of NC4CHO and its loss via OH
oxidation occur mostly during the daytime, so this source of propanone nitrate is predominantly during the daytime. On
nights when OH is present even at low concentrations it can be a sizeable source due to the relatively large amounts of
NC4CHO at night. Propanone nitrate is also formed from oxidation of NC4CHO by $O_3$. This is a relatively small source
except on nights when $O_3$ was not depleted (Fig. 4). (4-OH, 1-$ONO_2$)-IHN is also a source of propanone nitrate.

The modelled ratio propanone nitrate to δ-ICN is mostly much less than one (note the vertical axes scales in Fig.  18 are
different by a factor of 5). Conversely the measured propanone nitrate is typically a lot greater than the total δ-ICN observed
(Figs. 19 and 17). As discussed in Sect. 6.3, the model simulates considerably larger amounts of NC4CHO than observed
and getting the wrong balance between the various production and loss terms of NC4CHO (i.e. the δ-ICN) will likely impact
the modelled propanone nitrate.

Propanone nitrate can also be produced following the $NO_3$ addition to propene. This acts predominantly at night-time, but
the model results suggest it to be a relatively small source in this campaign.
Whilst the main source of propanone nitrate is during the daytime, the dominant sink in the model is the ventilation which
also maximises during the daytime, leading to a diel profile with similar magnitude daytime and night-time maxima (Fig.
19). As discussed above the evening ventilation might be overestimated, which could contribute to the discrepancy between
the modelled and observed propanone nitrate.

**6.5 δ-IHN**

The MCM simulates daytime peak mixing ratios for the δ-IHN (i.e. (1-OH, 4-$ONO_2$)-IHN and (4-OH, 1-$ONO_2$)-IHN),
consistent with production from OH addition to isoprene, of around 1-3 ppt (Fig. 20). However, as mentioned above, we
were unable to detect these IN in Beijing despite having been able to in the laboratory. The two δ-IHN are simulated to have
very similar mixing ratios during the daytime, but on several nights, enhancements of (4-OH, 1-$ONO_2$)-IHN are simulated





and these are coincident with enhanced modelled mixing ratios of the δ-ICN. This is illustrated both in the time series plot (Fig. 20) and diel plot (Fig. 21). Note that an event in which the modelled mixing ratios of (4-OH, 1-ONO$_2$)-IHN reached 0.5 ppb in the early hours of the 16$^{th}$ June 2017 (Sect. 6.6) skewed the night-time 15 minute mean values. Removing this event

from the modelled means gives a night-time maximum similar to the daytime one. As well as being formed by OH oxidation of isoprene, (4-OH, 1-ONO$_2$)-IHN is also formed in the MCM when the NISOPO2 radicals produced by NO$_3$ oxidation of isoprene react with other organic peroxy radicals. As discussed above in Sect. 6.3, NISOPO2 are mostly present during the daytime, but at that time this source of (4-OH, 1-ONO$_2$)-IHN is small compared to that from OH addition to isoprene. However, on certain nights NISOPO2 mixing ratios were simulated to be high leading to elevated mixing ratios of both (4-

OH, 1-ONO$_2$)-IHN and δ-ICN. Only a few of the simulated night-time peaks occurred at the time when we were making measurements, but we would have expected to detect (4-OH, 1-ONO$_2$)-IHN at the mixing ratios simulated (around 15-30 ppt). This adds further weight to the idea that the NO$_3$ production of the IN (δ-ICN and δ-IHN) may be overestimated in the model.

**6.6 Modelled event on 16$^{th}$ June 2017**

Between 01:00 and 02:00 local time in the morning of 16$^{th}$ June a spike in the observed isoprene mixing ratio of 7 ppb occurred. We believe this spike to be from a local anthropogenic source and probably very short-lived, but this led to elevated concentrations of isoprene in the model for a period of over an hour at a time when the observationally constrained

NO$_3$ mixing ratios were around 35-80 ppt. This led to high modelled concentrations of NISOPO2 (up to 3.1 x 10$^9$ cm$^{-3}$) and other organic peroxy radicals (1.1 x 10$^{10}$ cm$^{-3}$), leading to very high mixing ratios of both NC4CHO (~10 ppb), propanone nitrate (>1.5 ppb) and (1-OH, 4-ONO2)-IHN (0.5 ppb). Unfortunately, we did not have our system operating in the mode required to measure the ICN or δ-IHN at that time and nor were we measuring RO$_2$.

**7 Conclusions**

Observed mixing ratios of the IN appear to be strongly influenced by atmospheric mixing, which can to a large extent be accounted for in a box model by applying a ventilation term as a function of the mixed layer height. Model simulations of absolute mixing ratios of the IN are therefore sensitive to the ventilation term, but by considering ratios and relative

behaviours of the IN much of this sensitivity is cancelled out.





The observed β-IHN mixing ratios peak around midday and these levels are maintained until around sunset when they then decline to reach minimum values just after sunrise. This pattern is broadly similar to that observed during SOAS for total IHN but peak later in the day probably due to NO mixing ratios in Beijing remaining relatively higher into the afternoon and

so favouring the reaction of β-ISOPOO with NO over that with $HO_2$ for longer into the day. The two β-IHN are well correlated with an $R^2$ value of 0.85. The mean for the ratio (1-OH, 2-$ONO_2$)-IHN : (4-OH, 3-$ONO_2$)-IHN is 3.4 (standard deviation of 1.7) and exhibits no clear diel cycle. It should be noted that the ratio we obtain from our measurements is not based on an independent calibration for (1-OH, 2-$ONO_2$)-IHN, but based on the assumption that the analytical system has the same sensitivity to it as it does to (4-OH, 3-$ONO_2$)-IHN.


Examining the ratio of the two β-IHN in a box model demonstrates its sensitivity to NO, which affects the thermodynamic equilibrium of the β-ISOPOO and the competition between the reactions of the β-ISOPOO with NO and with $RO_2$, with lower NO mixing ratios favouring (1-OH, 2-$ONO_2$)-IHN over (4-OH, 3-$ONO_2$)-IHN in both cases. Interestingly the modelled ratio of (1-OH, 2-OO)-ISOPOO to (4-OH, 3-OO)-ISOPOO exceeds the kinetic ratio even at NO mixing ratios as

high as 100 ppb. This ratio, however, varies little for NO mixing ratios greater than 2 ppb and it is only below 1 ppb when this ratio increases substantially with decreasing NO. The MCM model underestimates the measured $RO_2$ mixing ratios so this will lead to an underestimation of the competition from the $RO_2$ for reaction with the β-ISOPOO. Consequently, this may cause modelled ratios of (1-OH, 2-$ONO_2$)-IHN to (4-OH, 3-$ONO_2$)-IHN to be underestimated, particularly at mixing ratios of NO below ~2 ppb. The relationship of the observed β-IHN ratio with NO is much weaker than modelled, partly due

to far fewer data points, but it agrees with the model simulation in so far as there tend to be larger ratios (up to ~6) at sub 1 ppb amounts of NO.

(1-OH, 2-$ONO_2$)-IHN also reacts more slowly with OH than does (4-OH, 3-$ONO_2$)-IHN and since this is the dominant chemical loss process for the β-IHN, the effect is to increase the ratio of (1-OH, 2-$ONO_2$)-IHN to (4-OH, 3-$ONO_2$)-IHN. The

simple model results demonstrate that the sensitivity of the β-IHN ratio to uncertainties in the ratio of these OH rate coefficients is of the same order as the difference between the modelled and observed β-IHN ratios.

Of the δ-ICN, the two *trans* isomers are observed to have the highest mixing ratios, with E-(1-$ONO_2$, 4-CO)-ICN being the most abundant. However, the mean C1:C4 isomer ratio is 1.4, which is considerably lower than would be expected based

solely on the addition of $NO_3$ to isoprene occurring in the C1 and C4 positions in a 6:1 ratio. This raises the question as to whether it is appropriate to represent the δ-ICN by a single C1 nitrated isomer, as done in the MCM, and will be depend on how similar their fates are. We observed the *trans*-ICN isomers to dominate over the *cis*-ICN isomers with a mean ratio of 7 far greater than the *trans*:*cis* ratio of 1 presumed by W2018 for the reaction of $NO_3$ addition to isoprene. This suggests that thermodynamic redistribution of the nitrated peroxy radicals formed from the reaction of the $NO_3$-isoprene adducts with $O_2$

may also be important.

The δ-ICN exhibit nocturnal peaks with maximum values in the early night and minimum values during the daytime consistent with formation from $NO_3$ addition to isoprene in the evening and a lifetime of the order of a few hours or less. Mixing ratios of 1-2 ppt persist through the daytime, which for the afternoon can be accounted for by the presence of 1-2 ppt

of $NO_3$. The MCM produces far more δ-ICN than observed, particularly at night but it also simulates an increase in the daytime δ-ICN that greatly exceeds that seen in the observations. Reaction of NO with NISOPO2, which comes from $NO_3$ addition to isoprene, is by the far the dominant source of δ-ICN in the MCM but, interestingly, it is predominantly during the daytime, due to the presence in Beijing of appreciable daytime amounts of $NO_3$ along with isoprene.

Observed and modelled propanone nitrate shows no clear diel cycle. The pattern can change from day to day, sometimes peaking during the daytime and sometimes at night. The modelled propanone nitrate is considerably less than observed. This might be due to large uncertainties in the measurement calibration, but it could also be linked to issues with the simulation of its precursor, δ-ICN, with the modelled ratios of δ-ICN to propanone nitrate being very different to the observed.

The main source of the δ-IHN is modelled to come from OH addition to isoprene, but on certain nights the source from $NO_3$ addition to isoprene led to mixing ratios of around 15-30 ppt of (4-OH, 1-$ONO_2$)-IHN coincident with high mixing ratios of δ-ICN. We were unable to detect δ-IHN despite the modelled mixing ratios being considerably greater on these nights than those we would expect to observe. This warrants further investigation.

This study demonstrates the value of speciated IN measurements to test our understanding of the isoprene degradation chemistry. Our interpretation is limited by the uncertainties in our measurements and relatively small data set, but highlights areas of the isoprene chemistry that warrant further study, in particular the $NO_3$ initiated isoprene degradation chemistry.

*Code Availability.* The MCM code is available from the authors on request.


*Data Availability*. The observational data, simple model data and diel cycles from the MCM in the figures are in the Supplementary Information. The 15 minute data from the MCM is available from the authors on request.

*Supplement.*


*Author contributions.* CER led the data interpretation and writing of the manuscript. GPM made the measurements of the IN with the assistance of YL. LKW did the MCM modelling. CER, WJB, SG, DEH, CNH, RLJ, JDL, XW and CY were involved in the project planning and leading the measurement groups. WJA, LRC, JRH, SK, LJK, BO, ES, FS and RW-M provided measurement data. All commented on the manuscript.




*Competing interests.* The authors declare that they have no conflict of interest.

*Acknowledgements.* We are grateful for funding provided by the UK Natural Environment Research Council (NERC), UK Medical Research Council and the Natural Science Foundation of China (NSFC) under the framework of the Newton
Innovation Fund (NERC grants NE/N006909/1, NE/N006895/1, NE/N006976/1 and NE/N00700X/1; NSFC grant 41571130031). ES and RW-M are grateful to the NERC SPHERES Doctoral Training Programme for funding PhD studentships. CER acknowledges Andrew Rickard (NCAS, University of York) for providing information on the MCM.

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





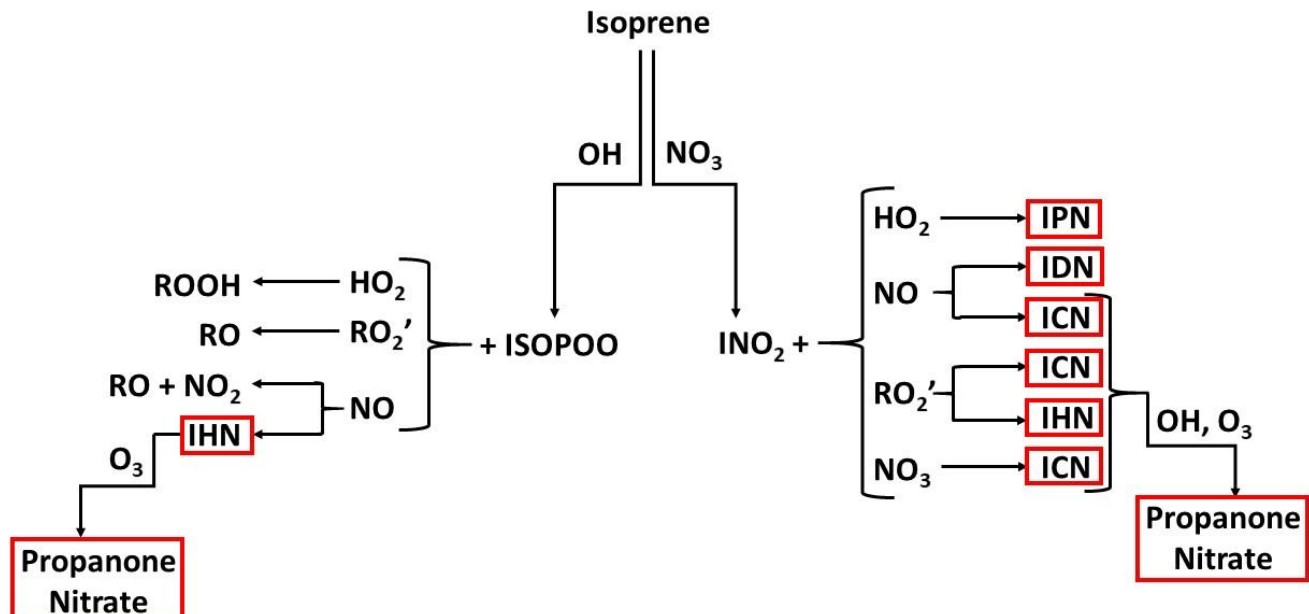


**Figure 1: Formation of IN (red boxes) from isoprene oxidation by OH and NO₃: isoprene hydroxy nitrates (IHN): isoprene hydroperoxy nitrates (IPN); isoprene dinitrates (IDN); isoprene carbonyl nitrates (ICN); and propanone nitrate.**





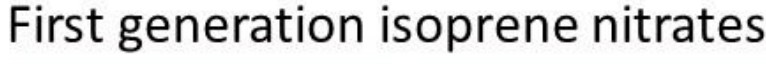

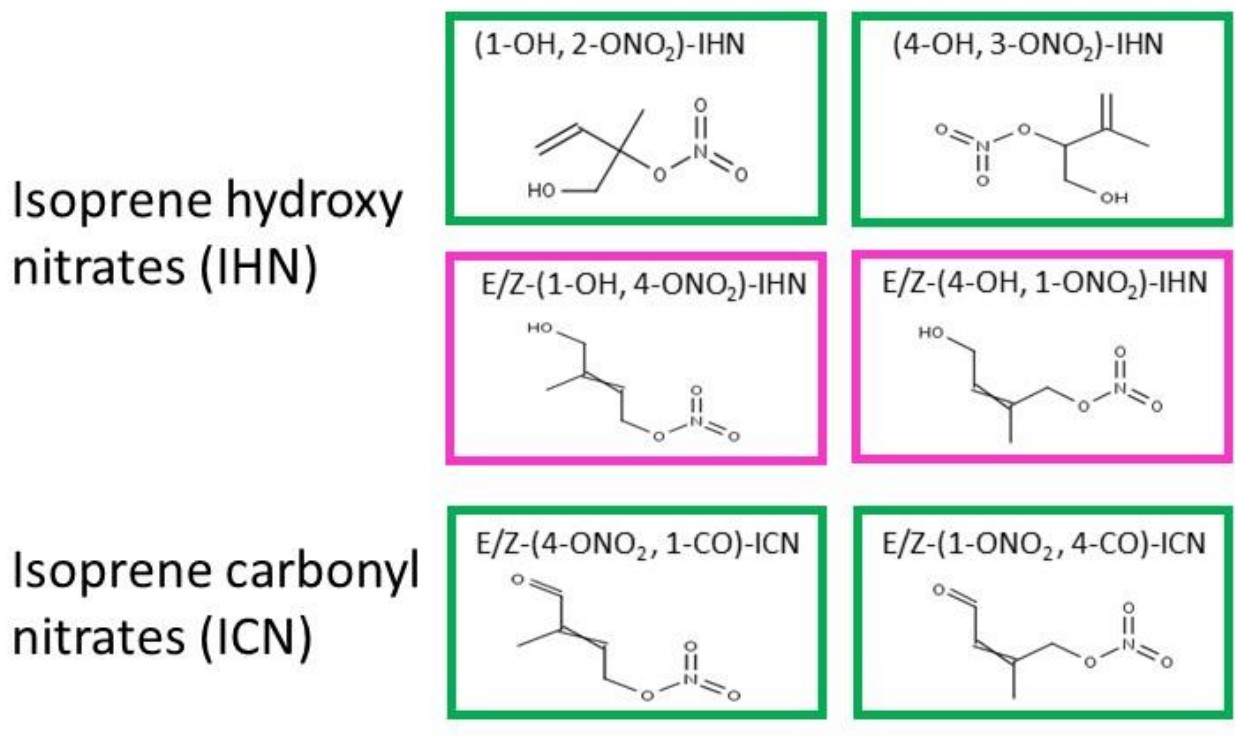

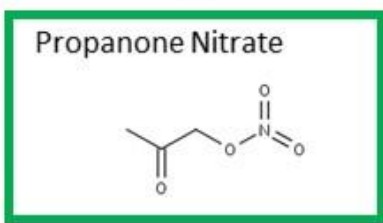


**Figure 2: Isoprene nitrates. Box colours: Green - measured in Beijing; Pink - measured by the analytical system previously in the laboratory, but not discernible in Beijing.**





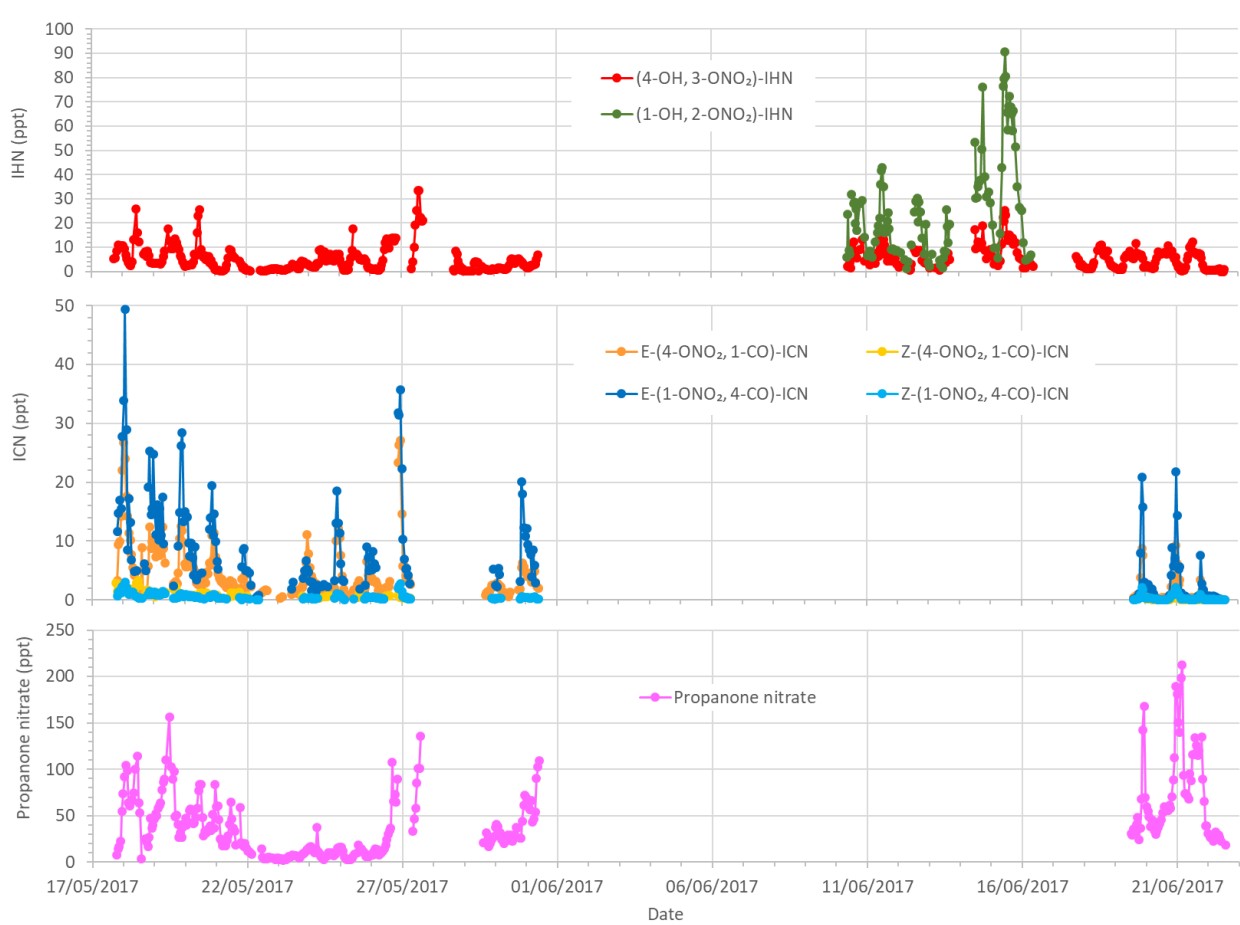


**Figure 3: Isoprene nitrates mixing ratios measured in Beijing.**





**Figure 4: Isoprene, CO, NO, NO₂ and O₃ mixing ratios measured in Beijing for the times corresponding to the IN data shown in Fig. 3.**





**Figure 5: Measured β-IHN mixing ratios and their ratio, (1-OH, 2-ONO₂)-IHN:(4-OH, 3-ONO₂)-IHN.**




**Figure 6: Measured δ-ICN mixing ratios and their ratios during the last four days of the summer campaign.**





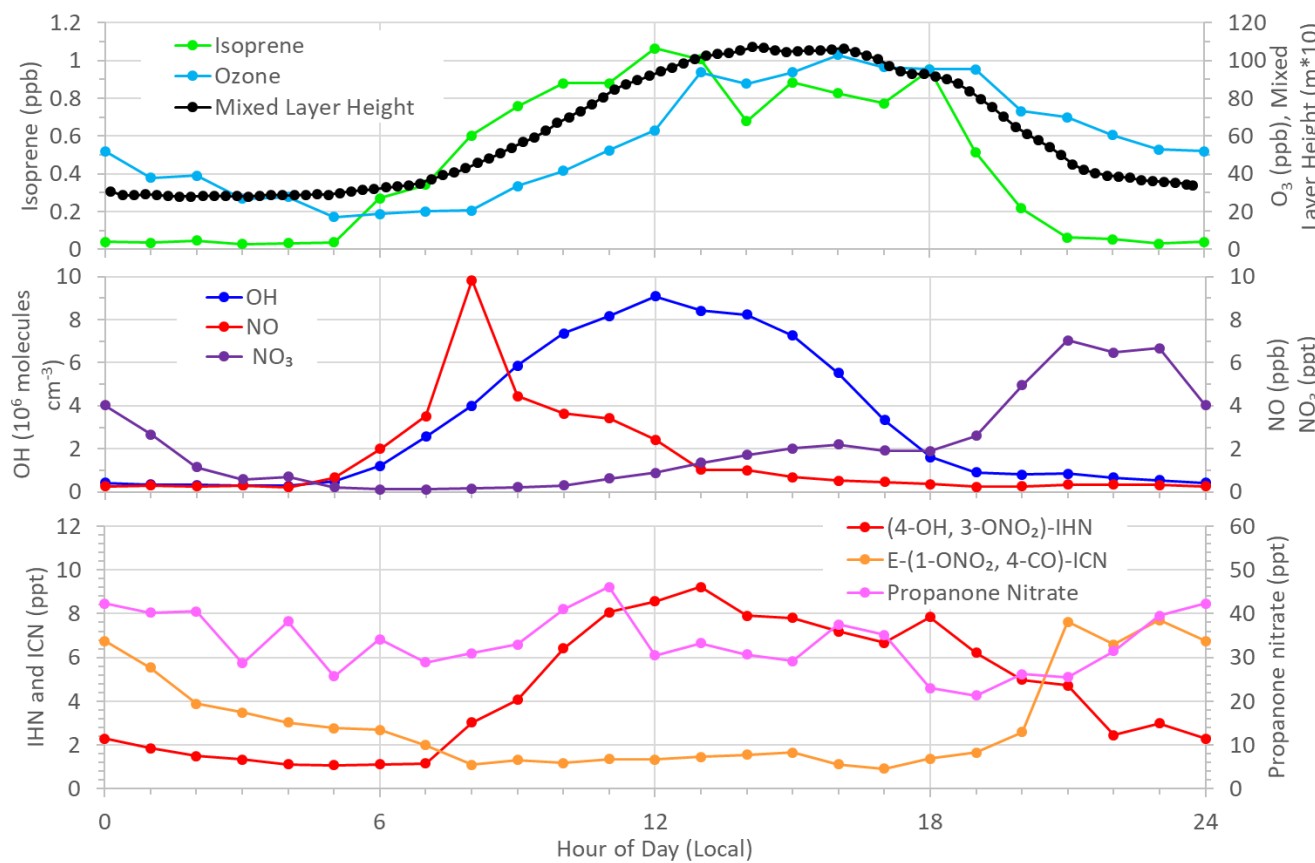

**Figure 7: Diel patterns of trace gases derived from the medians of the measured mixing ratios for each hour of the day. The observed 15 minute mean mixed layer heights used as input to the MCM model.**



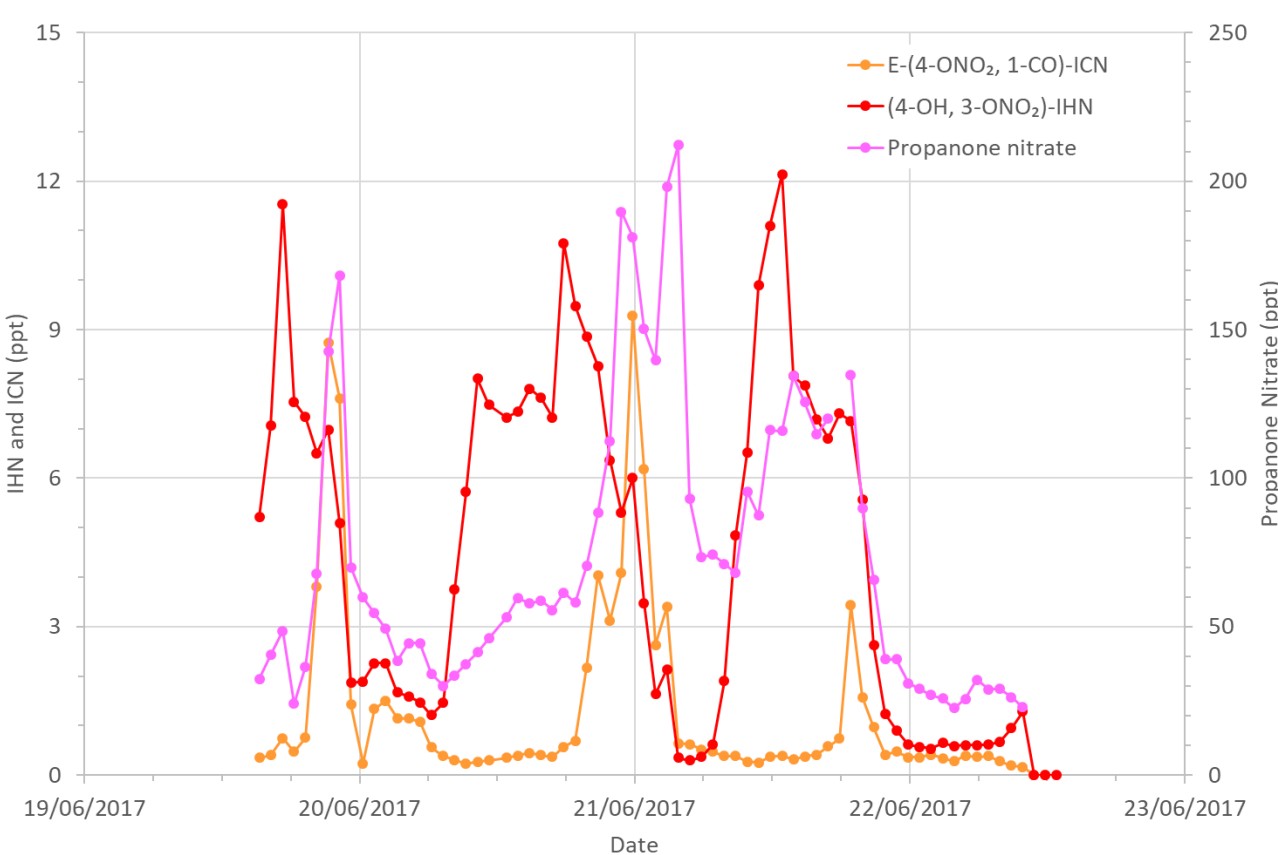


**Figure 8: Measured mixing ratios of (4-OH, 3-ONO₂)-IHN, E-(4-ONO₂, 1-CO)-ICN, and propanone nitrate during the last five days of the campaign.**





**Figure 9: Bottom row: Modelled and observed mixing ratios of (4-OH, 3-ONO₂)-IHN (left) and (1-OH, 2-ONO₂)-IHN (right). Top row: Calculated production rates of (4-OH, 3-ONO₂)-IHN (left) and (1-OH, 2-ONO₂)-IHN (right).**





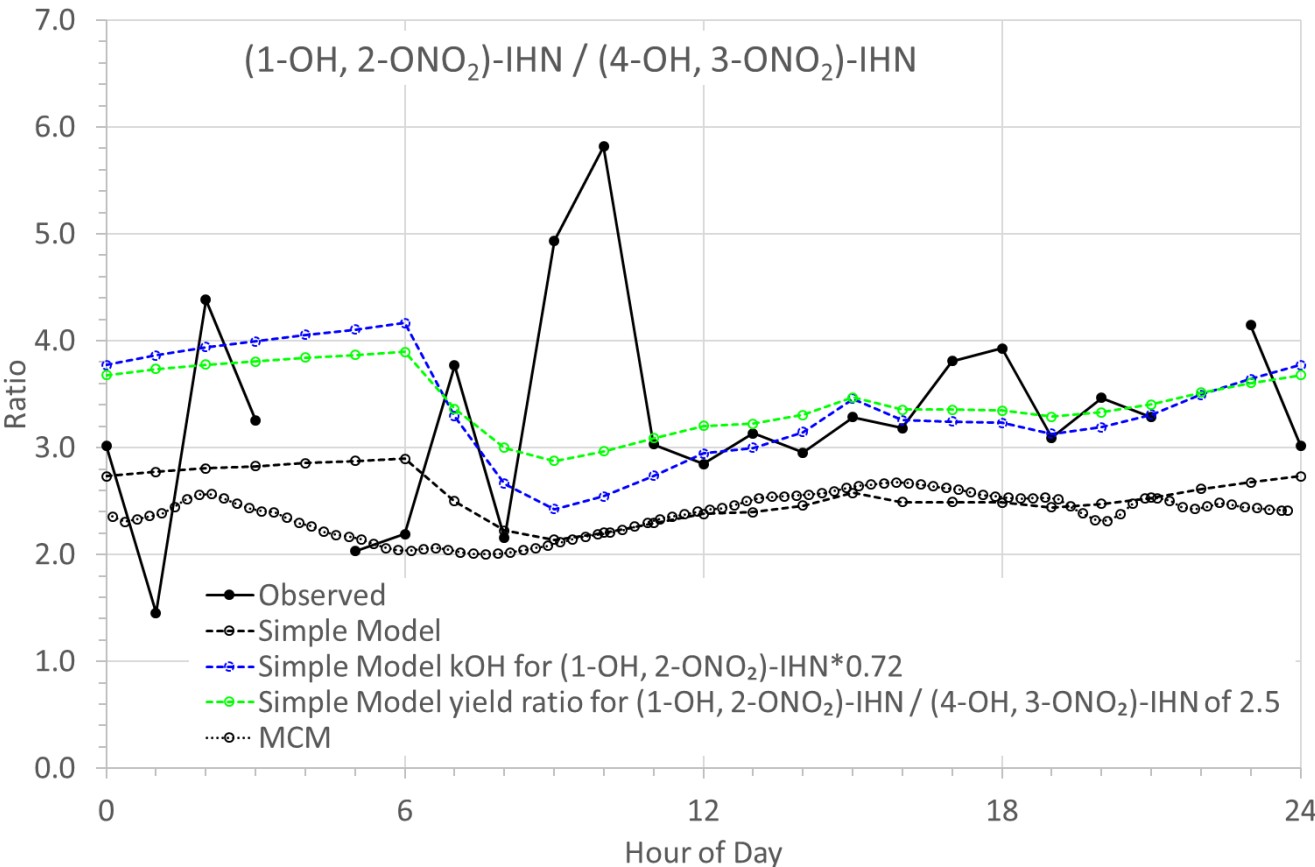

**Figure 10: Modelled and observed (1-OH, 2-ONO₂)-IHN / (4-OH, 3-ONO₂)-IHN ratio. Coloured lines come from**
**sensitivity runs in which the rates of production or losses of each of the β-IHN have been changed in the simple**
**model. The MCM run is described in Sect. 6.**





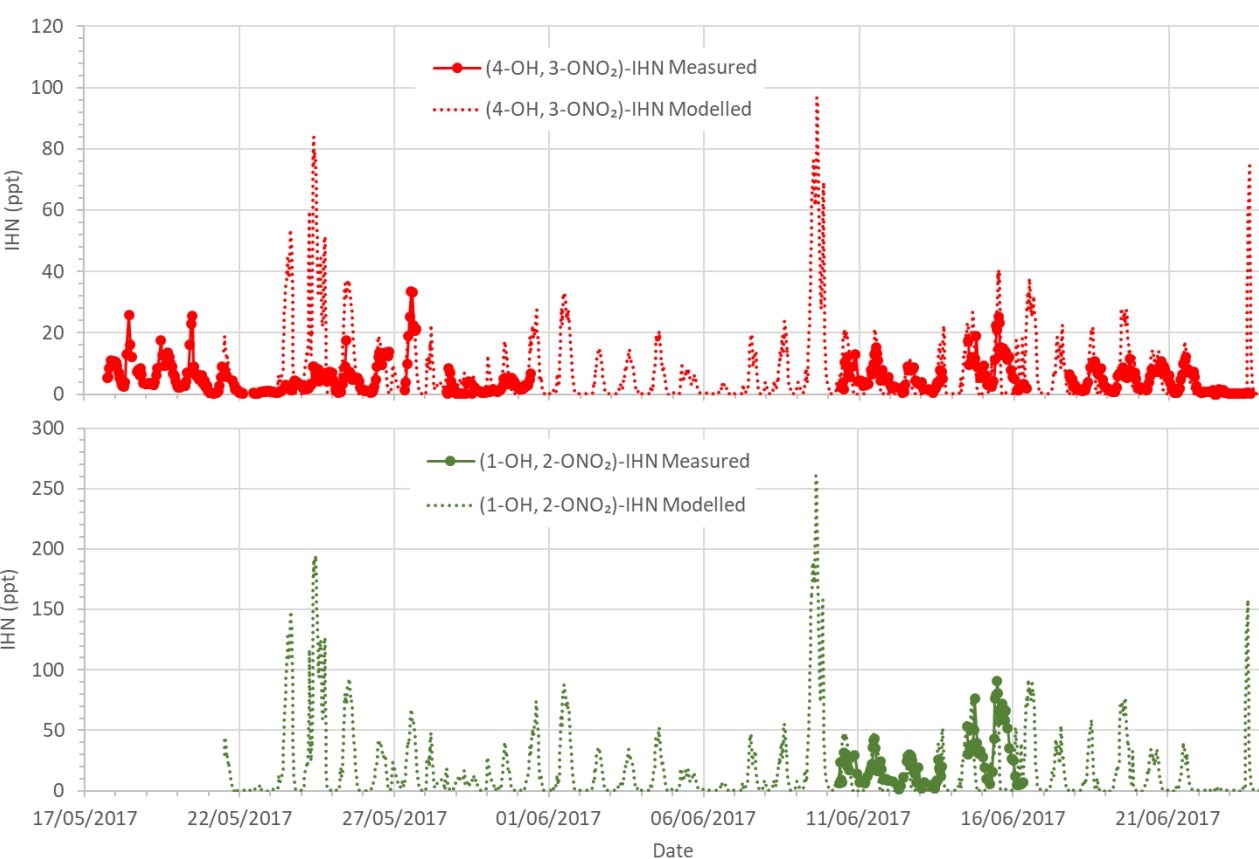

**Figure 11: Time series of β-IHN as modelled using the MCM and measured.**



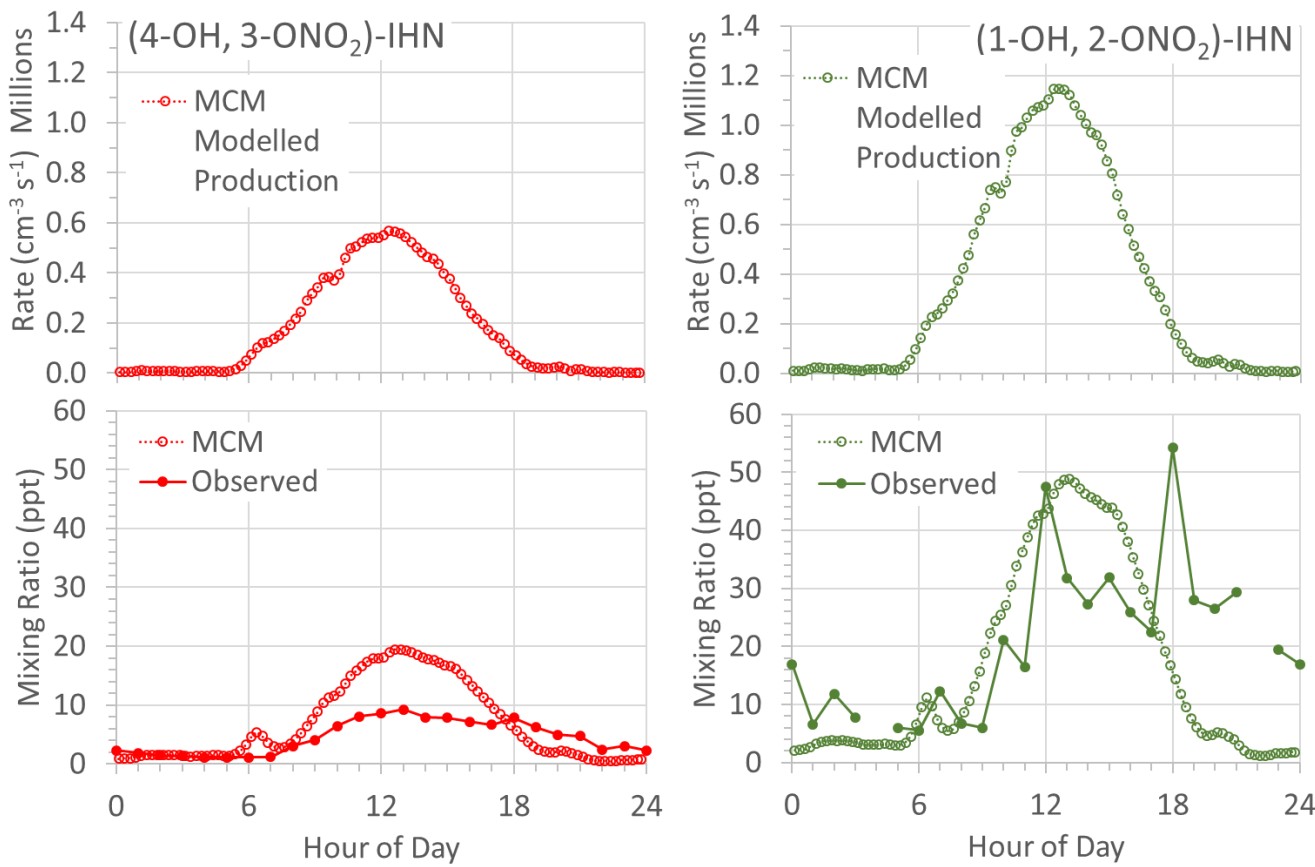

**Figure 12: Diel patterns of β-IHN as modelled using the MCM and measured. Bottom row: MCM modelled and observed (4-OH, 3-ONO₂)-IHN (left) and (1-OH, 2-ONO₂)-IHN (right). The modelled values are means for each 15 minute period of the day for the whole modelled period, whereas the observed values are hourly medians. Top row: Modelled production rates of (4-OH, 3-ONO₂)-IHN (left) and (1-OH, 2-ONO₂)-IHN (right).**


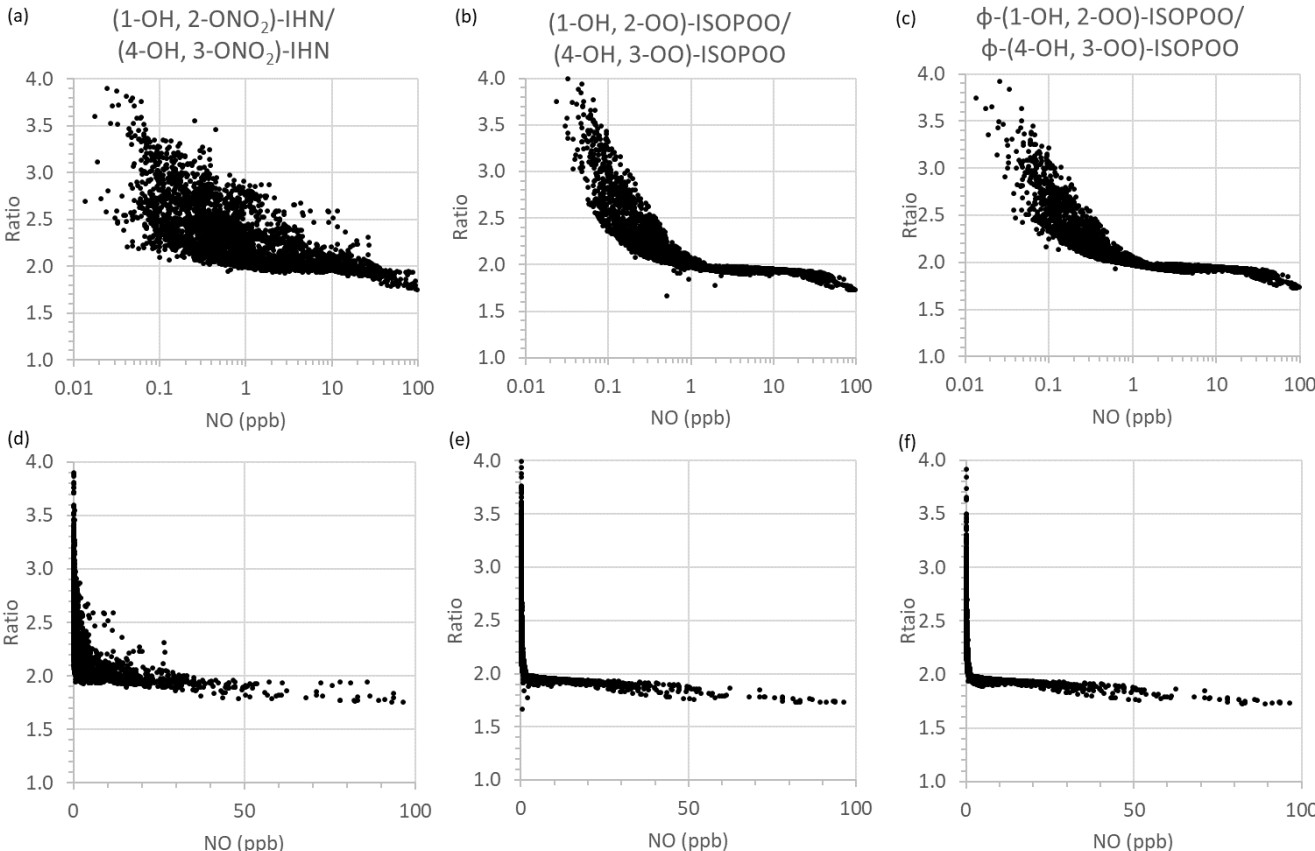

**Figure 13: MCM modelled parameters as a function of NO mixing ratio (top row logarithmic scale, bottom row linear scale). Left column: the ratio of (1-OH, 2-ONO₂)-IHN to (4-OH, 3-ONO₂)-IHN. Middle panel: the ratio of (1-OH, 2-OO)-ISOPOO to (4-OH, 3-OO)-ISOPOO. Right panel: the ratio of φ-(1-OH, 2-OO)-ISOPOO to φ- (4-OH, 3-OO)-ISOPOO.**



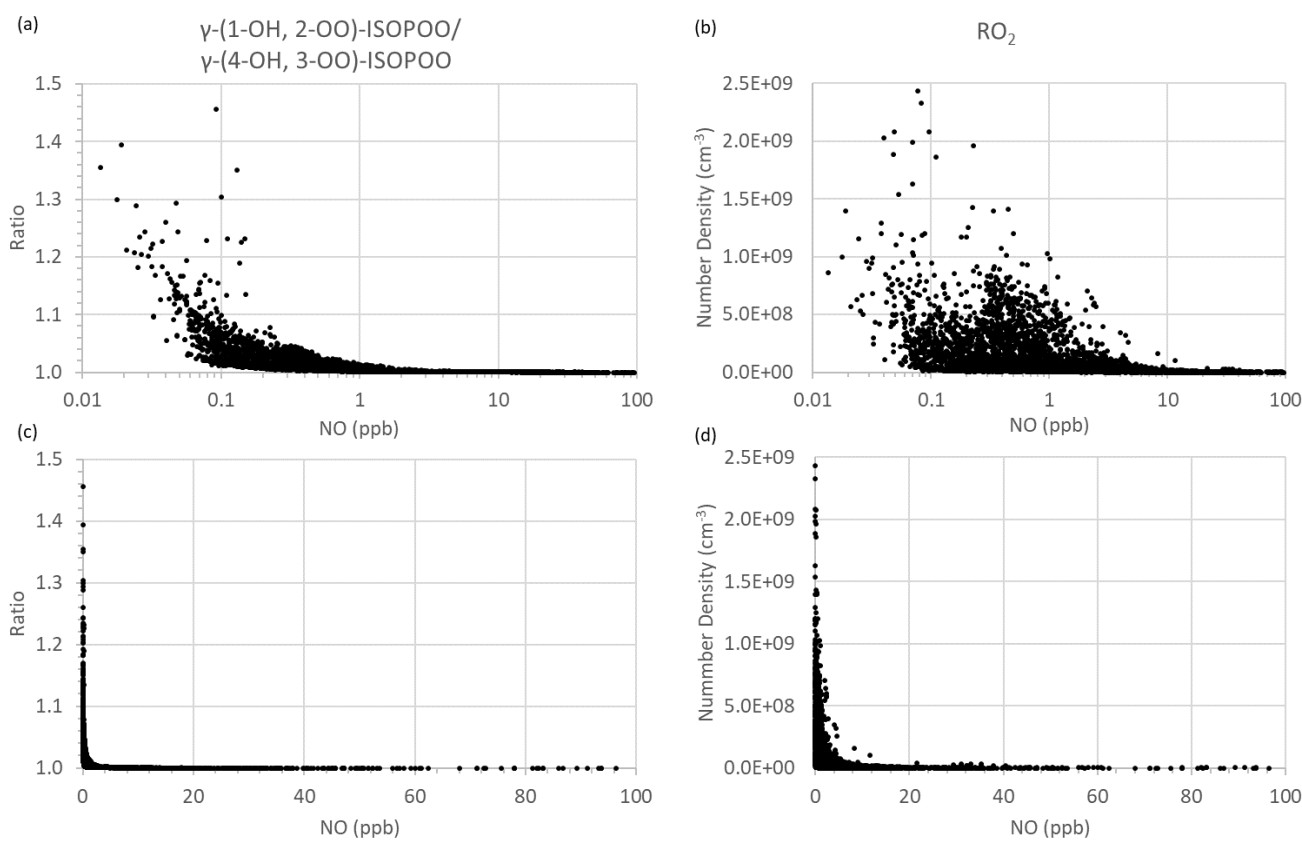

**Figure 14: MCM modelled parameters as a function of NO mixing ratio (top row logarithmic scale, bottom row linear scale). Left column: the ratio of γ-(1-OH, 2-OO)-ISOPOO to γ-(4-OH, 3-OO)-ISOPOO. Right panel: RO₂ number density.**





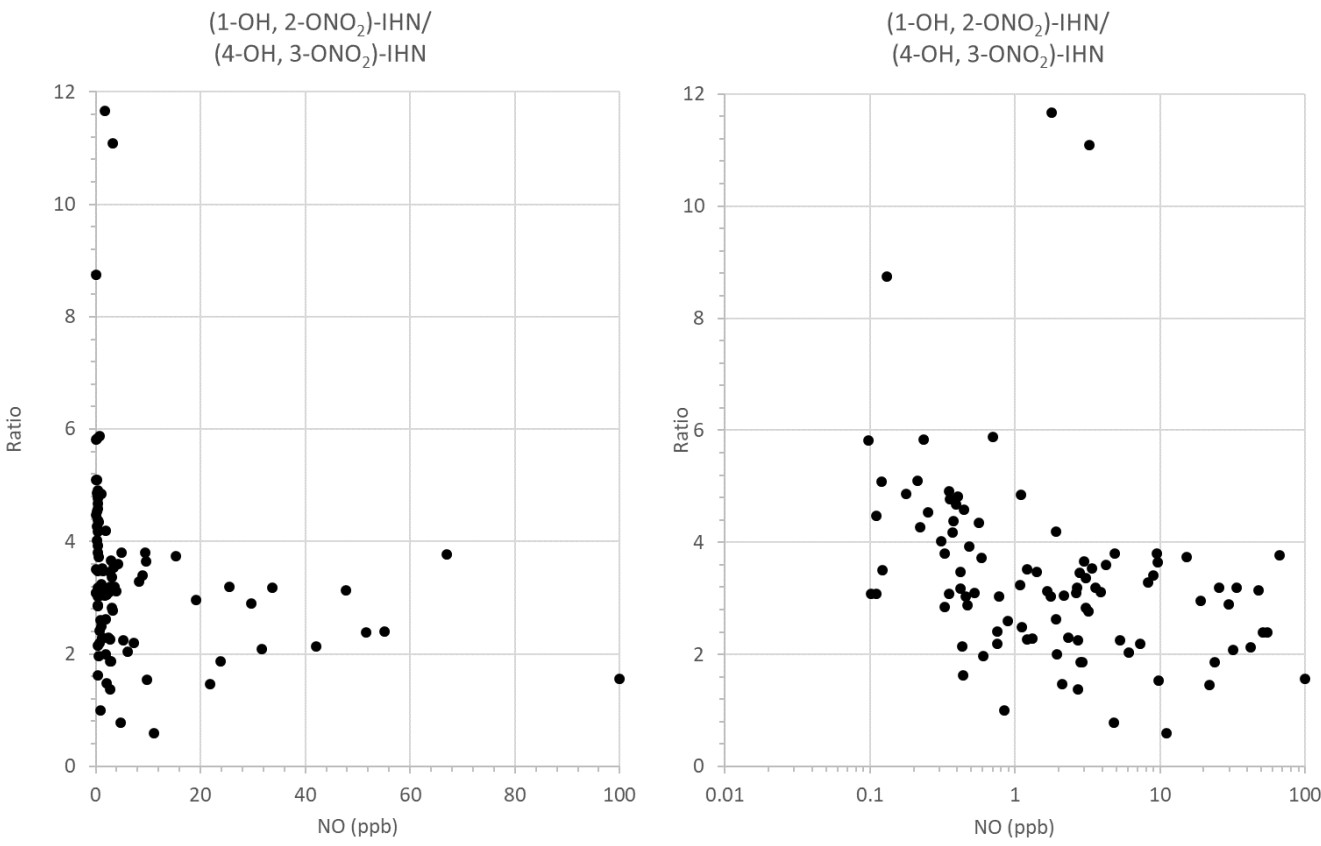


**Figure 15: Observed the ratio of (1-OH, 2-ONO₂)-IHN to (4-OH, 3-ONO₂)-IHN as a function of NO mixing ratio with (left) a logarithmic and (right) a linear x-axis.**





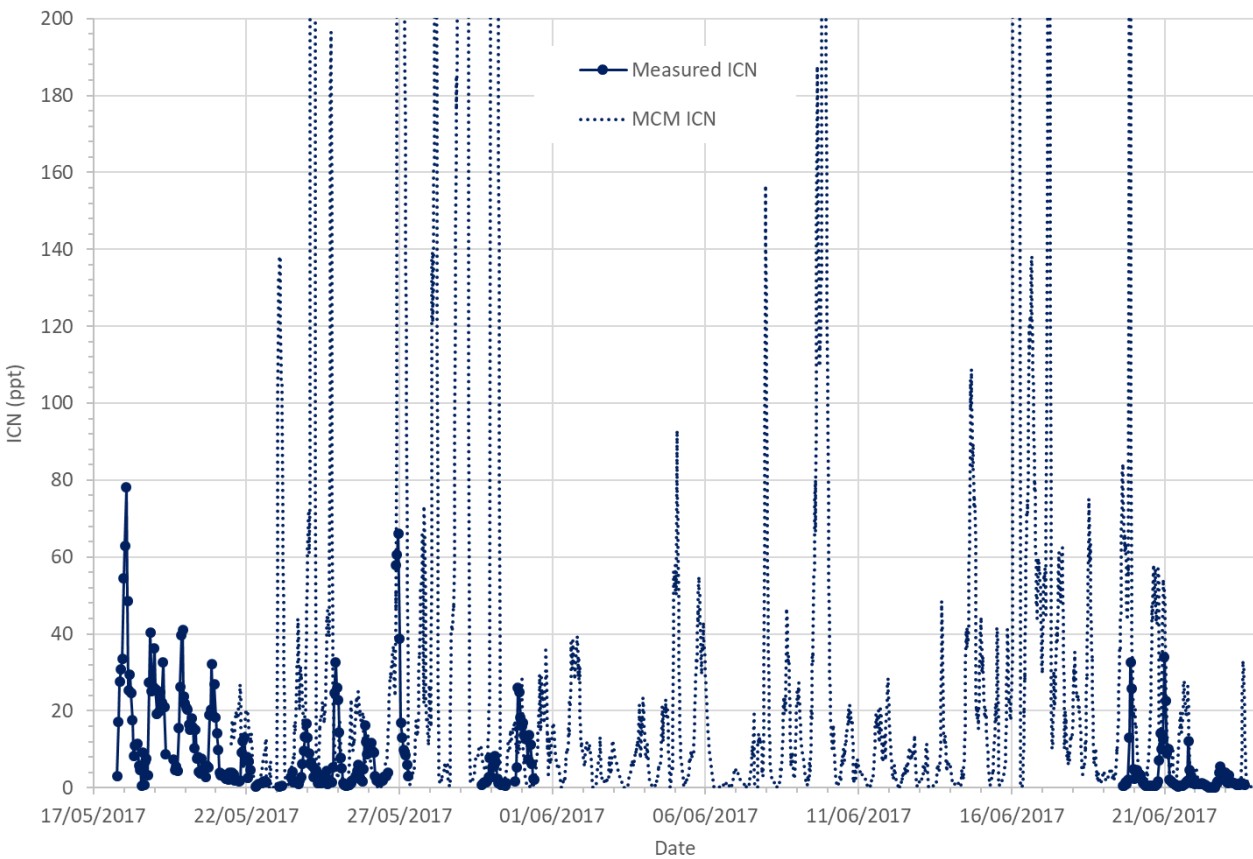

**Figure 16: Time series of total δ-ICN as modelled using the MCM and measured. For the MCM this is the species NC4CHO, whilst the measurements are the sum of the four δ-ICN (E and Z-(1-ONO₂, 4-CO)-ICN and E and Z-(4-ONO₂, 1-CO)-ICN).**





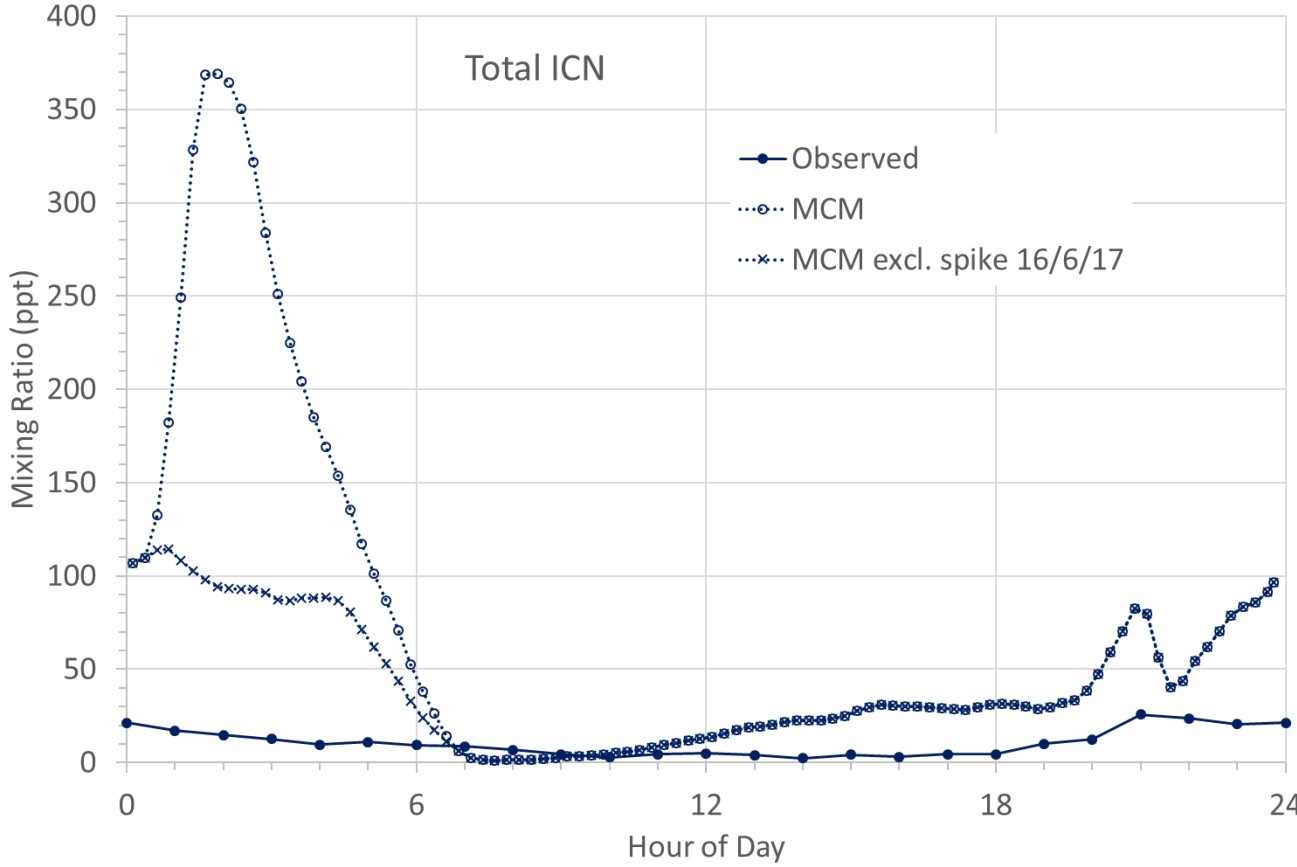

**Figure 17: Diel pattern of total ICN as modelled using the MCM and measured. For the MCM this is the specie NC4CHO, whilst the measurements are the sum of the four δ-ICN (E and Z-(1-ONO₂, 4-CO)-ICN and E and Z-(4-ONO₂, 1-CO)-ICN). The dotted line with crosses is calculated with the spike in the modelled ICN in the early morning of 16th June 2017 removed. The modelled values are means for each 15 minute period of the day for the whole modelled period.**





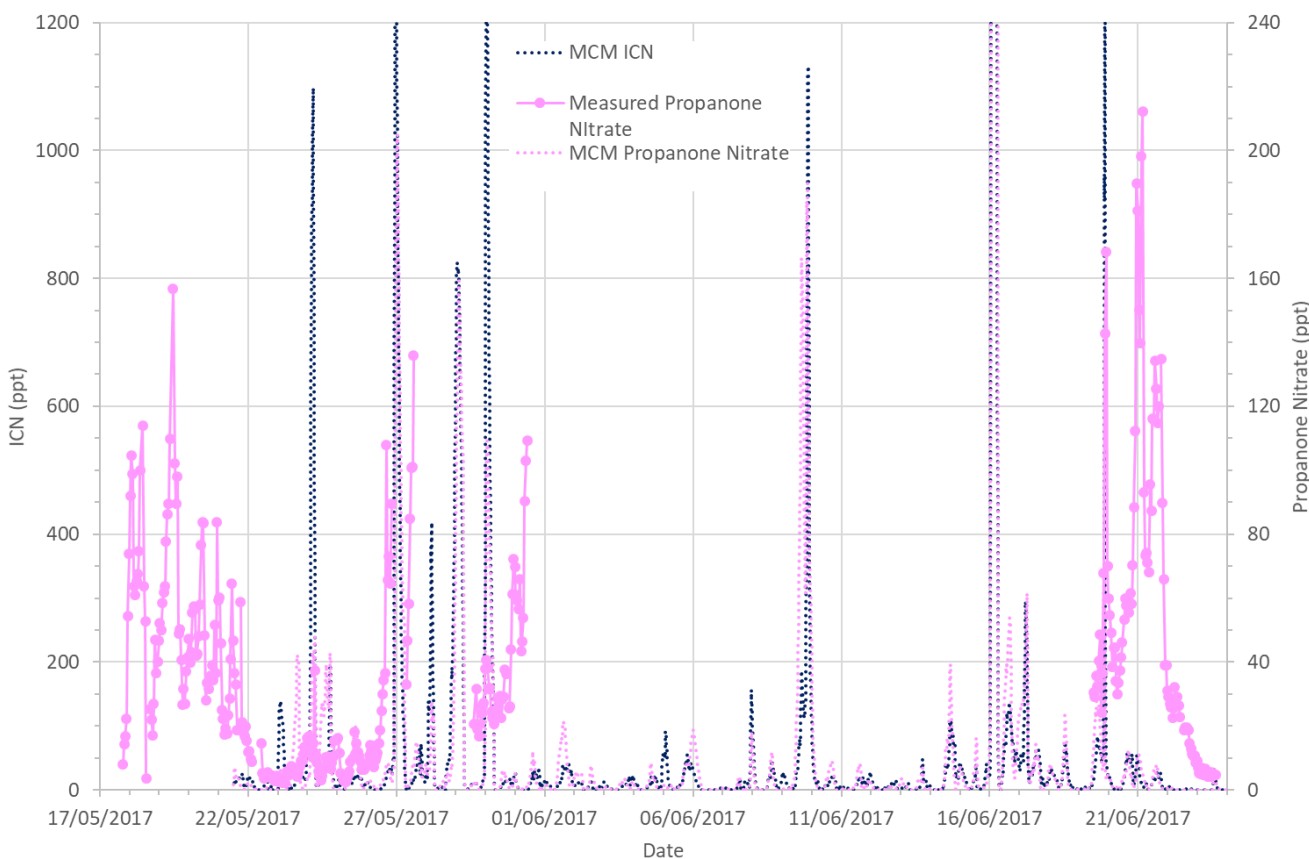

**Figure 18: Time series of propanone nitrate as modelled using the MCM and measured, along with modelled δ-ICN (NC4CHO).**



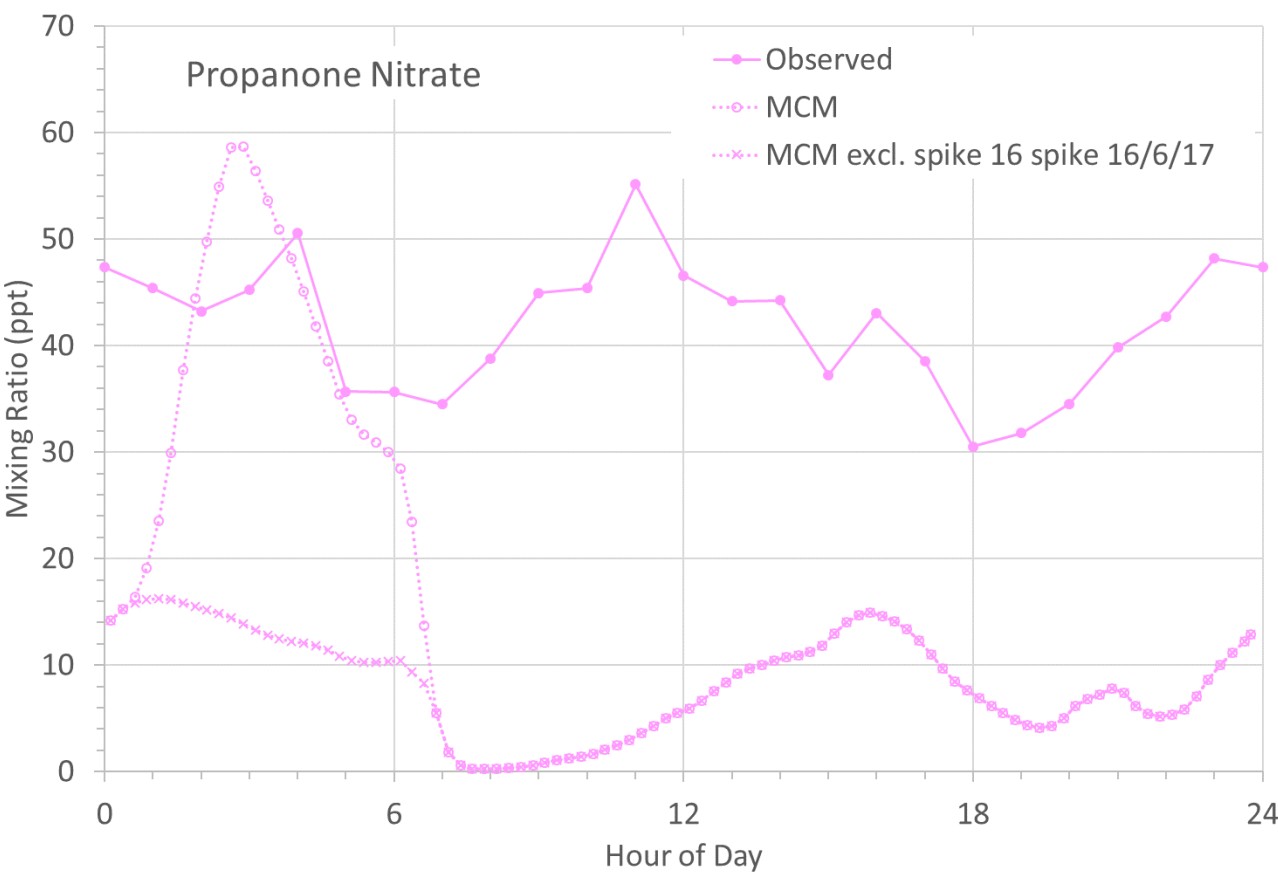

**Figure 19: Diel pattern of propanone as modelled using the MCM and measured. The dotted line with crosses is calculated with the spike in the modelled propanone nitrate in the early morning of 16[th] June 2017 removed. The modelled values are means for each 15 minute period of the day for the whole modelled period.**





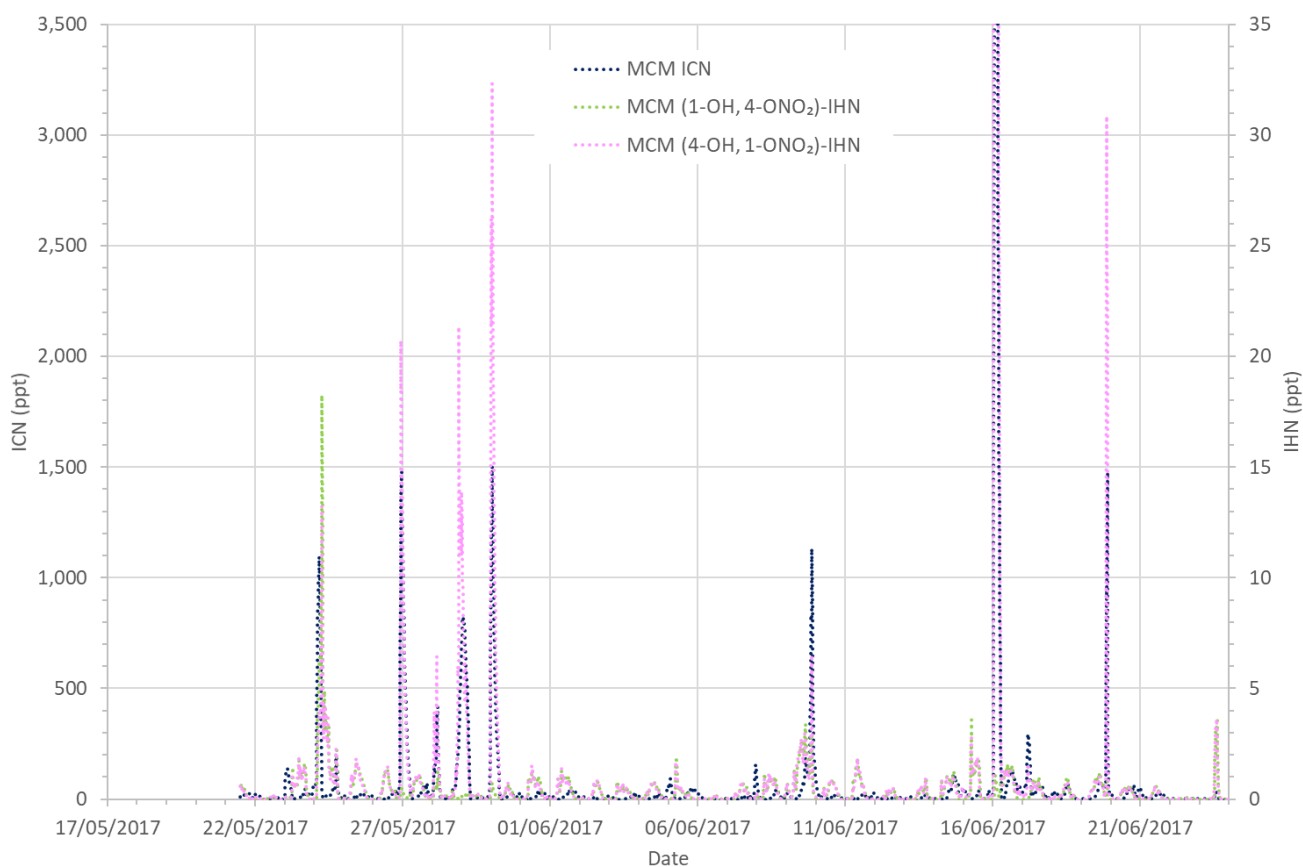

**Figure 20: Time series the MCM modelled δ-IHN ((1-OH, 4-ONO₂)-IHN and (4-OH, 1-ONO₂)-IHN) and δ-ICN (NC4CHO).**





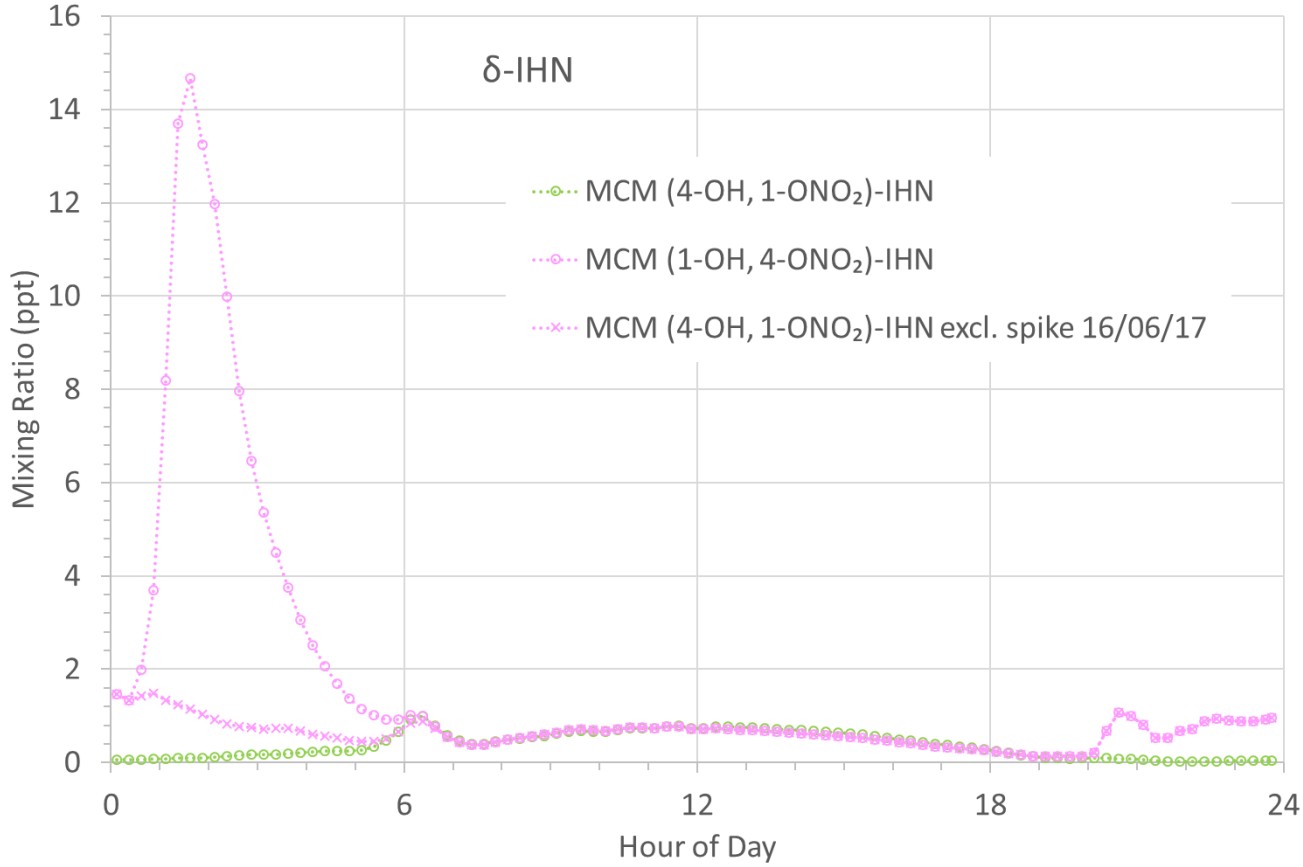


**Figure 21: Diel pattern of MCM modelled δ-IHN ((1-OH, 4-ONO₂)-IHN and (4-OH, 1-ONO₂)-IHN). The pink dotted line with crosses is calculated with the spike in the modelled (4-OH, 1-ONO₂)-IHN in the early morning of 16th June 2017 removed. The modelled values are means for each 15 minute period of the day for the whole modelled period.**