# Peer review of "Observations of speciated isoprene nitrates in Beijing: implications for isoprene chemistry"

_Atmospheric Chemistry and Physics, 2019_

## Referee Comment (RC1) · Anonymous Referee #1 · 9 Feb 2020

This data set is likely interesting. However, this paper is long, data rich and is not succinct in its analysis. It is very hard to tell which conclusions are unambiguously supported by the observations and which depend on assumptions about transmission and sensitivity. It is not currently accessible to a general reader of ACP. I recommend it be rejected.

Only the most determined reader will be able to wade through this and find the important information and three years from now, no one will be able to identify key ideas that should stand the test of time from ideas that are momentary arguments about different rates constants in a version of MCM and W2018. Today, no one not deeply steeped in

the isoprene chemistry will be able to read it and recognize the ideas being tested. It would greatly benefit from editing in collaboration with someone who is not as engaged in the details. I recommend it be rewritten with many fewer figures. The figures that remain should be chosen to demonstrate how the observations test competing ideas for the behavior of these nitrates.

In addition, the sections on MCM should be more clearly motivated–are there choices MCM has made that are in conflict with W2018. If so is there a logic to them or is MCM just not updated to be consistent with W2018 yet?

---

## Referee Comment (RC2) · Anonymous Referee #3 · 18 Feb 2020

The paper by Reeves et al. describes measurements of speciated organic nitrates that are produced from both OH and NO3 reaction with isoprene. Using a GC/MS approach, they were able to identify and quantify seven different "isoprene nitrates", specifically, two ïĄć-hydroxy nitrates, four ïĄď-carbonyl nitrates, and propanone nitrate, in Beijing during the winter of 2016 and summer of 2017. Isomers were generally (not always) identified by injections of samples of the individual synthesized isomers, and quantified with reasonable time resolution (it appears to be hourly, but that is not stated clearly in the manuscript; that should be clarified). What resulted was a highly unique data set for these compounds, in an isoprene-impacted urban environment, with very good supporting chemical measurements, including isoprene, NOx, HOx, RO2, NO3, HONO,

and HCHO. Many of these measurements are highly challenging. This makes this a highly unique and useful data set for chemically coupled species, and does indeed represent a great opportunity for testing the mechanism for isoprene photooxidation, and studying the impact of isoprene chemistry on the fate of NOx, and for production of ozone and particulate matter. This paper then should be published, and will be high impact, I believe, once one major flaw in the paper is repaired. Specifically, while the data are compared to simulations using MCM chemistry, for both absolute concentrations and ratios of coupled species, these comparisons are extremely difficult to interpret because there is no uncertainty analysis done for these seven compounds. And that lack of detailed uncertainty analysis is a problem in this case because of all the assumptions made, e.g. that sensitivities are the same for the 4,3-IN and the 1,2-IN, and because of issues related to losses of the compounds, e.g. on valves and other surfaces, that clearly have an impact, and these impacts can be different for different isomers, as the authors recognize. So, while they discuss that looking at ratios of isomer concentrations can remove the complexities of boundary layer dynamics, dilution, and ventilation, there is no discussion of the uncertainties of the ratios presented in the various analysis, discussed at length for figures 5, 6, 10, 13, 14, and 15. So, it is possible that the analyses of the these ratios and comparisons to the models are meaningful, but also possible that they contain systematic errors that make the comparison problematic. With no error bars on any of the data, it is impossible to know if the discussions and conclusions are meaningful. Given the likely very large (impressive!) effort in acquiring these data, this is an unfortunate oversight, and needs to be repaired before this paper is published. I recommend a section that does a detailed error analysis for measurements of each isomer, and presents a calculated uncertainty (which could be concentration-dependent) for each one, and also calculates the uncertainty for the ratios that are compared to MCM. The figures could include representative error bars, either on some points, or use shading to reflect the uncertainties, or some other approach. With this added information, this can be a great paper. I note that the last sentence in the paper says "Our interpretation is limited by the uncertainties in

our measurements and relatively small data set, but highlights areas of the isoprene chemistry that warrant further study, in particular the NO3 initiated isoprene degradation chemistry." This is good to recognize, but the reader has no idea what are the uncertainties in the measurements. Other comments and relatively minor issues are listed below, in the order they arose in the paper.

Comments/issues, in order Abstract – line 32 could say isoprene-derived organic nitrates (the first time)? Line 43 – The observed relationship. . . Line 53 – should say "from" the observed.

Line 92 – This key issue should be explained mechanistically, e.g. showing an example of an alkoxy radical that can decompose, releasing NO2.

Line 167 – sentence needs a period.

Section 3.2 – what do you know about the desorption efficiency from the Tenax trap? Since INs are olefinic, and there is lots of O3, what do you know about ozonolysis during sampling? Is the metal valve the only surface on which INs can be (differentially) lost? How do (will) all these things affect your calculated analytical uncertainties? When you knew you had some loss on the valve, did you apply any correction for this? If not, do you have asymmetric error bars? Lines 204 – 208 – how do these assumptions impact your calculated uncertainties? Line 278 – what exactly is the "large uncertainty"? Without these estimates, comparing to model results is an empty exercise.

Line 295 – should be "of" the summer campaign.

Line 322 – since you mention the "appreciable concentrations of OH at night", and there is a lot of interest in that subject, can you include some representative error bars in Figure 7? The same goes for NO3; I would like to repeat that there is some really lovely data in this paper, but it would help the reader to know things like LODs and uncertainties.

Line 357 – the ratio E-1,4 to E-4,1 is not in Figure 6.

Line 365 – yes, but we don't know what the uncertainties are!

Line 400 – I'll just note that alpha is not known to even two significant figures.

Line 415 – Is it known that the -OH group has no impact? What is the uncertainty here?

Line 417 – how does 4x10-5 s-1 compare to the magnitude of the calculated chemical reaction loss? (since you assume here that all the loss is uptake)

Line 443 – Is the upwind environment chemically comparable on a timescale relevant to the lifetimes of these species? If not, there could be significant advective dilution.

Line 460 and Figure 10. Consider that the difference between the simple model and the adjusted model is about 25%. Is the uncertainty in the measured ratio smaller than that? If not then this would not be a useful exercise.

Line 472 – I am not sure that your analytical system materials are a good proxy for vegetation or urban materials like pavement. And, at night, is the dominant deposition resistance the aerodynamic resistance? If so, we would expect more or less identical deposition rates for these isomers.

Line 504 – doesn't this imply that the glyoxal chemistry is very well known? Are there aromatic hydrocarbons present? Other glyoxal precursors? What is your confidence in the model production chemistry for glyoxal?

Line 552 – delete "the" before "using".

Line 573 – is there a statistically meaningful diel pattern for the observed ratio? It doesn't look like it to me.

Line 638 – it would be good to recognize that in chemically reactive environments, NO3 chemistry can be equally important in the daytime, if the NO3 production rate is greater in the daytime.

Line 647 – please recognize that the dilution term depends on the concentration of the species in the diluent air.

Figure 19 – this makes it clear that given the broad diel cycle, a lot of propanone nitrate arises from transport, and so likely can't be simulated well.

Line 703 – is it really mostly nighttime and unimportant? What do you know about the propene concentrations and their diel cycle?

Line 733 – why do you believe it to be anthropogenic? I think Section 6.6 could be dropped.

Conclusions – this section is entirely a summary. Instead of restating what is in the paper, can you draw conclusions about what we don't know that we should work on? What are the areas that warrant further study (your important last line)?

---

## Author Response (AR1)

**Response to Reviewers' Comments to Manuscript acp-2019-964 "Observations of speciated isoprene nitrates in Beijing: implications for isoprene chemistry" by Reeves et al.**

Reviewers' comments are in black upright font.

Our response is in blue italic font.

**Referee #1**

This data set is likely interesting.

*We believe the data are very interesting and our view is supported by reviewer #3 who says "This makes this a highly unique and useful data set for chemically coupled species, and does indeed represent a great opportunity for testing the mechanism for isoprene photooxidation, and studying the impact of isoprene chemistry on the fate of NOx, and for production of ozone and particulate matter.".*

However, this paper is long, data rich and is not succinct in its analysis. It is very hard to tell which conclusions are unambiguously supported by the observations and which depend on assumptions about transmission and sensitivity.

*On reflection, we agree that the paper is too long and not succinct in analysis and more clarity is required regarding which conclusions are unambiguously supported by the observations.*

*We have created a much-shortened revised version, in part by removing the simple model analysis completely and section 6.6. We have added uncertainties, and we have rewritten the abstract and conclusions to highlight the key findings.*

It is not currently accessible to a general reader of ACP. I recommend it be rejected. Only the most determined reader will be able to wade through this and find the important information and three years from now, no one will be able to identify key ideas that should stand the test of time from ideas that are momentary arguments about different rates constants in a version of MCM and W2018. Today, no one not deeply steeped in the isoprene chemistry will be able to read it and recognize the ideas being tested.

*There are many papers published on isoprene chemistry, demonstrating widespread interest in the subject. Many of these papers are themselves very detailed including several published in ACP. This paper identifies areas of uncertainty in mechanisms that can then be addressed through further research. Publishing these results are an important way to advance science.*

*In shortening the revised version, we have also aimed to make it more accessible to the general ACP reader.*

It would greatly benefit from editing in collaboration with someone who is not as engaged in the details. I recommend it be rewritten with many fewer figures. The figures that remain should be chosen to demonstrate how the observations test competing ideas for the behavior of these nitrates.

*In shortening the revised version, we have reduced the number of figures from 21 to 9. We have done this combining some figures, reducing the number of things plotted, removing some plots altogether and moving others to the Supplementary Information. We believe key scientific points are now more clearly illustrated.*

In addition, the sections on MCM should be more clearly motivated–are there choices MCM has made that are in conflict with W2018. If so is there a logic to them or is MCM just not updated to be consistent with W2018 yet?

*The MCM is a widely used chemical mechanism. There is a logic to the choices made, which for isoprene are primarily described in Jenkin et al (2015). Wennberg et al (2018) does consider some more recent findings, but both mechanisms are based on many assumptions, often with few constraining observations. We, therefore, believe it is important to test both against new observations.*

**Referee #3**

The paper by Reeves et al. describes measurements of speciated organic nitrates that are produced from both OH and $NO_3$ reaction with isoprene. Using a GC/MS approach, they were able to identify and quantify seven different "isoprene nitrates", specifically, two ïA̤c´-hydroxy nitrates, four ïA̤d'-carbonyl nitrates, and propanone nitrate, in Beijing during the winter of 2016 and summer of 2017. Isomers were generally (not always) identified by injections of samples of the individual synthesized isomers, and quantified with reasonable time resolution (it appears to be hourly, but that is not stated clearly in the manuscript; that should be clarified). What resulted was a highly unique data set for these compounds, in an isoprene-impacted urban environment, with very good supporting chemical measurements, including isoprene, NOx, HOx, $RO_2$, $NO_3$, HONO, and HCHO. Many of these measurements are highly challenging. This makes this a highly unique and useful data set for chemically coupled species, and does indeed represent a great opportunity for testing the mechanism for isoprene photooxidation, and studying the impact of isoprene chemistry on the fate of NOx, and for production of ozone and particulate matter. This paper then should be published, and will be high impact, I believe, ……

*We appreciate the reviewer recognising the importance of this data set and its value in testing the isoprene photooxidation mechanisms and potential for high impact.*

*The measurements were made approximately hourly. We have clarified this in the revised manuscript.*

…… once one major flaw in the paper is repaired. Specifically, while the data are compared to simulations using MCM chemistry, for both absolute concentrations and ratios of coupled species, these comparisons are extremely difficult to interpret because there is no uncertainty analysis done for these seven compounds. And that lack of detailed uncertainty analysis is a problem in this case because of all the assumptions made, e.g. that sensitivities are the same for the 4,3-IN and the 1,2-IN, and because of issues related to losses of the compounds, e.g. on valves and other surfaces, that clearly have an impact, and these impacts can be different for different isomers, as the authors recognize. So, while they discuss that looking at ratios of isomer concentrations can remove the complexities of boundary layer dynamics, dilution, and ventilation, there is no discussion of the uncertainties of the ratios presented in the various analysis, discussed at length for figures 5, 6, 10, 13, 14, and 15. So, it is possible that the analyses of the these ratios and comparisons to the models are meaningful, but also possible that they contain systematic errors that make the comparison problematic. With no error bars on any of the data, it is impossible to know if the discussions and conclusions are meaningful. Given the likely very large (impressive!) effort in acquiring these data, this is an unfortunate oversight, and needs to be repaired before this paper is published. I recommend a section that does a detailed error analysis for measurements of each isomer, and presents a calculated uncertainty (which could be concentration-dependent) for each one, and also calculates the uncertainty for the ratios that are compared to MCM. The figures could include representative error bars, either on some points, or use shading to reflect the uncertainties, or some other approach. With this added information, this can be a great paper. I note that the last sentence in the paper says "Our interpretation is limited by the uncertainties in our measurements and relatively small data set, but highlights areas of the isoprene chemistry that warrant further study, in particular the $NO_3$ initiated isoprene degradation chemistry." This is good to recognize, but the reader has no idea what are the uncertainties in the measurements.

*We accept these criticisms.*

*In the revised manuscript we have included a detailed uncertainty analysis (section 3.3), providing uncertainties for both concentrations and ratios, and included errors bars in the figures. We have modified the discussions and conclusions of the comparison with the model to reflect these uncertainties.*

Other comments and relatively minor issues are listed below, in the order they arose in the paper. Comments/issues, in order

Abstract – line 32 could say isoprene-derived organic nitrates (the first time)?

*Added*

Line 43 – The observed relationship. . .

*Corrected.*

Line 53 – should say "from" the observed.

*Corrected.*

Line 92 – This key issue should be explained mechanistically, e.g. showing an example of an alkoxy radical that can decompose, releasing $NO_2$.

*The Wennberg et al (2018) paper is cited and more information is given already in the Supplementary Information (section S1.3), so in the interest in shortening the paper, we decided not to add further explanation here.*

Line 167 – sentence needs a period.

*Corrected.*

Section 3.2 – what do you know about the desorption efficiency from the Tenax trap? Since INs are olefinic, and there is lots of $O_3$, what do you know about ozonolysis during sampling?

*The reviewer is correct to point out that olefinic compounds can be affected by trapping with oxidants, however in our instrument paper (Mills et al 2016, Atmos. Meas.Tech., 9, 4533-4545, doi: 10.5194/amt-9-4533-2016, 2016.) we have demonstrated that our trapping methods are unaffected by ozone or $NO_2$.*

Is the metal valve the only surface on which INs can be (differentially) lost? How do (will) all these things affect your calculated analytical uncertainties? When you knew you had some loss on the valve, did you apply any correction for this? If not, do you have asymmetric error bars?

*Regarding differential losses, the inlet and column are not substantially different from the analytical columns and conditions used by CalTech (e.g. Vasquez et al, Atmos. Meas. Tech., doi: 10.5194/amt-11-6815-2018, 2018) so any differential losses in these parts of our system are likely to be similar and very small, consistent with the Caltech group not reporting any such losses. Only the trap and metal valve are significantly different. The metal valve clearly had significant differential losses and we have stated in our experimental section that we have indeed applied corrections for these losses. We have included the uncertainties for these corrections in our uncertainty analysis.*

*It is possible that the glass sample trap may cause differential losses. For the IHNs we measured in Mills et al (2016), these are accounted for in the overall sensitivity from calibrations of single isomer samples, however we could not do this for the ICN. We used two different sample traps and fittings towards the end of the campaign (with the plastic valve in place), and did not notice any obvious changes in the nature of the data, but this was in a period when we were doing calibrations etc and so there was a period of many hours between the air samples on the two different traps. As far as desorption from the Tenax trap, that is also covered in the Mills et al (2016). Whilst we do not know the exact desorption efficiency or losses, they must be consistent and vary little as the instrument linearity and precision demonstrated in that paper are good and there is no observable carry-over to a subsequent blank.*

Lines 204 – 208 – how do these assumptions impact your calculated uncertainties?

*The assumptions the reviewer refers to here are regarding ion counts for IN that we were unable to directly calibrate for. We have included these in our measurement uncertainty analysis (section 3.3).*

Line 278 – what exactly is the "large uncertainty"? Without these estimates, comparing to model results is an empty exercise.

*We have now provided an uncertainty analysis and adjusted the text of this section accordingly.*

Line 295 – should be "of" the summer campaign.

*Corrected.*

Line 322 – since you mention the "appreciable concentrations of OH at night", and there is a lot of interest in that subject, can you include some representative error bars in Figure 7? The same goes for $NO_3$; I would like to repeat that there is some really lovely data in this paper, but it would help the reader to know things like LODs and uncertainties.

*We have added information on the uncertainties of the supporting data in the Supplementary Information. Measurement uncertainties for OH have been added as error bars to Fig. S2 (old Fig. 3) and shaded areas representing ±1 s.d. in the variability of values for each hour of the day have been added to Fig. 4 (old Fig. 7).*

Line 357 – the ratio E-1,4 to E-4,1 is not in Figure 6.

*Yes, this was an error in the text. Corrected.*

Line 365 – yes, but we don't know what the uncertainties are!

*Addressed in the revised manuscript with the addition of the uncertainty analysis.*

Line 400 – I'll just note that alpha is not known to even two significant figures.

*Whilst we agree with the reviewers comment this value is taken from MCM which is given to 3 significant figures. However, we have removed the simple model analysis, so this has been deleted anyway.*

Line 415 – Is it known that the -OH group has no impact? What is the uncertainty here?

*We use the photolysis rates in the MCM. Without measurements of the photolysis rates of some of the larger VOCs, the MCM uses measured rates for some of the smaller VOCS to represent those of the larger VOCs following the protocols set out in Jenkin et al (1997) and Saunders et al (2003). Jenkin et al (2015) updated the degradation scheme for isoprene, and although the photolysis rates of the higher generation nitrates with carbonyl groups were revised on the basis of work by Muller et al (2014; 2015), no changes were made to the photolysis rates of the hydroxy nitrates. Whilst we accept that the -OH group may have some impact, we believe that the MCM represents the state-of-the-art in terms of scientific understanding and so it is appropriate to use these rates.*

Line 417 – how does $4 \times 10^{-5}$ s-1 compare to the magnitude of the calculated chemical reaction loss? (since you assume here that all the loss is uptake)

*We have removed the simple model analysis and focussed the paper on the MCM model. This has therefore been removed.*

Line 443 – Is the upwind environment chemically comparable on a timescale relevant to the lifetimes of these species? If not, there could be significant advective dilution.

*We have removed the simple model analysis and focussed the paper on the MCM model. This has therefore been removed.*

Line 460 and Figure 10. Consider that the difference between the simple model and the adjusted model is about 25%. Is the uncertainty in the measured ratio smaller than that? If not then this would not be a useful exercise.

*We have removed the simple model analysis and focussed the paper on the MCM model. This has therefore been removed.*

Line 472 – I am not sure that your analytical system materials are a good proxy for vegetation or urban materials like pavement. And, at night, is the dominant deposition resistance the aerodynamic resistance? If so, we would expect more or less identical deposition rates for these isomers.

*Our analytical system materials may not be a good proxy for vegetation or urban materials, but the evidence that exists (i.e. difficulty of getting (1-OH, 2-ONO$_2$)-IHN through an analytical system and its fast rate of hydrolysis (W2018)) suggest that, if anything, (1-OH, 2-ONO$_2$)-IHN is more likely to have a faster deposition than that of (4-OH, 3-ONO$_2$)-IHN.*

Line 504 – doesn't this imply that the glyoxal chemistry is very well known? Are there aromatic hydrocarbons present? Other glyoxal precursors? What is your confidence in the model production chemistry for glyoxal?

*A range of aromatic species were measured (including benzene, toluene, ethyl benzene, xylenes and tri-methyl benzenes) and used to constrain the model as well as acetylene which is another important glyoxal precursor. As a further check on the physical loss rate imposed, however, the model was run unconstrained to HCHO using the same deposition rates and was found to reproduce the observed HCHO concentrations that were observed during the daytime, but under-predicted the concentrations at night.*

Line 552 – delete "the" before "using".

*Corrected.*

Line 573 – is there a statistically meaningful diel pattern for the observed ratio? It doesn't look like it to me.

*No, there is not. The text here is referring to the modelled rather than observed ratio.*

*We have clarified this in the text.*

Line 638 – it would be good to recognize that in chemically reactive environments, NO$_3$ chemistry can be equally important in the daytime, if the NO$_3$ production rate is greater in the daytime.

*We do not fully understand the point being made by the reviewer. Equally important to what? Night-time chemistry? OH chemistry? We do say "the production of δ-ICN in the model is mostly during the daytime, despite NO$_3$ usually being considered to be more important at night.". We think this is a clear message as to the importance of NO$_3$ chemistry during the daytime, based on looking at the modelled source of the δ-ICN.*

Line 647 – please recognize that the dilution term depends on the concentration of the species in the diluent air.

*Yes, this is an important point and we have added this to the discussion here.*

Figure 19 – this makes it clear that given the broad diel cycle, a lot of propanone nitrat arises from transport, and so likely can't be simulated well.

*Whilst transport may play a part, the broad diel cycle may also be due to there being both daytime and night-time sources. The chemical lifetime of propanone nitrate does mean that transport is important making it difficult to simulate the observations with a box model, but it is still useful to gain an insight into the dominant chemical production and loss processes.*

Line 703 – is it really mostly nighttime and unimportant? What do you know about the propene concentrations and their diel cycle?

*Yes, the source of propanone nitrate produced following the $NO_3$ addition to propene acts predominantly at night-time. Overall, the model results suggest this to be a relatively small source, however, looking at the fluxes again we can see that at night it is often calculated to be the dominant source, and so some of the night-time peaks in propanone nitrate may not come from isoprene.*

*The text has been changed to reflect this.*

Line 733 – why do you believe it to be anthropogenic? I think Section 6.6 could be dropped.

*This section has been dropped.*

Conclusions – this section is entirely a summary. Instead of restating what is in the paper, can you draw conclusions about what we don't know that we should work on? What are the areas that warrant further study (your important last line)?

*The Conclusions sections has been rewritten highlighting what we do not know and areas for further study as suggested.*

[revised manuscript text omitted]
. (2011), with an uncertainty of around 5% depending on the mixing ratio calculated following procedures set out in the ACTRIS Measurement Guidelines (Reimann et al, 2018). Measurements of OH, HO$_2$ and RO$_2$ were obtained using the fluorescence assay by gas expansion (FAGE) technique equipped with a scavenger inlet for OH, with the OH chem method used to obtain the background OH signal (Whalley et al., 2010; Whalley et al., 2018; Woodward-Massey et al., 2019). The median limit of detection (LOD) during the campaign was $6.1 \times 10^5$ molecule cm$^{-3}$ for OH, $2.8 \times 10^6$ molecule cm$^{-3}$ for HO$_2$ and $7.2 \times 10^6$ molecule cm$^{-3}$ for CH$_3$O$_2$ at a typical laser power of 11 mW for a 5 minute data acquisition cycle (SNR=2). The accuracy of the measurements was ~ 26 % (2 σ) and is derived from the error in the calibration.

A Thermo Environmental Instruments (TEI) 49i UV absorption analyser was used to measure O$_3$ (uncertainty 4.04%, precision 0.28 ppb). NO was measured using a TEI 42i (uncertainty 4.58%, precision 0.03 ppb), NO$_2$ by a Teledyne cavity attenuated phase shift (CAPS) instrument (uncertainty 5.73%, precision 0.04 ppb) and CO by a sensor box (uncertainty 9.14%, precision 2.14 ppb) (Smith et al., 2017). The precisions quoted above are 2 σ precisions calculated from the standard deviation of the zero calibration and then divided by square root of the number of measurements during a 15-minute averaging time. The O$_3$ uncertainty is derived from the uncertainty of its reaction with NO in the sample line. The uncertainties in the NO measurements are calculated as the sum of the uncertainty in calibration cylinder, the standard deviation of the calibration and the uncertainty due to reaction of O$_3$ with NO in the sample line. Both the NO and NO$_2$ calibration cylinders are traceable to the National Physics Laboratory NO scale. The CO uncertainty is the sum of the uncertainty in calibration cylinder and the standard deviation of the calibration.

NO$_3$ and glyoxal were measured using broadband cavity enhanced absorption spectroscopy (Kennedy et al., 2011). The measurement accuracy of NO$_3$ is around 1 ppt at 5 seconds sampling rate, which can be reduced by averaging, such that during the early afternoon the hourly mean uncertainties are greatest at around 0.5 ppt, i.e. 20% of the mean mixing ratio. HONO was measured by a long path absorption photometer (LOPAP) (Crilley et al., 2016). A proton transfer reaction-time of flight-mass spectrometer (PTR-ToF-MS) was used to measure multi-functional aromatics and monoterpenes, whilst HCHO was measured by LIF (Cryer, 2016). SO$_2$ was measured by a TEI 43i instrument. The

[revised manuscript text omitted]

**S4. Figures**

[Figure]

**Figure S1: Isoprene nitrates mixing ratios measured in Beijing. Error bars are the measurement uncertainties (see Sect. 3.2 for details).**

[Figure]

**Figure S2: Isoprene, CO, NO, NO$_2$ and O$_3$ mixing ratios measured in Beijing for the times corresponding to the IN data shown in Fig. S1. Error bars are the measurement uncertainties (see Sect. S2 for details).**

[Figure]

**Figure S3: Time series of β-IHN as modelled using the MCM and measured. Error bars are the measurement uncertainties (see Sect. 3.3 for details).**

[Figure]

**Figure S4: (a) Time series of total δ-ICN as modelled using the MCM and measured. For the MCM this is the species NC4CHO, whilst the measurements are the sum of the four δ-ICN (E and Z-(1-ONO₂, 4-CO)-ICN and E and Z-(4-ONO₂, 1-CO)-ICN). (b) Time series of propanone nitrate as modelled using the MCM and measured. Error bars are the measurement uncertainties (see Sect. 3.3 for details).**

[Figure]

**Figure S5: Time series the MCM modelled δ-IHN ((1-OH, 4-ONO₂)-IHN and (4-OH, 1-ONO₂)-IHN) and δ-ICN (NC4CHO).**

**Table S1: Data used for plotting Figs. 2, 3, S1, S3 and S4.**

| Date Time (DD/MM/YYYY hh:mm) | (4-OH, 3-ONO$_2$)-IHN (ppt) | (1-OH, 2-ONO$_2$)-IHN (ppt) | E-(4-ONO$_2$, 1-CO)-ICN (ppt) | Z-(1-ONO$_2$, 4-CO)-ICN (ppt) | E-(1-ONO$_2$, 4-CO)-ICN (ppt) | Z-(4-ONO$_2$, 1-CO)-ICN (ppt) | Propanone nitrate (ppt) |
|---|---|---|---|---|---|---|---|
| 17/05/2017 17:40 | 5.19 | | | | | | |
| 17/05/2017 18:40 | 5.56 | | | | | 2.95 | 8.03 |
| 17/05/2017 19:40 | 8.33 | | 3.27 | 0.73 | 11.65 | 1.46 | 14.42 |
| 17/05/2017 20:40 | 11.01 | | 9.37 | 1.36 | 14.69 | 2.24 | 16.69 |
| 17/05/2017 21:40 | 9.66 | | 9.95 | 1.58 | 16.91 | 2.36 | 22.21 |
| 17/05/2017 22:40 | 10.06 | | 14.27 | 1.17 | 15.50 | 2.57 | 54.42 |
| 17/05/2017 23:40 | 10.67 | | 22.08 | 2.18 | 27.72 | 2.45 | 73.78 |
| 18/05/2017 00:40 | 10.53 | | 26.68 | 2.46 | 33.86 | | 92.10 |
| 18/05/2017 01:40 | 9.30 | | 23.97 | 2.98 | 49.39 | 1.78 | 104.52 |
| 18/05/2017 02:40 | 6.94 | | 17.57 | 2.14 | 28.93 | | 98.76 |
| 18/05/2017 03:40 | 5.35 | | 14.24 | 1.45 | 8.45 | 1.18 | 63.62 |
| 18/05/2017 04:40 | 3.70 | | 11.38 | 0.95 | 17.16 | | 60.87 |
| 18/05/2017 05:40 | 2.72 | | 10.20 | 1.27 | 13.19 | | 65.45 |
| 18/05/2017 06:40 | 2.38 | | 7.68 | 1.26 | 6.76 | 1.89 | 67.56 |
| 18/05/2017 07:40 | 3.99 | | 5.50 | 1.28 | | 1.60 | 74.75 |
| 18/05/2017 08:40 | 13.04 | | 4.17 | 0.81 | 4.83 | 1.23 | 99.95 |
| 18/05/2017 10:44 | 25.69 | | 2.92 | 0.59 | 5.02 | 2.90 | 114.00 |
| 18/05/2017 11:44 | 16.09 | | 1.99 | 0.44 | | 3.23 | 63.81 |
| 18/05/2017 12:44 | 12.20 | | 1.85 | 0.35 | | 2.41 | 52.86 |
| 18/05/2017 13:44 | | | 0.45 | | | | 3.66 |
| 18/05/2017 14:44 | | | 8.85 | 0.24 | | | |
| 18/05/2017 15:44 | | | 0.78 | | | | |
| 18/05/2017 16:44 | 7.16 | | 0.86 | | 6.08 | | 25.31 |
| 18/05/2017 17:44 | 7.95 | | 1.23 | | 4.98 | 1.36 | 21.94 |
| 18/05/2017 18:44 | 6.09 | | 1.49 | | | 1.71 | 17.02 |
| 18/05/2017 19:44 | 8.35 | | 5.71 | 0.96 | 19.17 | 1.44 | 26.92 |
| 18/05/2017 20:44 | 6.44 | | 12.44 | 1.48 | 25.31 | 1.23 | 47.10 |
| 18/05/2017 21:44 | 3.52 | | 8.71 | 0.95 | 14.44 | 0.99 | 36.44 |
| 18/05/2017 22:44 | 3.67 | | 9.88 | 0.88 | 15.46 | | 40.04 |
| 18/05/2017 23:44 | 3.20 | | 10.19 | 1.32 | 24.77 | | 46.70 |
| 19/05/2017 00:44 | 3.35 | | 8.85 | 1.33 | 14.87 | 0.56 | 52.27 |
| 19/05/2017 01:44 | 3.47 | | 7.35 | 0.83 | 11.11 | | 49.96 |
| 19/05/2017 02:44 | 3.28 | | 7.90 | 0.88 | 16.14 | | 58.45 |
| 19/05/2017 03:44 | 3.50 | | 7.65 | 0.77 | 10.19 | 1.23 | 61.74 |
| 19/05/2017 04:44 | 3.35 | | 7.63 | 0.80 | 15.51 | 0.88 | 63.81 |
| 19/05/2017 05:44 | 3.12 | | 9.24 | 0.95 | 10.92 | 0.98 | 77.75 |
| 19/05/2017 06:44 | 3.96 | | 12.39 | 1.41 | 17.49 | 1.28 | 86.20 |
| 19/05/2017 07:44 | 6.12 | | 8.73 | 1.38 | 9.52 | 1.32 | 89.54 |

| | | | | | | |
|---|---|---|---|---|---|---|
| 19/05/2017 08:44 | 8.32 | | 6.33 | 1.00 | | 1.31 | 109.90 |
| 19/05/2017 11:23 | 17.43 | | | | | | 156.64 |
| 19/05/2017 12:37 | 12.60 | | | | | | 102.27 |
| 19/05/2017 13:47 | 9.34 | | | | | | 89.48 |
| 19/05/2017 14:54 | 9.37 | | 2.79 | 0.22 | 2.42 | 1.85 | 98.05 |
| 19/05/2017 15:54 | 10.97 | | 2.24 | 0.31 | | 1.99 | 48.87 |
| 19/05/2017 16:54 | 13.61 | | 3.11 | 0.24 | | 2.23 | 50.39 |
| 19/05/2017 17:54 | 12.02 | | 2.80 | 0.36 | | 1.35 | 40.58 |
| 19/05/2017 18:54 | 10.19 | | 4.54 | 0.40 | 9.13 | 1.38 | 26.59 |
| 19/05/2017 19:55 | 9.11 | | 10.40 | 0.94 | 14.93 | | 31.61 |
| 19/05/2017 20:55 | 6.65 | | 12.49 | 1.07 | 26.23 | | 26.95 |
| 19/05/2017 21:55 | 5.74 | | 11.75 | 0.95 | 28.35 | | 37.24 |
| 19/05/2017 22:55 | 4.03 | | 9.81 | 0.58 | 13.28 | | 41.60 |
| 19/05/2017 23:55 | 2.77 | | 6.18 | 0.62 | 15.03 | | 47.19 |
| 20/05/2017 00:55 | 2.16 | | 5.60 | 0.92 | 14.30 | | 39.88 |
| 20/05/2017 01:55 | 2.27 | | 5.61 | 0.66 | 14.15 | | 43.02 |
| 20/05/2017 02:55 | 2.34 | | 6.11 | 0.58 | 9.71 | | 55.37 |
| 20/05/2017 03:55 | 2.58 | | 6.90 | 0.81 | 7.39 | | 57.33 |
| 20/05/2017 04:55 | 2.79 | | 7.43 | 0.52 | 9.66 | 0.48 | 56.61 |
| 20/05/2017 05:55 | 2.83 | | 6.90 | 0.72 | 7.25 | 0.86 | 41.98 |
| 20/05/2017 06:55 | 4.12 | | 5.48 | 0.68 | 4.15 | | 42.67 |
| 20/05/2017 07:55 | 7.12 | | 4.79 | 0.47 | 9.03 | 0.73 | 48.21 |
| 20/05/2017 08:55 | 6.66 | | 2.85 | 0.49 | 3.38 | 0.79 | 58.02 |
| 20/05/2017 09:55 | 16.12 | | 2.66 | 0.51 | 4.15 | 0.81 | 76.71 |
| 20/05/2017 10:55 | 23.07 | | 2.77 | 0.42 | | 0.98 | 83.70 |
| 20/05/2017 11:54 | 25.60 | | 2.32 | 0.25 | | 1.14 | 83.42 |
| 20/05/2017 12:54 | 8.91 | | 1.76 | | 4.54 | 1.18 | 48.23 |
| 20/05/2017 13:54 | 6.70 | | 3.86 | | | | 28.17 |
| 20/05/2017 14:54 | 5.88 | | 3.52 | 0.12 | | 0.98 | 33.63 |
| 20/05/2017 15:54 | 6.48 | | 2.88 | | | | 31.61 |
| 20/05/2017 16:54 | 4.83 | | 4.64 | | | 1.23 | 35.20 |
| 20/05/2017 17:54 | 5.07 | | 4.39 | | | 0.91 | 35.71 |
| 20/05/2017 18:54 | 6.22 | | 6.28 | 0.62 | 11.98 | | 38.99 |
| 20/05/2017 19:54 | 3.53 | | 5.92 | 0.43 | 14.00 | | 34.29 |
| 20/05/2017 20:54 | 4.47 | | 11.00 | 0.81 | 19.46 | 0.87 | 51.66 |
| 20/05/2017 21:54 | 2.46 | | 6.87 | 0.77 | 11.01 | | 36.64 |
| 20/05/2017 22:54 | 2.75 | | 11.34 | 0.93 | 14.58 | | 83.63 |
| 20/05/2017 23:54 | 0.93 | | 7.33 | 0.65 | 9.90 | 0.49 | 59.36 |
| 21/05/2017 00:54 | 0.49 | | 6.39 | 0.42 | 6.57 | 0.74 | 60.23 |
| 21/05/2017 01:54 | 0.45 | | 4.41 | 0.23 | 5.17 | | 45.76 |
| 21/05/2017 02:54 | 0.36 | | 3.68 | 0.29 | | | 25.12 |
| 21/05/2017 03:54 | 0.26 | | 3.09 | 0.23 | | | 22.29 |
| 21/05/2017 04:54 | 0.25 | | 3.53 | | | | 17.48 |
| 21/05/2017 05:54 | 0.32 | | 2.70 | 0.24 | | | 19.34 |
| 21/05/2017 06:54 | 0.47 | | 2.76 | | | | 17.66 |

| | | | | | | | |
|---|---|---|---|---|---|---|---|
| 21/05/2017 07:54 | 1.05 | | 2.17 | 0.13 | | | 23.35 |
| 21/05/2017 08:54 | 2.78 | | 2.44 | | | | 28.53 |
| 21/05/2017 09:54 | 5.62 | | 2.23 | | | | 41.00 |
| 21/05/2017 10:59 | 8.98 | | 3.34 | | | 0.54 | 64.62 |
| 21/05/2017 11:59 | 8.57 | | 2.24 | | | 1.21 | 46.71 |
| 21/05/2017 12:59 | 7.15 | | 3.04 | | | 0.96 | 36.48 |
| 21/05/2017 13:59 | 6.24 | | 2.34 | | | 1.20 | 33.05 |
| 21/05/2017 14:59 | 5.60 | | 1.35 | | | 0.62 | 18.84 |
| 21/05/2017 15:56 | | | | | | | |
| 21/05/2017 16:56 | | | | | | | |
| 21/05/2017 18:05 | | | | 0.37 | | 1.35 | 58.72 |
| 21/05/2017 19:05 | 4.30 | | 1.92 | | | | 21.31 |
| 21/05/2017 20:05 | 3.06 | | 3.50 | 0.08 | 5.61 | | 18.12 |
| 21/05/2017 21:05 | 1.55 | | 3.52 | 0.24 | 8.45 | | 16.79 |
| 21/05/2017 22:05 | 1.66 | | 3.92 | 0.32 | 8.71 | | 19.83 |
| 21/05/2017 23:05 | 0.87 | | 1.81 | 0.26 | 4.96 | | 15.44 |
| 22/05/2017 00:05 | 0.51 | | 2.43 | | 4.91 | 0.70 | 11.82 |
| 22/05/2017 01:05 | 0.55 | | 2.50 | | | | 12.15 |
| 22/05/2017 02:05 | 0.39 | | 2.21 | 0.16 | 4.58 | | 9.80 |
| 22/05/2017 03:05 | 0.27 | | 2.20 | | 2.51 | | 8.73 |
| 22/05/2017 04:05 | | | | | | | |
| 22/05/2017 05:05 | | | | | | | |
| 22/05/2017 06:05 | | | | | | | |
| 22/05/2017 07:05 | | | | 0.02 | 0.39 | | |
| 22/05/2017 08:05 | | | | 0.02 | 0.59 | | |
| 22/05/2017 09:05 | | | | 0.02 | 0.87 | | |
| 22/05/2017 10:05 | | | | | | | |
| 22/05/2017 10:46 | 0.57 | | | | | | 14.70 |
| 22/05/2017 11:46 | 0.13 | | 1.14 | | | | 5.27 |
| 22/05/2017 12:46 | 0.14 | | 1.46 | | | | 3.67 |
| 22/05/2017 13:46 | 0.28 | | 1.58 | | | | 4.16 |
| 22/05/2017 14:46 | 0.30 | | 1.78 | | | | 3.81 |
| 22/05/2017 15:44 | 0.47 | | 1.55 | | | | 5.43 |
| 22/05/2017 16:44 | 0.66 | | | | | | 4.85 |
| 22/05/2017 17:44 | 0.76 | | | | | | 4.63 |
| 22/05/2017 18:44 | 0.79 | | | | | | 4.29 |
| 22/05/2017 19:44 | 1.00 | | | | | | 4.54 |
| 22/05/2017 20:45 | 0.78 | | | | | | 3.94 |
| 22/05/2017 21:45 | 1.03 | | | | | | 2.87 |
| 22/05/2017 22:45 | 1.09 | | | | | | 4.10 |
| 22/05/2017 23:45 | 1.02 | | | | | | 4.58 |
| 23/05/2017 00:45 | 0.81 | | | | | | 2.61 |
| 23/05/2017 01:45 | 0.70 | | 0.30 | | | | 3.45 |
| 23/05/2017 02:45 | 0.69 | | | | | | 3.58 |
| 23/05/2017 03:45 | 0.54 | | 0.48 | | | | 2.18 |

| | | | | | | | |
|---|---|---|---|---|---|---|---|
| 23/05/2017 04:45 | 0.47 | | | | | | 2.67 |
| 23/05/2017 05:45 | 0.46 | | | | | | 2.38 |
| 23/05/2017 06:45 | 0.82 | | | | | | 4.31 |
| 23/05/2017 07:45 | 1.08 | | | | | | 5.72 |
| 23/05/2017 08:45 | 1.11 | | | | | | 5.10 |
| 23/05/2017 10:05 | | | | | | | |
| 23/05/2017 10:51 | 1.88 | | 0.92 | | 1.90 | | 7.47 |
| 23/05/2017 11:51 | 2.90 | | 1.12 | | 2.97 | | 7.13 |
| 23/05/2017 12:51 | 2.48 | | 1.46 | | | | 6.58 |
| 23/05/2017 13:51 | 1.96 | | 1.53 | | | | 6.75 |
| 23/05/2017 14:51 | 1.94 | | 1.53 | | | | 5.83 |
| 23/05/2017 15:51 | 1.26 | | 1.34 | | | | 4.29 |
| 23/05/2017 16:51 | 1.24 | | 0.99 | | | | 5.44 |
| 23/05/2017 17:51 | 3.36 | | 1.95 | | | | 7.45 |
| 23/05/2017 18:51 | 4.40 | | 2.11 | | | 0.68 | 9.15 |
| 23/05/2017 19:51 | 3.14 | | 2.58 | 0.11 | 3.60 | | 9.06 |
| 23/05/2017 20:51 | 3.94 | | 4.14 | 0.35 | 5.03 | | 11.19 |
| 23/05/2017 21:51 | 3.48 | | 6.24 | 0.32 | 6.62 | | 13.39 |
| 23/05/2017 22:51 | 3.04 | | 11.15 | 0.34 | 5.25 | | 13.74 |
| 23/05/2017 23:51 | 2.97 | | 7.80 | 0.31 | 4.72 | 0.27 | 16.35 |
| 24/05/2017 00:51 | 2.38 | | 5.49 | 0.30 | 3.15 | | 16.76 |
| 24/05/2017 01:51 | 1.96 | | 4.01 | 0.35 | 1.54 | 0.39 | 12.82 |
| 24/05/2017 02:51 | 2.07 | | 3.58 | 0.19 | 2.99 | | 13.33 |
| 24/05/2017 03:51 | 1.70 | | 2.97 | 0.19 | 2.46 | | 10.30 |
| 24/05/2017 04:51 | 1.89 | | 2.61 | 0.20 | | | 14.49 |
| 24/05/2017 05:51 | 1.79 | | 2.70 | 0.26 | 1.48 | 0.24 | 37.48 |
| 24/05/2017 06:51 | 3.20 | | 2.00 | 0.26 | 2.47 | | 10.95 |
| 24/05/2017 07:51 | 3.05 | | 0.84 | | | 0.32 | 8.69 |
| 24/05/2017 08:51 | 8.87 | | 0.68 | | | 0.47 | 5.96 |
| 24/05/2017 09:51 | 9.05 | | 0.71 | | 2.17 | | 4.80 |
| 24/05/2017 10:51 | 7.07 | | 1.15 | | | | 3.06 |
| 24/05/2017 11:51 | 4.69 | | 0.59 | | 2.62 | | 4.58 |
| 24/05/2017 12:51 | 8.20 | | 0.97 | | | 0.53 | 6.19 |
| 24/05/2017 13:51 | 4.24 | | 1.15 | | | 0.56 | 7.30 |
| 24/05/2017 14:51 | 5.83 | | 1.51 | | 2.20 | 0.63 | 10.19 |
| 24/05/2017 15:51 | 4.59 | | 0.89 | | | | 9.15 |
| 24/05/2017 16:51 | 7.00 | | 1.42 | | | 0.69 | 10.49 |
| 24/05/2017 17:51 | 5.22 | | 1.54 | | | | 9.26 |
| 24/05/2017 18:51 | 4.71 | | 1.36 | | | | 6.60 |
| 24/05/2017 19:51 | 4.23 | | 1.93 | 0.22 | 3.27 | | 8.54 |
| 24/05/2017 20:51 | 6.93 | | 10.03 | 0.55 | 13.09 | 0.95 | 10.39 |
| 24/05/2017 21:51 | 7.24 | | 12.96 | 1.07 | 18.55 | | 15.56 |
| 24/05/2017 22:51 | 6.61 | | 12.08 | 0.87 | 13.04 | | 15.76 |
| 24/05/2017 23:51 | 7.02 | | 10.63 | 0.98 | 11.34 | | 15.70 |
| 25/05/2017 00:51 | 4.84 | | 7.61 | 0.52 | 6.18 | | 16.12 |

| Date/Time | | | | | | | |
|---|---|---|---|---|---|---|---|
| 25/05/2017 01:51 | 3.16 | | 4.06 | 0.35 | 3.45 | | 11.46 |
| 25/05/2017 02:51 | 1.35 | | 1.86 | 0.14 | 3.10 | | 5.23 |
| 25/05/2017 03:51 | 0.66 | | 1.03 | 0.08 | | | |
| 25/05/2017 04:51 | 0.48 | | 0.74 | | | | 3.10 |
| 25/05/2017 05:51 | 0.59 | | 0.62 | | | | 2.88 |
| 25/05/2017 06:51 | 0.85 | | 0.61 | | | | 2.42 |
| 25/05/2017 07:51 | 3.25 | | 0.84 | | | | 3.18 |
| 25/05/2017 08:51 | 5.42 | | 0.86 | | | | 4.81 |
| 25/05/2017 09:51 | 8.88 | | 1.24 | | | | 5.91 |
| 25/05/2017 10:51 | 17.47 | | 1.69 | 0.11 | | | 8.92 |
| 25/05/2017 11:51 | | | | | | | |
| 25/05/2017 12:51 | 6.40 | | | | | | 9.96 |
| 25/05/2017 13:50 | 7.28 | | 2.56 | | | 0.47 | 18.20 |
| 25/05/2017 14:50 | 6.21 | | 3.29 | | | 1.00 | 11.45 |
| 25/05/2017 15:50 | 4.85 | | 2.84 | | 1.85 | 1.27 | 14.62 |
| 25/05/2017 16:50 | 4.82 | | 2.52 | | | | 11.83 |
| 25/05/2017 17:50 | 4.57 | | 2.14 | | | | 10.69 |
| 25/05/2017 18:50 | 4.62 | | 1.64 | 0.11 | | | 8.72 |
| 25/05/2017 19:50 | 5.29 | | 2.16 | 0.17 | 2.52 | | 6.34 |
| 25/05/2017 20:50 | 4.25 | | 6.52 | 0.63 | 9.06 | | 7.89 |
| 25/05/2017 21:50 | 2.02 | | 4.23 | 0.43 | 7.18 | | 6.40 |
| 25/05/2017 22:50 | 1.60 | | 3.44 | 0.35 | 4.96 | | 7.23 |
| 25/05/2017 23:50 | 1.40 | | 3.55 | 0.42 | 5.41 | | 7.89 |
| 26/05/2017 00:50 | 1.16 | | 3.68 | 0.36 | 5.06 | | 7.93 |
| 26/05/2017 01:50 | 0.97 | | 3.10 | 0.48 | 8.16 | | 10.90 |
| 26/05/2017 02:50 | 0.96 | | 3.16 | 0.37 | 6.15 | | 13.96 |
| 26/05/2017 03:50 | 1.04 | | 3.20 | 0.39 | 5.49 | | 12.13 |
| 26/05/2017 04:50 | 0.88 | | 2.77 | 0.29 | | | 13.66 |
| 26/05/2017 05:50 | 0.49 | | 1.64 | 0.37 | | | 7.32 |
| 26/05/2017 06:50 | 0.63 | | 1.97 | 0.34 | | | 9.43 |
| 26/05/2017 07:50 | 1.29 | | 1.37 | 0.15 | | | 10.37 |
| 26/05/2017 08:50 | 2.57 | | 1.47 | 0.28 | | | 11.72 |
| 26/05/2017 09:50 | 4.58 | | 1.45 | 0.17 | 0.16 | | 14.65 |
| 26/05/2017 10:50 | 8.92 | | 1.58 | | | 0.33 | 18.72 |
| 26/05/2017 11:50 | 11.99 | | 2.06 | | | 0.68 | 24.63 |
| 26/05/2017 12:50 | 13.52 | | 2.27 | | | 0.71 | 30.08 |
| 26/05/2017 13:50 | 13.20 | | 2.20 | | | 0.74 | 34.49 |
| 26/05/2017 14:50 | 9.57 | | 3.17 | | | 0.71 | 36.69 |
| 26/05/2017 16:02 | 13.39 | | | | | | 107.95 |
| 26/05/2017 17:02 | 13.64 | | | | | | 65.58 |
| 26/05/2017 18:02 | 13.68 | | | | | | 73.04 |
| 26/05/2017 19:02 | 12.46 | | | | | | 64.44 |
| 26/05/2017 20:02 | 13.72 | | | | | | 89.41 |
| 26/05/2017 21:02 | | | 23.27 | 2.02 | 31.85 | 0.75 | |
| 26/05/2017 22:02 | | | 26.33 | 2.39 | 31.35 | 0.60 | |

| | | | | | | | |
|---|---|---|---|---|---|---|---|
| 26/05/2017 23:02 | | | 27.10 | 2.77 | 35.71 | 0.44 | |
| 27/05/2017 00:02 | | | 14.62 | 1.47 | 22.24 | 0.40 | |
| 27/05/2017 01:02 | | | 5.72 | 0.56 | 10.28 | 0.30 | |
| 27/05/2017 02:02 | | | 5.72 | 0.45 | 6.91 | | |
| 27/05/2017 03:02 | | | 4.12 | 0.59 | 5.18 | | |
| 27/05/2017 04:02 | | | 3.23 | 0.33 | 5.32 | 0.31 | |
| 27/05/2017 05:02 | | | 3.67 | 0.19 | 4.24 | 0.35 | |
| 27/05/2017 06:02 | | | 2.93 | 0.25 | 2.81 | | |
| 27/05/2017 07:02 | | | 2.60 | 0.20 | | 0.32 | |
| 27/05/2017 08:02 | 1.26 | | | | | | 32.98 |
| 27/05/2017 09:02 | 3.92 | | | | | | 46.78 |
| 27/05/2017 10:02 | 9.85 | | | | | | 58.31 |
| 27/05/2017 11:02 | 19.07 | | | | | | 84.93 |
| 27/05/2017 12:02 | 25.31 | | | | | | 100.82 |
| 27/05/2017 13:02 | 33.39 | | | | | | 101.07 |
| 27/05/2017 14:02 | 33.28 | | | | | | 135.89 |
| 27/05/2017 15:02 | 22.41 | | | | | | |
| 27/05/2017 16:02 | 20.76 | | | | | | |
| 27/05/2017 17:02 | 21.14 | | | | | | |
| 28/05/2017 12:28 | | | | | | | |
| 28/05/2017 13:49 | | | | | | | |
| 28/05/2017 14:45 | | | | | | | |
| 28/05/2017 15:40 | | | | | | | |
| 28/05/2017 16:22 | 0.81 | | | | | | |
| 28/05/2017 17:22 | 0.29 | | | | | | |
| 28/05/2017 18:22 | 8.50 | | | | | | |
| 28/05/2017 19:22 | 7.02 | | | | | | |
| 28/05/2017 20:22 | 4.15 | | | | | | |
| 28/05/2017 21:22 | 2.34 | | | | | | |
| 28/05/2017 22:22 | 0.62 | | | | | | |
| 28/05/2017 23:22 | 1.00 | | | | | | |
| 29/05/2017 00:22 | 0.14 | | | | | | |
| 29/05/2017 01:22 | 0.05 | | | | | | |
| 29/05/2017 02:22 | 0.10 | | | | | | |
| 29/05/2017 03:22 | 0.10 | | | | | | |
| 29/05/2017 04:22 | 0.19 | | | | | | |
| 29/05/2017 05:22 | 0.14 | | | | | | |
| 29/05/2017 06:22 | 0.19 | | | | | | |
| 29/05/2017 07:22 | 0.29 | | | | | | |
| 29/05/2017 08:22 | 3.44 | | | | | | |
| 29/05/2017 09:22 | 4.06 | | | | | | |
| 29/05/2017 10:22 | 3.94 | | | | | | |
| 29/05/2017 11:22 | 3.71 | | | | | | |
| 29/05/2017 12:22 | 0.12 | | | | | | |
| 29/05/2017 13:25 | | | | | | | |

| Date/Time | | | | | | | |
|---|---|---|---|---|---|---|---|
| 29/05/2017 14:37 | 2.06 | | | | | | 20.75 |
| 29/05/2017 16:50 | 1.12 | | 0.85 | | | | 31.72 |
| 29/05/2017 17:50 | 0.95 | | 1.20 | | | | 18.88 |
| 29/05/2017 18:50 | 0.78 | | 1.42 | | | | 16.74 |
| 29/05/2017 19:50 | 0.55 | | 1.51 | | | | 20.51 |
| 29/05/2017 20:50 | 0.50 | | 1.77 | | | | 24.01 |
| 29/05/2017 21:50 | 0.54 | | 2.46 | 0.31 | | | 26.49 |
| 29/05/2017 22:50 | 1.12 | | 2.53 | 0.15 | 5.20 | | 27.35 |
| 29/05/2017 23:50 | 0.89 | | 2.57 | 0.27 | | | 37.85 |
| 30/05/2017 00:50 | 0.93 | | 2.55 | 0.17 | 2.48 | | 40.57 |
| 30/05/2017 01:50 | 0.74 | | 2.94 | 0.27 | 2.27 | | 38.13 |
| 30/05/2017 02:50 | 1.34 | | 2.61 | 0.31 | 5.41 | | 31.61 |
| 30/05/2017 03:50 | 1.38 | | 1.94 | 0.32 | 4.38 | | 31.99 |
| 30/05/2017 04:50 | 1.51 | | 2.18 | 0.27 | | | 24.51 |
| 30/05/2017 05:50 | 1.02 | | 1.16 | | | | 22.52 |
| 30/05/2017 06:50 | 1.04 | | 1.28 | | | | 20.52 |
| 30/05/2017 07:50 | 0.86 | | 0.80 | | | | 22.18 |
| 30/05/2017 08:50 | 0.93 | | 1.18 | | | | 23.73 |
| 30/05/2017 09:50 | 1.18 | | 1.10 | | | | 28.90 |
| 30/05/2017 10:50 | 1.63 | | 0.55 | | | | 26.62 |
| 30/05/2017 11:50 | 2.77 | | 1.33 | | | | 29.33 |
| 30/05/2017 12:50 | 4.90 | | | | | | 22.59 |
| 30/05/2017 13:47 | 5.37 | | | | | | 28.25 |
| 30/05/2017 14:47 | 4.24 | | | | | | 29.23 |
| 30/05/2017 15:47 | 3.42 | | | | | | 37.52 |
| 30/05/2017 16:47 | 4.98 | | | | | | 36.37 |
| 30/05/2017 17:47 | | | | | | | |
| 30/05/2017 18:47 | 5.20 | | 1.66 | | | | 25.58 |
| 30/05/2017 19:47 | 5.06 | | 1.90 | 0.19 | 3.20 | | 26.18 |
| 30/05/2017 20:47 | 4.27 | | 5.50 | 0.57 | 20.03 | | 43.95 |
| 30/05/2017 21:47 | 3.61 | | 6.31 | 0.58 | 17.98 | | 61.32 |
| 30/05/2017 22:47 | 2.81 | | 5.59 | 0.44 | 12.29 | | 72.36 |
| 30/05/2017 23:47 | 2.29 | | 4.71 | 0.42 | 10.83 | | 69.82 |
| 31/05/2017 00:47 | 1.75 | | 4.30 | 0.44 | 12.13 | | 60.93 |
| 31/05/2017 01:47 | 1.60 | | 3.88 | 0.45 | 9.35 | | 59.20 |
| 31/05/2017 02:47 | 1.52 | | 4.04 | 0.31 | 8.51 | | 56.68 |
| 31/05/2017 03:47 | 2.41 | | 4.62 | 0.34 | 7.70 | | 66.02 |
| 31/05/2017 04:47 | 2.04 | | 3.07 | 0.35 | 3.90 | | 43.40 |
| 31/05/2017 05:47 | 2.87 | | 4.88 | 0.43 | 8.43 | | 46.30 |
| 31/05/2017 06:47 | 2.75 | | 4.81 | 0.48 | 5.93 | | 53.77 |
| 31/05/2017 07:47 | 3.02 | | 3.02 | 0.35 | 2.86 | | 90.40 |
| 31/05/2017 08:47 | 4.80 | | 2.19 | 0.21 | | | 102.99 |
| 31/05/2017 09:47 | 6.71 | | 1.98 | 0.15 | | | 109.41 |
| 10/06/2017 09:16 | | 6.02 | | | | | |
| 10/06/2017 10:01 | 2.12 | 23.54 | | | | | |

| Date/Time | Value 1 | Value 2 | | | | | |
|---|---|---|---|---|---|---|---|
| 10/06/2017 10:46 | 2.85 | 8.87 | | | | | |
| 10/06/2017 11:31 | 3.81 | 7.11 | | | | | |
| 10/06/2017 12:16 | 1.46 | | | | | | |
| 10/06/2017 13:01 | 10.43 | 31.80 | | | | | |
| 10/06/2017 13:46 | | | | | | | |
| 10/06/2017 14:31 | 12.18 | 27.98 | | | | | |
| 10/06/2017 15:16 | 8.28 | 28.81 | | | | | |
| 10/06/2017 15:48 | 6.18 | 19.74 | | | | | |
| 10/06/2017 16:28 | 8.25 | 25.83 | | | | | |
| 10/06/2017 17:09 | 5.54 | 16.83 | | | | | |
| 10/06/2017 19:08 | 9.04 | 27.95 | | | | | |
| 10/06/2017 21:08 | 13.04 | 29.30 | | | | | |
| 10/06/2017 23:07 | 4.25 | 14.25 | | | | | |
| 11/06/2017 01:07 | 4.54 | 6.61 | | | | | |
| 11/06/2017 03:07 | 2.69 | 8.43 | | | | | |
| 11/06/2017 05:06 | 3.83 | 5.99 | | | | | |
| 11/06/2017 07:06 | 3.25 | 12.26 | | | | | |
| 11/06/2017 09:05 | 7.74 | 16.11 | | | | | |
| 11/06/2017 09:50 | 6.33 | 18.70 | | | | | |
| 11/06/2017 10:35 | 6.73 | 22.11 | | | | | |
| 11/06/2017 11:20 | 9.81 | 35.83 | | | | | |
| 11/06/2017 12:05 | 13.00 | 41.50 | | | | | |
| 11/06/2017 12:50 | 15.14 | 42.80 | | | | | |
| 11/06/2017 13:35 | 13.30 | 34.87 | | | | | |
| 11/06/2017 14:20 | 10.88 | 16.02 | | | | | |
| 11/06/2017 15:05 | 7.20 | 17.90 | | | | | |
| 11/06/2017 15:50 | 7.42 | 16.27 | | | | | |
| 11/06/2017 16:35 | 4.31 | 20.89 | | | | | |
| 11/06/2017 17:20 | 5.76 | 24.13 | | | | | |
| 11/06/2017 18:05 | 7.74 | 17.72 | | | | | |
| 11/06/2017 19:35 | 4.38 | 8.92 | | | | | |
| 11/06/2017 21:04 | 5.09 | 9.51 | | | | | |
| 11/06/2017 22:34 | 5.54 | 8.51 | | | | | |
| 12/06/2017 00:04 | 3.00 | 8.70 | | | | | |
| 12/06/2017 01:33 | 1.92 | | | | | | |
| 12/06/2017 03:03 | 2.41 | 7.68 | | | | | |
| 12/06/2017 04:32 | | | | | | | |
| 12/06/2017 06:02 | | 4.86 | | | | | |
| 12/06/2017 07:31 | 1.75 | 1.03 | | | | | |
| 12/06/2017 09:01 | 1.75 | 3.83 | | | | | |
| 12/06/2017 09:46 | 0.39 | | | | | | |
| 12/06/2017 10:31 | 0.79 | | | | | | |
| 12/06/2017 11:16 | 3.08 | 10.90 | | | | | |
| 12/06/2017 12:04 | | | | | | | |
| 12/06/2017 13:42 | 8.90 | 24.61 | | | | | |

| | | | | | | |
|---|---|---|---|---|---|---|
| 12/06/2017 14:27 | 7.91 | 28.92 | | | | |
| 12/06/2017 15:48 | 8.62 | 30.32 | | | | |
| 12/06/2017 16:33 | 8.48 | 20.47 | | | | |
| 12/06/2017 17:19 | 8.26 | 28.70 | | | | |
| 12/06/2017 18:04 | 8.60 | 24.70 | | | | |
| 12/06/2017 19:34 | 4.51 | 13.89 | | | | |
| 12/06/2017 21:03 | 3.61 | 7.12 | | | | |
| 12/06/2017 22:33 | 3.31 | 19.44 | | | | |
| 13/06/2017 00:03 | 3.76 | 3.75 | | | | |
| 13/06/2017 01:32 | 1.50 | 2.06 | | | | |
| 13/06/2017 03:02 | | 7.19 | | | | |
| 13/06/2017 04:31 | | | | | | |
| 13/06/2017 06:01 | 1.53 | | | | | |
| 13/06/2017 07:30 | | | | | | |
| 13/06/2017 09:00 | 0.39 | 4.54 | | | | |
| 13/06/2017 09:45 | | 2.74 | | | | |
| 13/06/2017 11:23 | 1.98 | 1.55 | | | | |
| 13/06/2017 12:06 | 2.91 | 5.43 | | | | |
| 13/06/2017 12:51 | 3.78 | 8.52 | | | | |
| 13/06/2017 13:36 | 3.59 | 8.13 | | | | |
| 13/06/2017 14:21 | 7.92 | 25.62 | | | | |
| 13/06/2017 15:06 | 7.18 | 18.67 | | | | |
| 13/06/2017 15:51 | 7.41 | 12.02 | | | | |
| 13/06/2017 16:36 | 4.99 | 19.61 | | | | |
| 14/06/2017 12:29 | 17.25 | 53.40 | | | | |
| 14/06/2017 13:13 | 9.51 | 30.26 | | | | |
| 14/06/2017 13:58 | 10.06 | 30.48 | | | | |
| 14/06/2017 14:43 | 12.24 | 34.89 | | | | |
| 14/06/2017 15:28 | 11.91 | 36.67 | | | | |
| 14/06/2017 16:13 | 10.11 | 37.66 | | | | |
| 14/06/2017 16:58 | 9.93 | 37.82 | | | | |
| 14/06/2017 17:43 | 11.53 | 50.48 | | | | |
| 14/06/2017 18:28 | 18.87 | 75.97 | | | | |
| 14/06/2017 19:58 | 8.77 | 39.25 | | | | |
| 14/06/2017 21:28 | 5.31 | 30.86 | | | | |
| 14/06/2017 22:57 | 6.46 | 32.90 | | | | |
| 15/06/2017 00:27 | 9.13 | 28.21 | | | | |
| 15/06/2017 01:56 | 6.28 | 19.03 | | | | |
| 15/06/2017 03:26 | 3.06 | 9.75 | | | | |
| 15/06/2017 04:56 | 4.10 | 9.81 | | | | |
| 15/06/2017 06:25 | 2.31 | 5.52 | | | | |
| 15/06/2017 07:55 | 4.17 | 15.60 | | | | |
| 15/06/2017 09:24 | 11.28 | 42.95 | | | | |
| 15/06/2017 10:09 | 22.41 | 76.29 | | | | |
| 15/06/2017 10:54 | 20.89 | 79.56 | | | | |

| | | | | | | | |
|---|---|---|---|---|---|---|---|
| 15/06/2017 11:39 | 25.20 | 90.63 | | | | | |
| 15/06/2017 12:23 | 23.25 | 80.38 | | | | | |
| 15/06/2017 13:08 | 15.12 | 65.78 | | | | | |
| 15/06/2017 13:53 | 12.75 | 58.43 | | | | | |
| 15/06/2017 14:38 | 14.63 | 68.35 | | | | | |
| 15/06/2017 15:23 | 14.99 | 72.21 | | | | | |
| 15/06/2017 16:08 | 13.85 | 68.04 | | | | | |
| 15/06/2017 16:53 | 13.63 | 64.99 | | | | | |
| 15/06/2017 17:38 | 13.93 | 58.14 | | | | | |
| 15/06/2017 18:23 | 11.38 | 66.37 | | | | | |
| 15/06/2017 19:53 | 12.03 | 51.40 | | | | | |
| 15/06/2017 21:22 | 7.75 | 35.12 | | | | | |
| 15/06/2017 22:52 | 5.46 | 26.58 | | | | | |
| 16/06/2017 00:22 | 4.95 | 25.23 | | | | | |
| 16/06/2017 01:51 | 1.35 | 11.84 | | | | | |
| 16/06/2017 03:21 | 1.34 | 4.71 | | | | | |
| 16/06/2017 04:50 | 2.29 | 4.90 | | | | | |
| 16/06/2017 06:20 | 2.79 | 5.58 | | | | | |
| 16/06/2017 07:49 | 3.16 | 6.74 | | | | | |
| 16/06/2017 09:19 | 2.00 | | | | | | |
| 17/06/2017 17:42 | | | | | | | |
| 17/06/2017 18:46 | 6.31 | | | | | | |
| 17/06/2017 19:46 | 4.91 | | | | | | |
| 17/06/2017 20:46 | 4.97 | | | | | | |
| 17/06/2017 21:46 | 2.37 | | | | | | |
| 17/06/2017 22:46 | 2.95 | | | | | | |
| 17/06/2017 23:46 | 1.99 | | | | | | |
| 18/06/2017 00:46 | 1.97 | | | | | | |
| 18/06/2017 01:46 | 1.39 | | | | | | |
| 18/06/2017 02:46 | 1.29 | | | | | | |
| 18/06/2017 03:46 | 1.19 | | | | | | |
| 18/06/2017 04:46 | 1.09 | | | | | | |
| 18/06/2017 05:46 | 1.35 | | | | | | |
| 18/06/2017 06:46 | 1.28 | | | | | | |
| 18/06/2017 07:46 | 2.36 | | | | | | |
| 18/06/2017 08:46 | 3.80 | | | | | | |
| 18/06/2017 11:16 | 8.67 | | | | | | |
| 18/06/2017 12:16 | | | | | | | |
| 18/06/2017 13:16 | 10.53 | | | | | | |
| 18/06/2017 14:16 | 10.82 | | | | | | |
| 18/06/2017 15:16 | 9.09 | | | | | | |
| 18/06/2017 16:16 | 6.91 | | | | | | |
| 18/06/2017 17:16 | 6.55 | | | | | | |
| 18/06/2017 18:16 | 8.27 | | | | | | |
| 18/06/2017 19:16 | 8.33 | | | | | | |

| | | | | | | | |
|---|---|---|---|---|---|---|---|
| 18/06/2017 20:16 | 4.76 | | | | | | |
| 18/06/2017 21:16 | 3.09 | | | | | | |
| 18/06/2017 22:16 | 2.29 | | | | | | |
| 18/06/2017 23:16 | 2.21 | | | | | | |
| 19/06/2017 00:16 | 1.69 | | | | | | |
| 19/06/2017 01:16 | 1.45 | | | | | | |
| 19/06/2017 02:16 | 1.14 | | | | | | |
| 19/06/2017 03:16 | 0.95 | | | | | | |
| 19/06/2017 04:16 | 0.83 | | | | | | |
| 19/06/2017 05:16 | 0.88 | | | | | | |
| 19/06/2017 06:16 | 0.83 | | | | | | |
| 19/06/2017 07:16 | 2.08 | | | | | | |
| 19/06/2017 08:16 | 5.72 | | | | | | |
| 19/06/2017 09:16 | 6.67 | | | | | | |
| 19/06/2017 10:16 | 6.79 | | | | | | |
| 19/06/2017 11:16 | | | | | | | |
| 19/06/2017 12:16 | 8.57 | | | | | | 30.45 |
| 19/06/2017 13:16 | 7.85 | | | | | | 29.24 |
| 19/06/2017 14:16 | 5.46 | | | | | | 35.68 |
| 19/06/2017 15:16 | 5.22 | | 0.35 | 0.03 | 0.11 | | 32.48 |
| 19/06/2017 16:16 | 7.06 | | 0.41 | 0.06 | 0.16 | | 40.45 |
| 19/06/2017 17:16 | 11.53 | | 0.75 | 0.16 | 0.44 | | 48.54 |
| 19/06/2017 18:17 | 7.54 | | 0.47 | 0.10 | 0.31 | | 24.14 |
| 19/06/2017 19:16 | 7.24 | | 0.76 | 0.26 | 1.06 | | 36.53 |
| 19/06/2017 20:16 | 6.51 | | 3.81 | 0.88 | 7.93 | 0.41 | 67.83 |
| 19/06/2017 21:16 | 6.98 | | 8.74 | 2.11 | 20.91 | 0.90 | 142.65 |
| 19/06/2017 22:17 | 5.10 | | 7.61 | 1.66 | 15.80 | 0.69 | 168.21 |
| 19/06/2017 23:16 | 1.86 | | 1.43 | 0.39 | 2.98 | | 69.92 |
| 20/06/2017 00:17 | 1.89 | | 0.24 | 0.26 | 1.92 | | 59.92 |
| 20/06/2017 01:17 | 2.26 | | 1.34 | 0.34 | 2.50 | 0.14 | 54.60 |
| 20/06/2017 02:16 | 2.26 | | 1.50 | 0.36 | 2.67 | 0.12 | 49.25 |
| 20/06/2017 03:17 | 1.67 | | 1.15 | 0.26 | 2.03 | 0.10 | 38.48 |
| 20/06/2017 04:17 | 1.60 | | 1.15 | 0.22 | 2.01 | 0.09 | 44.52 |
| 20/06/2017 05:17 | 1.47 | | 1.08 | 0.27 | 1.85 | 0.08 | 44.50 |
| 20/06/2017 06:16 | 1.21 | | 0.57 | 0.20 | 1.01 | 0.06 | 34.15 |
| 20/06/2017 07:17 | 1.47 | | 0.38 | 0.16 | 0.56 | 0.03 | 30.09 |
| 20/06/2017 08:17 | 3.76 | | 0.31 | 0.13 | 0.26 | 0.02 | 33.54 |
| 20/06/2017 09:17 | 5.72 | | 0.23 | 0.11 | 0.11 | 0.03 | 37.38 |
| 20/06/2017 10:17 | 8.01 | | 0.28 | 0.07 | 0.07 | 0.02 | 41.37 |
| 20/06/2017 11:17 | 7.48 | | 0.30 | 0.05 | 0.08 | 0.03 | 46.07 |
| 20/06/2017 12:49 | 7.22 | | 0.36 | 0.06 | 0.07 | 0.04 | 53.18 |
| 20/06/2017 13:49 | 7.34 | | 0.40 | 0.04 | 0.09 | 0.06 | 59.79 |
| 20/06/2017 14:49 | 7.81 | | 0.45 | 0.05 | 0.08 | 0.04 | 57.81 |
| 20/06/2017 15:49 | 7.62 | | 0.41 | 0.06 | 0.11 | 0.04 | 58.83 |
| 20/06/2017 16:49 | 7.23 | | 0.37 | 0.06 | 0.13 | 0.02 | 55.59 |

| | | | | | | | |
|---|---|---|---|---|---|---|---|
| 20/06/2017 17:49 | 10.75 | | 0.57 | 0.15 | 0.31 | 0.05 | 61.42 |
| 20/06/2017 18:49 | 9.48 | | 0.70 | 0.24 | 0.75 | 0.05 | 58.17 |
| 20/06/2017 19:49 | 8.86 | | 2.18 | 0.62 | 4.15 | 0.23 | 70.45 |
| 20/06/2017 20:49 | 8.27 | | 4.04 | 0.91 | 8.90 | 0.40 | 88.35 |
| 20/06/2017 21:49 | 6.36 | | 3.12 | 0.88 | 5.75 | 0.29 | 112.38 |
| 20/06/2017 22:49 | 5.31 | | 4.09 | 1.08 | 7.02 | 0.36 | 189.71 |
| 20/06/2017 23:49 | 6.01 | | 9.28 | 2.10 | 21.74 | 0.95 | 181.21 |
| 21/06/2017 00:49 | 3.48 | | 6.18 | 1.46 | 14.41 | 0.60 | 150.17 |
| 21/06/2017 01:49 | 1.64 | | 2.63 | 0.80 | 5.26 | 0.22 | 139.67 |
| 21/06/2017 02:49 | 2.14 | | 3.40 | 0.73 | 5.62 | 0.29 | 198.08 |
| 21/06/2017 03:49 | 0.36 | | 0.64 | 0.18 | 1.20 | 0.05 | 212.10 |
| 21/06/2017 04:49 | 0.30 | | 0.63 | 0.13 | 0.81 | 0.04 | 93.21 |
| 21/06/2017 05:49 | 0.37 | | 0.52 | 0.17 | 0.72 | 0.03 | 73.44 |
| 21/06/2017 06:49 | 0.62 | | 0.48 | 0.18 | 0.61 | 0.04 | 74.17 |
| 21/06/2017 07:49 | 1.91 | | 0.39 | 0.16 | 0.41 | 0.04 | 71.11 |
| 21/06/2017 08:49 | 4.85 | | 0.39 | 0.14 | 0.24 | 0.04 | 68.14 |
| 21/06/2017 09:49 | 6.51 | | 0.27 | 0.10 | 0.11 | 0.04 | 95.44 |
| 21/06/2017 10:49 | 9.89 | | 0.26 | 0.06 | 0.05 | 0.03 | 87.41 |
| 21/06/2017 11:49 | 11.09 | | 0.37 | 0.08 | 0.06 | 0.03 | 116.18 |
| 21/06/2017 12:52 | 12.14 | | 0.40 | 0.09 | 0.08 | 0.02 | 115.90 |
| 21/06/2017 13:52 | 8.07 | | 0.31 | 0.05 | 0.06 | 0.03 | 134.33 |
| 21/06/2017 14:52 | 7.87 | | 0.38 | 0.10 | 0.16 | 0.03 | 125.57 |
| 21/06/2017 15:52 | 7.19 | | 0.41 | 0.11 | 0.34 | 0.05 | 114.89 |
| 21/06/2017 16:52 | 6.81 | | 0.58 | 0.21 | 0.68 | 0.06 | 119.98 |
| 21/06/2017 17:52 | 7.31 | | 0.74 | 0.28 | 0.76 | 0.06 | |
| 21/06/2017 18:52 | 7.16 | | 3.43 | 0.93 | 7.52 | 0.35 | 134.83 |
| 21/06/2017 19:52 | 5.57 | | 1.58 | 0.27 | 2.73 | 0.15 | 89.72 |
| 21/06/2017 20:52 | 2.62 | | 0.97 | 0.29 | 1.74 | 0.09 | 65.92 |
| 21/06/2017 21:52 | 1.25 | | 0.42 | 0.08 | 0.74 | 0.04 | 39.02 |
| 21/06/2017 22:52 | 0.91 | | 0.48 | 0.11 | 0.68 | 0.04 | 38.97 |
| 21/06/2017 23:52 | 0.61 | | 0.36 | 0.05 | 0.50 | 0.03 | 30.98 |
| 22/06/2017 00:52 | 0.57 | | 0.36 | 0.08 | 0.63 | 0.03 | 29.04 |
| 22/06/2017 01:52 | 0.53 | | 0.41 | 0.09 | 0.66 | 0.04 | 27.03 |
| 22/06/2017 02:52 | 0.65 | | 0.34 | 0.06 | 0.57 | 0.03 | 25.85 |
| 22/06/2017 03:52 | 0.59 | | 0.28 | 0.06 | 0.56 | 0.03 | 22.61 |
| 22/06/2017 04:52 | 0.61 | | 0.40 | 0.10 | 0.62 | 0.04 | 25.67 |
| 22/06/2017 05:52 | 0.61 | | 0.37 | 0.08 | 0.59 | 0.03 | 32.20 |
| 22/06/2017 06:52 | 0.62 | | 0.39 | 0.08 | 0.43 | 0.03 | 28.87 |
| 22/06/2017 07:52 | 0.68 | | 0.28 | 0.07 | 0.27 | 0.03 | 29.04 |
| 22/06/2017 08:52 | 0.95 | | 0.20 | 0.06 | 0.20 | 0.02 | 26.33 |
| 22/06/2017 09:52 | 1.28 | | 0.17 | 0.05 | 0.12 | | 22.94 |
| 22/06/2017 10:50 | 0.00 | | 0.00 | 0.00 | 0.00 | | |
| 22/06/2017 11:50 | 0.00 | | 0.00 | 0.00 | 0.00 | | |
| 22/06/2017 12:50 | 0.00 | | | | | | |
| 22/06/2017 13:27 | 0.77 | | 0.06 | 0.02 | | | 18.83 |

| | | | | | | |
|---|---|---|---|---|---|---|
| 22/06/2017 14:27 | 1.49 | | 0.08 | 0.03 | 0.03 | | 19.58 |
| 22/06/2017 15:27 | 1.06 | | 0.01 | 0.00 | 0.00 | | 19.50 |
| 22/06/2017 16:44 | 0.85 | | 0.41 | 0.03 | 0.70 | | 18.67 |
| 22/06/2017 17:44 | 1.16 | | 0.85 | 0.02 | 0.75 | | 14.54 |
| 22/06/2017 18:44 | 0.43 | | 1.07 | 0.02 | 2.91 | | 12.85 |
| 22/06/2017 19:44 | 0.41 | | 1.25 | 0.01 | 4.31 | | 11.32 |
| 22/06/2017 20:44 | 0.31 | | 0.95 | 0.01 | 3.36 | | 10.85 |
| 22/06/2017 21:44 | 0.25 | | 1.05 | 0.01 | 2.16 | | 10.86 |
| 22/06/2017 22:44 | 0.13 | | 0.95 | 0.02 | 1.08 | | 8.57 |
| 22/06/2017 23:44 | 0.14 | | 0.91 | 0.02 | 3.05 | | 9.06 |
| 23/06/2017 00:44 | 0.12 | | 1.10 | 0.02 | 0.51 | | 6.87 |
| 23/06/2017 01:44 | 0.11 | | 1.17 | 0.01 | 0.11 | | 5.19 |
| 23/06/2017 02:44 | 0.11 | | 0.95 | 0.03 | 2.35 | | 5.10 |
| 23/06/2017 03:44 | 0.10 | | 1.08 | 0.03 | 0.26 | | 6.81 |
| 23/06/2017 04:44 | 0.08 | | 0.86 | 0.01 | 0.64 | | 4.61 |
| 23/06/2017 05:44 | 0.08 | | 0.59 | 0.02 | 0.52 | | 4.68 |
| 23/06/2017 06:44 | 0.08 | | 0.82 | 0.01 | 0.37 | | 6.56 |
| 23/06/2017 07:44 | 0.07 | | 0.51 | 0.01 | 0.38 | | 4.91 |
| 23/06/2017 08:44 | 0.09 | | 0.28 | 0.03 | 0.54 | | 4.10 |
| 23/06/2017 09:44 | 0.14 | | 0.60 | 0.01 | 0.41 | | |
| 23/06/2017 10:44 | 0.14 | | 0.53 | 0.02 | 0.28 | | 4.27 |
| 23/06/2017 11:44 | 0.42 | | 0.96 | 0.01 | 0.36 | | 5.43 |
| 23/06/2017 12:44 | 0.33 | | 0.80 | 0.02 | 0.42 | | 4.42 |
| 23/06/2017 13:44 | 0.43 | | 0.83 | 0.02 | 0.10 | | 5.03 |
| 23/06/2017 14:57 | 0.27 | | 0.47 | 0.04 | 0.17 | | 4.70 |

**Table S2: Data plotted in Figs. 7 and S2.**

| Date Time (DD/MM/YYYY hh:mm) | Isoprene (ppb) | O$_3$ (ppb) | CO (ppm) | NO (ppb) | NO$_2$ (ppb) |
|---|---|---|---|---|---|
| 17/05/2017 17:40 | | 115 | 0.532 | 0.38 | |
| 17/05/2017 18:40 | | 107 | 0.497 | 0.17 | |
| 17/05/2017 19:40 | | 103 | 0.649 | 0.09 | |
| 17/05/2017 20:40 | | 108 | 0.708 | 0.10 | |
| 17/05/2017 21:40 | | 75 | 0.862 | 0.10 | |
| 17/05/2017 22:40 | | 42 | 1.048 | 0.21 | |
| 17/05/2017 23:40 | | 25 | 1.101 | 0.16 | |
| 18/05/2017 00:40 | | 30 | 0.989 | 0.13 | |
| 18/05/2017 01:40 | | 24 | 1.015 | 0.14 | |
| 18/05/2017 02:40 | | 26 | 0.786 | 0.12 | |
| 18/05/2017 03:40 | | 10 | 0.862 | 0.13 | |
| 18/05/2017 04:40 | | 13 | 0.758 | 0.14 | |
| 18/05/2017 05:40 | | 15 | 0.701 | 1.07 | |
| 18/05/2017 06:40 | | 27 | 0.711 | 2.69 | |
| 18/05/2017 07:40 | | 34 | 0.709 | 5.18 | |
| 18/05/2017 08:40 | | 44 | 0.851 | 5.85 | |
| 18/05/2017 10:44 | | 103 | 0.959 | 1.76 | |
| 18/05/2017 11:44 | | 127 | 0.631 | 0.53 | |
| 18/05/2017 12:44 | | 136 | 0.599 | 0.30 | |
| 18/05/2017 13:44 | | 137 | 0.557 | 0.32 | |
| 18/05/2017 14:44 | | 142 | 0.597 | 0.31 | |
| 18/05/2017 15:44 | 1.92 | 134 | 0.512 | 0.35 | |
| 18/05/2017 16:44 | 1.81 | 117 | 0.402 | 0.32 | |
| 18/05/2017 17:44 | 1.05 | 113 | 0.442 | 0.18 | |
| 18/05/2017 18:44 | 0.95 | 107 | 0.511 | 0.16 | |
| 18/05/2017 19:44 | 0.30 | 101 | 0.723 | 0.12 | |
| 18/05/2017 20:44 | 3.10 | 106 | 0.827 | 0.11 | |
| 18/05/2017 21:44 | 0.04 | 97 | 0.701 | 0.11 | |
| 18/05/2017 22:44 | 0.06 | 80 | 0.605 | 0.12 | |
| 18/05/2017 23:44 | 0.07 | 62 | 0.624 | 0.17 | |
| 19/05/2017 00:44 | 0.09 | 70 | 0.561 | 0.10 | |
| 19/05/2017 01:44 | 0.03 | 71 | 0.529 | 0.09 | |
| 19/05/2017 02:44 | 0.04 | 59 | 0.549 | 0.09 | |
| 19/05/2017 03:44 | 0.06 | 55 | 0.536 | 0.10 | |
| 19/05/2017 04:44 | 0.09 | 35 | 0.604 | 0.17 | |
| 19/05/2017 05:44 | 0.14 | 21 | 0.732 | 0.80 | |
| 19/05/2017 06:44 | 0.66 | 13 | 0.824 | 6.60 | |
| 19/05/2017 07:44 | 0.92 | 9 | 0.911 | 23.81 | |
| 19/05/2017 08:44 | 1.14 | 29 | 0.864 | 10.79 | |
| 19/05/2017 11:23 | 1.79 | 112 | 0.842 | 1.06 | |
| 19/05/2017 12:37 | 2.39 | 143 | 0.736 | 0.51 | |

| | | | | |
|---|---|---|---|---|
| 19/05/2017 13:47 | | 132 | 0.621 | 0.33 |
| 19/05/2017 14:54 | | 133 | 0.559 | 0.35 |
| 19/05/2017 15:54 | 2.67 | 129 | 0.534 | 2.65 |
| 19/05/2017 16:54 | 1.72 | 137 | 0.551 | 0.26 |
| 19/05/2017 17:54 | 0.95 | 114 | 0.416 | 0.18 |
| 19/05/2017 18:54 | | 95 | 0.533 | 0.16 |
| 19/05/2017 19:55 | 0.43 | 68 | 0.401 | 0.17 |
| 19/05/2017 20:55 | 0.07 | 58 | 0.325 | 0.18 |
| 19/05/2017 21:55 | 0.04 | 57 | 0.338 | 0.14 |
| 19/05/2017 22:55 | 0.04 | 53 | 0.375 | 0.13 |
| 19/05/2017 23:55 | | 54 | 0.404 | 0.10 |
| 20/05/2017 00:55 | 0.03 | 49 | 0.390 | 0.12 |
| 20/05/2017 01:55 | 0.05 | 45 | 0.446 | 0.10 |
| 20/05/2017 02:55 | 0.07 | 27 | 0.515 | 0.37 |
| 20/05/2017 03:55 | 4.26 | 29 | 0.559 | 0.13 |
| 20/05/2017 04:55 | 0.10 | 21 | 0.521 | 0.12 |
| 20/05/2017 05:55 | 5.42 | 5 | 0.592 | 8.89 |
| 20/05/2017 06:55 | 1.00 | 4 | 0.746 | 40.92 |
| 20/05/2017 07:55 | 1.14 | 7 | 0.784 | 44.72 |
| 20/05/2017 08:55 | 0.89 | 32 | 0.638 | 8.86 |
| 20/05/2017 09:55 | 1.58 | 54 | 0.837 | 4.51 |
| 20/05/2017 10:55 | 1.72 | 62 | 1.041 | 4.09 |
| 20/05/2017 11:54 | 1.69 | 104 | 0.601 | 1.47 |
| 20/05/2017 12:54 | 1.64 | 122 | 0.388 | 0.51 |
| 20/05/2017 13:54 | 1.56 | 110 | 0.286 | 0.48 |
| 20/05/2017 14:54 | 2.38 | 111 | 0.281 | 0.43 |
| 20/05/2017 15:54 | | 103 | 0.237 | 0.46 |
| 20/05/2017 16:54 | | 114 | 0.350 | 0.28 |
| 20/05/2017 17:54 | | 116 | 0.423 | 0.26 |
| 20/05/2017 18:54 | | 104 | 0.493 | 0.21 |
| 20/05/2017 19:54 | | 77 | 0.452 | 0.26 |
| 20/05/2017 20:54 | | 88 | 0.577 | 0.27 |
| 20/05/2017 21:54 | | 77 | 0.538 | 0.18 |
| 20/05/2017 22:54 | | 65 | 0.738 | 0.13 |
| 20/05/2017 23:54 | 0.05 | 68 | 0.838 | 0.20 |
| 21/05/2017 00:54 | 0.04 | 64 | 0.726 | 0.14 |
| 21/05/2017 01:54 | 0.03 | 50 | 0.566 | 0.14 |
| 21/05/2017 02:54 | 0.02 | 40 | 0.550 | 0.16 |
| 21/05/2017 03:54 | 0.02 | 32 | 0.888 | 0.17 |
| 21/05/2017 04:54 | 0.03 | 26 | 1.019 | 0.24 |
| 21/05/2017 05:54 | 0.04 | 19 | 1.010 | 2.04 |
| 21/05/2017 06:54 | 0.13 | 23 | 0.951 | 3.04 |
| 21/05/2017 07:54 | 0.21 | 24 | 0.971 | 4.06 |
| 21/05/2017 08:54 | 0.56 | 32 | 0.874 | 3.45 |
| 21/05/2017 09:54 | 0.88 | 53 | 0.822 | 2.17 |

| | | | | | |
|---|---|---|---|---|---|
| 21/05/2017 10:59 | 0.74 | 68 | 0.817 | 1.81 | |
| 21/05/2017 11:59 | 0.98 | 75 | 0.781 | 1.73 | |
| 21/05/2017 12:59 | | 95 | 0.730 | 0.89 | 12.23 |
| 21/05/2017 13:59 | 0.68 | 99 | 0.576 | 0.56 | 7.99 |
| 21/05/2017 14:59 | | 94 | 0.415 | 0.50 | 7.50 |
| 21/05/2017 15:56 | 0.83 | 90 | 0.260 | 0.44 | 7.33 |
| 21/05/2017 16:56 | 0.73 | 81 | 0.220 | 0.45 | 8.73 |
| 21/05/2017 18:05 | 0.63 | 69 | 0.299 | 0.41 | 13.72 |
| 21/05/2017 19:05 | 0.47 | 68 | 0.415 | 0.41 | 13.16 |
| 21/05/2017 20:05 | 0.27 | 54 | 0.403 | 0.10 | 16.74 |
| 21/05/2017 21:05 | 0.10 | 52 | 0.294 | 0.16 | 14.89 |
| 21/05/2017 22:05 | 0.05 | 54 | 0.308 | 0.27 | 11.59 |
| 21/05/2017 23:05 | 0.05 | 53 | 0.306 | 0.44 | 10.11 |
| 22/05/2017 00:05 | 0.05 | 51 | 0.293 | | 7.00 |
| 22/05/2017 01:05 | 0.03 | 38 | 0.416 | 0.30 | 13.72 |
| 22/05/2017 02:05 | 0.04 | 34 | 0.401 | 0.10 | 11.71 |
| 22/05/2017 03:05 | 0.03 | 26 | 0.651 | 0.24 | 18.27 |
| 22/05/2017 04:05 | 0.08 | 26 | 0.819 | 0.10 | 17.07 |
| 22/05/2017 05:05 | 0.01 | 17 | 1.164 | 0.24 | 22.47 |
| 22/05/2017 06:05 | | 16 | 1.370 | 0.94 | 21.96 |
| 22/05/2017 07:05 | | 12 | 1.449 | 2.38 | 25.71 |
| 22/05/2017 08:05 | | 10 | 1.304 | 1.24 | 25.41 |
| 22/05/2017 09:05 | | 9 | 1.404 | 2.14 | 24.74 |
| 22/05/2017 10:05 | | 10 | 1.196 | 3.84 | 23.25 |
| 22/05/2017 10:46 | | 12 | 1.131 | 4.17 | 23.37 |
| 22/05/2017 11:46 | | 20 | 0.697 | 3.03 | 15.31 |
| 22/05/2017 12:46 | | 18 | 0.454 | 3.36 | 16.48 |
| 22/05/2017 13:46 | | 24 | 0.335 | 1.57 | 11.66 |
| 22/05/2017 14:46 | | 21 | 0.315 | 2.57 | 13.63 |
| 22/05/2017 15:44 | | 18 | 0.262 | 2.99 | 16.79 |
| 22/05/2017 16:44 | | 17 | 0.212 | 3.14 | 17.56 |
| 22/05/2017 17:44 | | 12 | 0.284 | 2.91 | 22.10 |
| 22/05/2017 18:44 | | 7 | 0.284 | 4.18 | 26.26 |
| 22/05/2017 19:44 | | 2 | 0.196 | 2.61 | 26.15 |
| 22/05/2017 20:45 | | 3 | 0.113 | 2.01 | 23.15 |
| 22/05/2017 21:45 | 0.48 | 1 | 0.230 | 10.99 | 26.08 |
| 22/05/2017 22:45 | | 1 | 0.175 | 14.74 | 26.31 |
| 22/05/2017 23:45 | 0.36 | 1 | 0.180 | 17.50 | 24.08 |
| 23/05/2017 00:45 | 0.32 | 1 | 0.037 | 13.25 | 24.58 |
| 23/05/2017 01:45 | 0.32 | 1 | 0.080 | 16.82 | 23.19 |
| 23/05/2017 02:45 | 0.35 | 1 | 0.221 | 26.91 | 21.74 |
| 23/05/2017 03:45 | | 1 | 0.150 | 29.86 | 21.52 |
| 23/05/2017 04:45 | 0.42 | 2 | 0.148 | 43.76 | 17.36 |
| 23/05/2017 05:45 | 0.49 | 2 | 0.122 | 47.62 | 20.13 |
| 23/05/2017 06:45 | 0.32 | 6 | | 14.71 | 24.14 |

| | | | | | |
|---|---|---|---|---|---|
| 23/05/2017 07:45 | 0.35 | 17 | | 9.84 | 21.55 |
| 23/05/2017 08:45 | 0.52 | 24 | 0.156 | 7.71 | 19.08 |
| 23/05/2017 10:05 | 0.54 | 32 | | 5.45 | 16.62 |
| 23/05/2017 10:51 | 0.84 | 42 | 0.137 | 2.85 | 10.33 |
| 23/05/2017 11:51 | 0.92 | 48 | 0.173 | 7.01 | 10.53 |
| 23/05/2017 12:51 | 0.91 | 56 | 0.175 | 1.74 | 8.24 |
| 23/05/2017 13:51 | | 61 | 0.114 | 1.30 | 5.50 |
| 23/05/2017 14:51 | 1.06 | 61 | 0.095 | 0.98 | 6.03 |
| 23/05/2017 15:51 | 1.42 | 56 | 0.094 | 1.06 | 6.09 |
| 23/05/2017 16:51 | 1.47 | 63 | 0.185 | 1.02 | 9.31 |
| 23/05/2017 17:51 | 1.03 | 79 | 0.291 | 0.66 | 11.80 |
| 23/05/2017 18:51 | 0.64 | 71 | 0.343 | 0.48 | 16.19 |
| 23/05/2017 19:51 | 0.24 | 62 | 0.306 | 0.30 | 15.85 |
| 23/05/2017 20:51 | 0.10 | 47 | 0.346 | 1.06 | 22.24 |
| 23/05/2017 21:51 | 0.08 | 36 | 0.389 | 0.73 | 24.32 |
| 23/05/2017 22:51 | 0.09 | 30 | 0.350 | 1.04 | 24.27 |
| 23/05/2017 23:51 | 0.17 | 8 | 0.444 | 4.81 | 41.27 |
| 24/05/2017 00:51 | 0.16 | 4 | 0.433 | 9.39 | 44.41 |
| 24/05/2017 01:51 | 0.27 | 2 | 0.464 | 23.52 | 45.31 |
| 24/05/2017 02:51 | 4.41 | 2 | 0.742 | 58.06 | 43.94 |
| 24/05/2017 03:51 | 4.96 | 2 | 0.667 | 86.67 | 40.98 |
| 24/05/2017 04:51 | 6.13 | 1 | 0.621 | 84.02 | 41.81 |
| 24/05/2017 05:51 | | 1 | 0.486 | 50.71 | 36.48 |
| 24/05/2017 06:51 | | 6 | 0.467 | 33.58 | 38.58 |
| 24/05/2017 07:51 | | 9 | 0.379 | 33.94 | 40.23 |
| 24/05/2017 08:51 | | 21 | 0.320 | 17.04 | 33.59 |
| 24/05/2017 09:51 | | 42 | 0.171 | 3.43 | 14.71 |
| 24/05/2017 10:51 | | 52 | 0.136 | 2.74 | 12.52 |
| 24/05/2017 11:51 | | 57 | 0.121 | 11.44 | 11.84 |
| 24/05/2017 12:51 | 1.04 | 69 | 0.144 | 1.81 | 11.65 |
| 24/05/2017 13:51 | 1.16 | 78 | 0.127 | 0.87 | 8.16 |
| 24/05/2017 14:51 | 1.52 | 84 | 0.149 | 0.78 | 7.99 |
| 24/05/2017 15:51 | 0.83 | 75 | 0.134 | 0.83 | 10.60 |
| 24/05/2017 16:51 | 1.09 | 81 | 0.226 | 0.77 | 11.33 |
| 24/05/2017 17:51 | 4.59 | 74 | 0.211 | 0.67 | 10.61 |
| 24/05/2017 18:51 | 0.54 | 63 | 0.232 | 0.63 | 14.92 |
| 24/05/2017 19:51 | 0.27 | 60 | 0.222 | 0.66 | 12.78 |
| 24/05/2017 20:51 | 0.08 | 47 | 0.266 | 0.84 | 20.00 |
| 24/05/2017 21:51 | 0.06 | 38 | 0.302 | 0.61 | 23.43 |
| 24/05/2017 22:51 | 0.12 | 13 | 0.379 | 1.71 | 42.69 |
| 24/05/2017 23:51 | 0.24 | 3 | 0.497 | 7.78 | 53.25 |
| 25/05/2017 00:51 | 6.11 | 2 | 0.713 | 43.72 | 52.43 |
| 25/05/2017 01:51 | 0.44 | 1 | 0.515 | 38.02 | 46.93 |
| 25/05/2017 02:51 | | 1 | 0.372 | 29.11 | 45.07 |
| 25/05/2017 03:51 | 0.17 | 1 | 0.242 | 19.62 | 39.01 |

| | | | | | |
|---|---|---|---|---|---|
| 25/05/2017 04:51 | 0.23 | 1 | 0.276 | 19.84 | 42.48 |
| 25/05/2017 05:51 | 0.57 | 2 | 0.303 | 27.72 | 37.06 |
| 25/05/2017 06:51 | 0.92 | 12 | 0.263 | 11.32 | 30.45 |
| 25/05/2017 07:51 | 1.58 | 17 | 0.306 | 11.87 | 27.68 |
| 25/05/2017 08:51 | 1.44 | 38 | 0.207 | 2.98 | 12.52 |
| 25/05/2017 09:51 | 1.50 | 48 | 0.189 | 1.97 | 10.71 |
| 25/05/2017 10:51 | 1.86 | 47 | 0.464 | 4.23 | 20.06 |
| 25/05/2017 11:51 | 1.36 | 59 | 0.654 | 3.04 | 19.41 |
| 25/05/2017 12:51 | | 74 | 0.635 | 1.67 | 15.12 |
| 25/05/2017 13:50 | | 88 | 0.562 | 1.01 | 11.81 |
| 25/05/2017 14:50 | | 91 | 0.547 | 1.31 | 14.46 |
| 25/05/2017 15:50 | | 94 | 0.459 | 0.86 | 11.43 |
| 25/05/2017 16:50 | | 82 | 0.352 | 0.84 | 13.11 |
| 25/05/2017 17:50 | | 77 | 0.377 | 0.49 | 10.76 |
| 25/05/2017 18:50 | | 68 | 0.391 | 0.32 | 12.34 |
| 25/05/2017 19:50 | 0.26 | 57 | 0.345 | 0.50 | 14.44 |
| 25/05/2017 20:50 | 0.11 | 42 | 0.374 | 0.62 | 19.39 |
| 25/05/2017 21:50 | 0.04 | 40 | 0.281 | 0.59 | 16.08 |
| 25/05/2017 22:50 | 0.03 | 37 | 0.281 | 0.48 | 16.56 |
| 25/05/2017 23:50 | 0.02 | 32 | 0.310 | 0.88 | 17.51 |
| 26/05/2017 00:50 | 0.00 | 27 | 0.334 | 0.81 | 20.12 |
| 26/05/2017 01:50 | 0.00 | 26 | 0.367 | 0.43 | 18.46 |
| 26/05/2017 02:50 | 0.00 | 22 | 0.397 | 0.54 | 19.13 |
| 26/05/2017 03:50 | | 13 | 0.429 | 0.94 | 28.43 |
| 26/05/2017 04:50 | 0.04 | 1 | 0.513 | 7.48 | 41.12 |
| 26/05/2017 05:50 | 0.11 | 2 | 0.427 | 19.84 | 34.74 |
| 26/05/2017 06:50 | 0.10 | 9 | 0.427 | 15.31 | 30.38 |
| 26/05/2017 07:50 | 0.47 | 10 | 0.455 | 19.18 | 34.49 |
| 26/05/2017 08:50 | | 16 | 0.460 | 17.50 | 31.06 |
| 26/05/2017 09:50 | 0.65 | 21 | 0.489 | 13.19 | 31.97 |
| 26/05/2017 10:50 | 0.66 | 30 | 0.530 | 11.29 | 31.60 |
| 26/05/2017 11:50 | 0.92 | 44 | 0.528 | 6.92 | 31.21 |
| 26/05/2017 12:50 | 0.73 | 59 | 0.466 | 3.79 | 25.62 |
| 26/05/2017 13:50 | 0.66 | 79 | 0.393 | 2.06 | 20.41 |
| 26/05/2017 14:50 | 0.57 | 111 | 0.372 | 0.86 | 13.13 |
| 26/05/2017 16:02 | 0.51 | 122 | 0.404 | 0.43 | 11.68 |
| 26/05/2017 17:02 | 0.82 | 123 | 0.464 | 0.62 | 18.56 |
| 26/05/2017 18:02 | 0.65 | 119 | 0.498 | 0.33 | 17.51 |
| 26/05/2017 19:02 | 0.49 | 118 | 0.476 | 0.25 | 16.93 |
| 26/05/2017 20:02 | 0.11 | 92 | 0.596 | 0.51 | 29.48 |
| 26/05/2017 21:02 | 0.06 | 83 | 0.629 | 0.37 | 30.79 |
| 26/05/2017 22:02 | 0.17 | 62 | 0.731 | 0.49 | 38.30 |
| 26/05/2017 23:02 | 2.28 | 50 | 0.558 | 0.28 | 26.55 |
| 27/05/2017 00:02 | | 43 | 0.575 | 0.54 | 23.01 |
| 27/05/2017 01:02 | | 36 | 0.644 | 0.41 | 19.28 |

| | | | | | |
|---|---|---|---|---|---|
| 27/05/2017 02:02 | 0.06 | 27 | 0.547 | 0.16 | 21.47 |
| 27/05/2017 03:02 | 0.02 | 29 | 0.510 | 0.12 | 19.75 |
| 27/05/2017 04:02 | 0.00 | 27 | 0.653 | 0.57 | 22.47 |
| 27/05/2017 05:02 | 0.03 | 15 | 0.853 | 3.41 | 41.68 |
| 27/05/2017 06:02 | 0.32 | 19 | 0.878 | 4.81 | 38.50 |
| 27/05/2017 07:02 | 0.17 | 28 | 1.168 | 6.24 | 35.04 |
| 27/05/2017 08:02 | 0.32 | 41 | 1.084 | 4.74 | 27.35 |
| 27/05/2017 09:02 | 0.52 | 44 | 1.013 | 5.56 | 28.28 |
| 27/05/2017 10:02 | 0.84 | 51 | 0.955 | 4.74 | 27.28 |
| 27/05/2017 11:02 | 0.82 | 60 | 0.926 | 4.63 | 30.07 |
| 27/05/2017 12:02 | 0.76 | 80 | 1.069 | 2.43 | 22.76 |
| 27/05/2017 13:02 | 1.14 | 94 | 1.266 | 1.88 | 22.59 |
| 27/05/2017 14:02 | 1.29 | 107 | 1.312 | 1.38 | 22.07 |
| 27/05/2017 15:02 | 0.93 | 130 | 1.137 | 0.89 | 18.87 |
| 27/05/2017 16:02 | 1.03 | 144 | 1.160 | 0.59 | 18.61 |
| 27/05/2017 17:02 | | 141 | 1.304 | 0.42 | 19.42 |
| 28/05/2017 12:28 | | 78 | 0.367 | 0.73 | 8.86 |
| 28/05/2017 13:49 | | 65 | 0.121 | 0.70 | 6.88 |
| 28/05/2017 14:45 | | 57 | 0.094 | 0.69 | 6.20 |
| 28/05/2017 15:40 | 0.00 | 144 | 0.873 | 0.27 | 11.58 |
| 28/05/2017 16:22 | 1.27 | 156 | 1.211 | 0.25 | 14.36 |
| 28/05/2017 17:22 | | 184 | 2.055 | | 15.65 |
| 28/05/2017 18:22 | | 157 | 2.057 | 0.11 | 15.98 |
| 28/05/2017 19:22 | | 137 | 2.077 | 0.12 | 20.00 |
| 28/05/2017 20:22 | | 117 | 0.860 | 0.13 | 9.97 |
| 28/05/2017 21:22 | | 104 | 0.727 | 0.12 | 13.50 |
| 28/05/2017 22:22 | | 95 | 0.705 | 0.15 | 11.30 |
| 28/05/2017 23:22 | | 69 | 0.753 | 0.10 | 14.25 |
| 29/05/2017 00:22 | | 51 | 0.784 | 0.10 | 15.13 |
| 29/05/2017 01:22 | | 47 | 0.696 | 0.11 | 14.32 |
| 29/05/2017 02:22 | | 46 | 0.831 | 0.55 | 16.52 |
| 29/05/2017 03:22 | | 37 | 0.820 | 0.08 | 20.25 |
| 29/05/2017 04:22 | | 38 | 0.734 | 0.95 | 20.04 |
| 29/05/2017 05:22 | | 36 | 0.519 | 0.16 | 22.60 |
| 29/05/2017 06:22 | | 47 | 0.507 | 0.34 | 16.88 |
| 29/05/2017 07:22 | | 39 | 0.499 | 1.30 | 20.26 |
| 29/05/2017 08:22 | | 41 | 0.281 | 1.52 | 15.07 |
| 29/05/2017 09:22 | | 39 | 0.225 | 1.46 | 13.94 |
| 29/05/2017 10:22 | | 34 | 0.377 | 2.48 | 18.94 |
| 29/05/2017 11:22 | | 39 | 0.262 | 0.90 | 11.72 |
| 29/05/2017 12:22 | | 44 | 0.223 | 0.93 | 9.62 |
| 29/05/2017 13:25 | | 49 | 0.257 | 1.03 | 7.45 |
| 29/05/2017 14:37 | | 44 | 0.335 | 1.75 | 12.87 |
| 29/05/2017 16:50 | | 58 | 0.307 | 0.41 | 9.02 |
| 29/05/2017 17:50 | | 61 | 0.303 | 0.22 | 7.46 |

| | | | | | |
|---|---|---|---|---|---|
| 29/05/2017 18:50 | | 58 | 0.324 | 0.37 | 10.32 |
| 29/05/2017 19:50 | 0.03 | 55 | 0.344 | 1.07 | 13.10 |
| 29/05/2017 20:50 | 0.03 | 56 | 0.378 | 0.69 | 13.81 |
| 29/05/2017 21:50 | 0.04 | 47 | 0.408 | 0.85 | 20.25 |
| 29/05/2017 22:50 | 0.04 | 38 | 0.367 | 0.44 | 18.00 |
| 29/05/2017 23:50 | 4.13 | 34 | 0.407 | 0.18 | 20.42 |
| 30/05/2017 00:50 | 0.03 | 29 | 0.410 | 0.14 | 23.12 |
| 30/05/2017 01:50 | 0.04 | 17 | 0.484 | 5.29 | 39.37 |
| 30/05/2017 02:50 | 0.05 | 8 | 0.389 | 1.01 | 37.30 |
| 30/05/2017 03:50 | 0.04 | 21 | 0.323 | 1.24 | 17.45 |
| 30/05/2017 04:50 | 0.05 | 14 | 0.334 | 0.66 | 25.02 |
| 30/05/2017 05:50 | 0.05 | 21 | 0.313 | 1.09 | 25.28 |
| 30/05/2017 06:50 | 0.37 | 17 | 0.358 | 3.98 | 26.62 |
| 30/05/2017 07:50 | | 11 | 0.469 | 13.61 | 35.26 |
| 30/05/2017 08:50 | 0.19 | 23 | 0.393 | 3.09 | 24.71 |
| 30/05/2017 09:50 | 0.46 | 36 | 0.394 | 2.24 | 17.80 |
| 30/05/2017 10:50 | 0.47 | 34 | 0.427 | 3.36 | 23.04 |
| 30/05/2017 11:50 | 0.48 | 41 | 0.474 | 4.86 | 20.13 |
| 30/05/2017 12:50 | 0.62 | 54 | 0.483 | 3.34 | 16.91 |
| 30/05/2017 13:47 | 0.68 | 66 | 0.406 | 2.27 | 14.06 |
| 30/05/2017 14:47 | 0.65 | 86 | 0.301 | 0.95 | 7.37 |
| 30/05/2017 15:47 | 0.44 | 98 | 0.295 | 0.83 | 8.87 |
| 30/05/2017 16:47 | 0.53 | 86 | 0.326 | 0.79 | 13.26 |
| 30/05/2017 17:47 | 0.49 | 87 | 0.331 | 0.41 | 10.88 |
| 30/05/2017 18:47 | 0.30 | 80 | 0.362 | 0.21 | 11.56 |
| 30/05/2017 19:47 | 0.09 | 73 | 0.425 | 0.38 | 17.54 |
| 30/05/2017 20:47 | 0.04 | 70 | 0.440 | 0.22 | 14.64 |
| 30/05/2017 21:47 | 0.06 | 57 | 0.511 | 0.32 | 19.15 |
| 30/05/2017 22:47 | | 48 | 0.574 | 0.23 | 17.58 |
| 30/05/2017 23:47 | | 39 | 0.554 | 0.29 | 18.45 |
| 31/05/2017 00:47 | 0.07 | 30 | 0.563 | 0.49 | 19.97 |
| 31/05/2017 01:47 | 0.09 | 21 | 0.574 | 0.41 | 20.67 |
| 31/05/2017 02:47 | 0.08 | 17 | 0.550 | 0.73 | 20.93 |
| 31/05/2017 03:47 | 0.10 | 1 | 0.596 | 10.62 | 44.81 |
| 31/05/2017 04:47 | 0.09 | 1 | 0.665 | 25.60 | 46.45 |
| 31/05/2017 05:47 | 0.28 | 2 | 0.656 | 24.66 | 42.85 |
| 31/05/2017 06:47 | 0.29 | 5 | 0.761 | 27.65 | 48.40 |
| 31/05/2017 07:47 | | 20 | 0.741 | 10.78 | 39.56 |
| 31/05/2017 08:47 | | 36 | 0.712 | 7.24 | 35.36 |
| 31/05/2017 09:47 | 1.37 | 53 | 0.727 | 5.48 | 33.83 |
| 10/06/2017 09:16 | 0.76 | 34 | 0.446 | 4.27 | 13.40 |
| 10/06/2017 10:01 | 0.91 | 35 | 0.437 | 3.25 | 13.71 |
| 10/06/2017 10:46 | 0.91 | 46 | 0.323 | 3.91 | 15.16 |
| 10/06/2017 11:31 | 1.03 | 40 | 0.259 | 2.83 | 11.21 |
| 10/06/2017 12:16 | | 52 | 0.289 | 2.18 | 8.97 |

| | | | | | |
|---|---|---|---|---|---|
| 10/06/2017 13:01 | | 51 | 0.392 | 2.16 | 10.10 |
| 10/06/2017 13:46 | | 57 | 0.424 | 2.38 | 11.39 |
| 10/06/2017 14:31 | 0.81 | 61 | 0.402 | 2.34 | 12.08 |
| 10/06/2017 15:16 | 0.70 | 67 | 0.285 | 1.41 | 9.53 |
| 10/06/2017 15:48 | 0.70 | 72 | 0.294 | 2.68 | 12.39 |
| 10/06/2017 16:28 | | 72 | 0.224 | 1.67 | 10.56 |
| 10/06/2017 17:09 | | 76 | 0.227 | 0.78 | 9.22 |
| 10/06/2017 19:08 | 0.33 | 56 | 0.253 | 2.65 | 21.39 |
| 10/06/2017 21:08 | 0.15 | 35 | 0.232 | 5.31 | 22.86 |
| 10/06/2017 23:07 | 0.03 | 26 | 0.421 | 3.08 | 23.30 |
| 11/06/2017 01:07 | 0.13 | 7 | 0.403 | 21.86 | 56.96 |
| 11/06/2017 03:07 | 0.19 | 1 | 0.580 | 47.73 | 64.59 |
| 11/06/2017 05:06 | | 1 | 0.620 | 99.96 | 61.24 |
| 11/06/2017 07:06 | | 3 | 0.565 | 66.99 | 43.97 |
| 11/06/2017 09:05 | 0.78 | 11 | 0.570 | 31.70 | 43.20 |
| 11/06/2017 09:50 | 1.27 | 19 | 0.569 | 19.09 | 39.15 |
| 11/06/2017 10:35 | 1.14 | 33 | 0.439 | 8.29 | 24.00 |
| 11/06/2017 11:20 | 1.06 | 34 | 0.377 | 9.60 | 29.13 |
| 11/06/2017 12:05 | 1.06 | 52 | 0.349 | 3.60 | 18.02 |
| 11/06/2017 12:50 | 0.97 | 67 | 0.294 | 3.09 | 14.16 |
| 11/06/2017 13:35 | 0.68 | 88 | 0.250 | 1.93 | 10.68 |
| 11/06/2017 14:20 | 0.70 | 91 | 0.326 | 2.11 | 12.61 |
| 11/06/2017 15:05 | 0.70 | 99 | 0.362 | 1.11 | 8.10 |
| 11/06/2017 15:50 | 0.51 | 110 | 0.401 | 0.75 | 7.35 |
| 11/06/2017 16:35 | 0.34 | 116 | 0.499 | 1.10 | 15.41 |
| 11/06/2017 17:20 | 0.28 | 115 | 0.604 | 1.93 | 24.98 |
| 11/06/2017 18:05 | 0.28 | 117 | 0.618 | 1.31 | 18.64 |
| 11/06/2017 19:35 | 0.10 | 90 | 0.658 | 6.12 | 27.40 |
| 11/06/2017 21:04 | 0.04 | 46 | 0.796 | 23.84 | 77.41 |
| 11/06/2017 22:34 | 0.04 | 33 | 0.426 | 9.81 | 43.07 |
| 12/06/2017 00:04 | 0.06 | 4 | 0.424 | 29.68 | 75.62 |
| 12/06/2017 01:33 | 0.05 | 5 | 0.552 | 15.19 | 71.39 |
| 12/06/2017 03:03 | 0.04 | 4 | 0.466 | 25.46 | 69.21 |
| 12/06/2017 04:32 | 0.02 | 13 | 0.586 | 7.17 | 41.40 |
| 12/06/2017 06:02 | 0.05 | 15 | 1.299 | 7.04 | 39.75 |
| 12/06/2017 07:31 | 0.58 | 26 | 1.190 | 11.15 | 40.20 |
| 12/06/2017 09:01 | 0.62 | 30 | 1.032 | 7.29 | 36.29 |
| 12/06/2017 09:46 | 0.96 | 39 | 0.967 | 8.65 | 29.57 |
| 12/06/2017 10:31 | 1.04 | 50 | 0.840 | 4.25 | 23.84 |
| 12/06/2017 11:16 | | 53 | 0.793 | 3.42 | 24.31 |
| 12/06/2017 12:04 | | 63 | 0.712 | 2.31 | 16.83 |
| 12/06/2017 13:42 | | 65 | 0.794 | 3.22 | 18.43 |
| 12/06/2017 14:27 | | 67 | 0.782 | 2.97 | 18.83 |
| 12/06/2017 15:48 | | 80 | 0.768 | 1.21 | 14.21 |
| 12/06/2017 16:33 | 0.87 | 82 | 0.691 | 0.76 | 13.78 |

| | | | | | |
|---|---|---|---|---|---|
| 12/06/2017 17:19 | 0.41 | 82 | 0.667 | 0.42 | 13.36 |
| 12/06/2017 18:04 | 0.39 | 72 | 0.727 | 0.47 | 18.39 |
| 12/06/2017 19:34 | 0.21 | 67 | 0.640 | 0.11 | 17.71 |
| 12/06/2017 21:03 | 0.00 | 62 | 0.644 | 0.60 | 18.65 |
| 12/06/2017 22:33 | 0.00 | 58 | 0.574 | 0.70 | 15.86 |
| 13/06/2017 00:03 | 0.00 | 58 | 0.492 | 0.85 | 12.58 |
| 13/06/2017 01:32 | 0.00 | 50 | 0.457 | 2.72 | 21.00 |
| 13/06/2017 03:02 | 0.01 | 55 | 0.387 | 0.63 | 15.99 |
| 13/06/2017 04:31 | 0.00 | 41 | 0.448 | 0.85 | 22.00 |
| 13/06/2017 06:01 | 0.04 | 24 | 0.522 | 2.92 | 38.80 |
| 13/06/2017 07:30 | | 29 | 0.621 | 2.66 | 33.95 |
| 13/06/2017 09:00 | 0.06 | 28 | 0.739 | 1.78 | 30.00 |
| 13/06/2017 09:45 | 0.14 | 38 | 0.629 | 5.08 | 27.05 |
| 13/06/2017 11:23 | 0.68 | 35 | 0.864 | 4.81 | 29.18 |
| 13/06/2017 12:06 | 0.40 | 49 | 0.596 | 2.92 | 23.40 |
| 13/06/2017 12:51 | 0.40 | 63 | 0.502 | 2.71 | 15.78 |
| 13/06/2017 13:36 | 0.53 | 77 | 0.432 | 1.21 | 9.84 |
| 13/06/2017 14:21 | 0.62 | 82 | 0.391 | 1.09 | 9.85 |
| 13/06/2017 15:06 | 0.38 | 84 | 0.415 | 0.89 | 9.43 |
| 13/06/2017 15:51 | 0.38 | 87 | 0.360 | 0.44 | 6.56 |
| 13/06/2017 16:36 | 0.53 | 87 | 0.340 | 0.48 | 7.87 |
| 14/06/2017 12:29 | 1.51 | 137 | 0.395 | 0.52 | 9.87 |
| 14/06/2017 13:13 | 1.20 | 107 | 0.234 | 0.42 | 6.17 |
| 14/06/2017 13:58 | 1.20 | 93 | 0.179 | 0.46 | 5.97 |
| 14/06/2017 14:43 | 1.28 | 92 | 0.154 | 0.33 | 5.08 |
| 14/06/2017 15:28 | 1.84 | 88 | 0.139 | 0.35 | 5.50 |
| 14/06/2017 16:13 | | 78 | 0.167 | 0.58 | 7.83 |
| 14/06/2017 16:58 | | 77 | 0.182 | 0.33 | 8.59 |
| 14/06/2017 17:43 | 1.52 | 81 | 0.245 | 0.38 | 11.14 |
| 14/06/2017 18:28 | | 73 | 0.271 | 0.31 | 17.15 |
| 14/06/2017 19:58 | | 68 | 0.268 | 0.11 | 14.67 |
| 14/06/2017 21:28 | | 70 | 0.260 | 0.10 | 14.62 |
| 14/06/2017 22:57 | 0.02 | 63 | 0.402 | 0.12 | 17.17 |
| 15/06/2017 00:27 | 0.03 | 52 | 0.400 | 0.10 | 15.41 |
| 15/06/2017 01:56 | 0.10 | 7 | 0.659 | 1.76 | 50.51 |
| 15/06/2017 03:26 | 7.11 | 4 | 0.803 | 33.70 | 51.79 |
| 15/06/2017 04:56 | 0.32 | 3 | 0.627 | 55.14 | 46.81 |
| 15/06/2017 06:25 | 2.70 | 3 | 0.620 | 51.52 | 39.93 |
| 15/06/2017 07:55 | 1.80 | 21 | 0.429 | 15.34 | 40.96 |
| 15/06/2017 09:24 | 2.92 | 29 | 0.450 | 9.50 | 40.83 |
| 15/06/2017 10:09 | | 41 | 0.557 | 9.00 | 43.33 |
| 15/06/2017 10:54 | | 57 | 0.529 | 4.86 | 35.53 |
| 15/06/2017 11:39 | | 71 | 0.566 | 4.25 | 35.72 |
| 15/06/2017 12:23 | 2.29 | 83 | 0.422 | 2.79 | 27.27 |
| 15/06/2017 13:08 | 2.29 | 130 | 0.334 | 0.56 | 11.01 |

| | | | | | |
|---|---|---|---|---|---|
| 15/06/2017 13:53 | 0.00 | 119 | 0.268 | 0.45 | 7.86 |
| 15/06/2017 14:38 | | 121 | 0.218 | 0.39 | 8.13 |
| 15/06/2017 15:23 | | 122 | 0.195 | 0.40 | 8.06 |
| 15/06/2017 16:08 | | 102 | 0.151 | 0.35 | 8.06 |
| 15/06/2017 16:53 | | 96 | 0.179 | 0.36 | 9.01 |
| 15/06/2017 17:38 | | 104 | 0.232 | 0.37 | 11.04 |
| 15/06/2017 18:23 | | 118 | 0.320 | 0.23 | 13.77 |
| 15/06/2017 19:53 | | 88 | 0.350 | 0.22 | 16.87 |
| 15/06/2017 21:22 | | 77 | 0.446 | 0.25 | 20.89 |
| 15/06/2017 22:52 | 0.00 | 86 | 0.560 | 0.18 | 18.07 |
| 16/06/2017 00:22 | 0.02 | 82 | 0.679 | 0.21 | 18.04 |
| 16/06/2017 01:51 | 7.21 | 67 | 0.771 | 0.13 | 21.24 |
| 16/06/2017 03:21 | 0.00 | 55 | 0.931 | 0.12 | 22.90 |
| 16/06/2017 04:50 | 0.02 | 52 | 0.779 | 0.43 | 19.69 |
| 16/06/2017 06:20 | 1.35 | 32 | 0.843 | 1.95 | 34.41 |
| 16/06/2017 07:49 | 2.88 | 11 | 0.716 | 42.07 | 66.62 |
| 16/06/2017 09:19 | 2.96 | 62 | 0.711 | 4.61 | 40.29 |
| 17/06/2017 17:42 | 0.95 | 133 | 0.526 | 0.15 | 12.05 |
| 17/06/2017 18:46 | 0.66 | 126 | 0.607 | 0.12 | 14.30 |
| 17/06/2017 19:46 | 0.22 | 117 | 0.626 | 0.25 | 17.94 |
| 17/06/2017 20:46 | 0.04 | 114 | 0.837 | 0.34 | 18.94 |
| 17/06/2017 21:46 | | 110 | 0.944 | 0.11 | 18.32 |
| 17/06/2017 22:46 | 0.03 | 107 | 0.574 | 0.38 | 10.13 |
| 17/06/2017 23:46 | 0.02 | 94 | 0.570 | 0.25 | 11.52 |
| 18/06/2017 00:46 | 0.07 | 74 | 0.572 | 0.11 | 18.21 |
| 18/06/2017 01:46 | 0.09 | 55 | 0.705 | 0.29 | 24.75 |
| 18/06/2017 02:46 | 0.03 | 56 | 0.558 | 0.13 | 16.66 |
| 18/06/2017 03:46 | 0.03 | 53 | 0.486 | 0.11 | 13.20 |
| 18/06/2017 04:46 | 0.03 | 37 | 0.523 | 0.16 | 21.90 |
| 18/06/2017 05:46 | 0.27 | 43 | 0.550 | 0.31 | 18.15 |
| 18/06/2017 06:46 | 0.38 | 48 | 0.544 | 0.69 | 16.44 |
| 18/06/2017 07:46 | 0.88 | 55 | 0.588 | 1.18 | 15.84 |
| 18/06/2017 08:46 | 0.79 | 74 | 0.563 | 0.96 | 13.12 |
| 18/06/2017 11:16 | 1.41 | 128 | 0.730 | 0.47 | 10.43 |
| 18/06/2017 12:16 | 1.76 | 149 | 0.912 | 0.33 | 9.37 |
| 18/06/2017 13:16 | | 154 | 0.864 | 0.31 | 8.87 |
| 18/06/2017 14:16 | | 151 | 0.765 | 0.26 | 8.68 |
| 18/06/2017 15:16 | 0.16 | 135 | 0.559 | 0.11 | 10.63 |
| 18/06/2017 16:16 | 0.06 | 132 | 0.787 | 0.22 | 11.26 |
| 18/06/2017 17:16 | 1.30 | 104 | 0.833 | 0.53 | 14.38 |
| 18/06/2017 18:16 | 0.81 | 56 | 1.020 | 0.31 | 30.11 |
| 18/06/2017 19:16 | 0.14 | 77 | 0.869 | 1.00 | 26.26 |
| 18/06/2017 20:16 | 0.03 | 74 | 0.707 | 0.28 | 21.20 |
| 18/06/2017 21:16 | 0.03 | 64 | 0.708 | 0.44 | 23.42 |
| 18/06/2017 22:16 | 0.02 | 76 | 0.526 | 0.46 | 13.70 |

| | | | | | |
|---|---|---|---|---|---|
| 18/06/2017 23:16 | 0.03 | 68 | 0.507 | 0.44 | 14.78 |
| 19/06/2017 00:16 | 0.02 | 65 | 0.530 | 0.18 | 12.96 |
| 19/06/2017 01:16 | 0.00 | 61 | 0.497 | 0.79 | 13.66 |
| 19/06/2017 02:16 | 0.03 | 52 | 0.437 | 0.24 | 16.51 |
| 19/06/2017 03:16 | 0.02 | 43 | 0.430 | 0.18 | 20.20 |
| 19/06/2017 04:16 | 0.03 | 45 | 0.390 | 0.15 | 17.09 |
| 19/06/2017 05:16 | 0.14 | 45 | 0.353 | 0.54 | 14.37 |
| 19/06/2017 06:16 | | 40 | 0.551 | 1.25 | 17.71 |
| 19/06/2017 07:16 | 0.74 | 45 | 0.423 | 1.88 | 14.58 |
| 19/06/2017 08:16 | 0.86 | 49 | 0.400 | 2.57 | 15.28 |
| 19/06/2017 09:16 | 0.76 | 54 | 0.325 | 2.13 | 13.48 |
| 19/06/2017 10:16 | 1.13 | 68 | 0.325 | 1.61 | 10.04 |
| 19/06/2017 11:16 | 0.87 | 81 | 0.369 | 1.19 | 10.43 |
| 19/06/2017 12:16 | 1.19 | 97 | 0.358 | 0.74 | 8.16 |
| 19/06/2017 13:16 | 1.17 | 105 | 0.302 | 0.47 | 7.51 |
| 19/06/2017 14:16 | | 121 | 0.270 | 0.31 | 6.76 |
| 19/06/2017 15:16 | 1.83 | 122 | 0.222 | 0.35 | 7.24 |
| 19/06/2017 16:16 | 1.04 | 115 | 0.297 | 0.38 | 7.96 |
| 19/06/2017 17:16 | | 107 | 0.374 | 0.36 | 12.77 |
| 19/06/2017 18:17 | 1.30 | 82 | 0.184 | 0.47 | 10.08 |
| 19/06/2017 19:16 | 0.67 | 104 | 0.331 | 0.21 | 10.52 |
| 19/06/2017 20:16 | 0.31 | 77 | 0.684 | 0.21 | 27.00 |
| 19/06/2017 21:16 | 6.96 | 83 | 0.590 | 0.28 | 20.65 |
| 19/06/2017 22:17 | 0.10 | 59 | 0.726 | 0.18 | 34.49 |
| 19/06/2017 23:16 | 0.02 | 69 | 0.683 | 0.15 | 13.06 |
| 20/06/2017 00:17 | | 67 | 0.650 | 0.22 | 13.37 |
| 20/06/2017 01:17 | | 57 | 0.526 | 0.22 | 11.97 |
| 20/06/2017 02:16 | 0.02 | 45 | 0.421 | 0.17 | 21.87 |
| 20/06/2017 03:17 | 0.00 | 55 | 0.360 | 0.10 | 13.06 |
| 20/06/2017 04:17 | 0.00 | 51 | 0.444 | 0.27 | 12.00 |
| 20/06/2017 05:17 | 0.00 | 48 | 0.462 | 0.21 | 15.34 |
| 20/06/2017 06:16 | 0.08 | 52 | 0.383 | 0.38 | 12.51 |
| 20/06/2017 07:17 | 0.28 | 54 | 0.428 | 1.38 | 14.27 |
| 20/06/2017 08:17 | 0.60 | 60 | 0.367 | 1.61 | 13.44 |
| 20/06/2017 09:17 | 0.63 | 64 | 0.417 | 2.12 | 16.07 |
| 20/06/2017 10:17 | 0.75 | 75 | 0.432 | 1.51 | 13.53 |
| 20/06/2017 11:17 | 0.63 | 88 | 0.396 | 0.89 | 9.38 |
| 20/06/2017 12:49 | 0.78 | 103 | 0.362 | 0.56 | 6.57 |
| 20/06/2017 13:49 | 1.22 | 115 | 0.393 | 0.61 | 8.24 |
| 20/06/2017 14:49 | 1.40 | 109 | 0.381 | 0.43 | 7.46 |
| 20/06/2017 15:49 | 1.07 | 110 | 0.368 | 0.41 | 8.25 |
| 20/06/2017 16:49 | | 111 | 0.350 | 0.33 | 7.01 |
| 20/06/2017 17:49 | 0.99 | 108 | 0.385 | 0.27 | 9.41 |
| 20/06/2017 18:49 | 0.99 | 103 | 0.410 | 0.19 | 11.27 |
| 20/06/2017 19:49 | 0.13 | 105 | 0.539 | 0.26 | 13.54 |

| | | | | | |
|---|---|---|---|---|---|
| 20/06/2017 20:49 | 0.04 | 100 | 0.730 | 0.11 | 15.77 |
| 20/06/2017 21:49 | 0.08 | 80 | 0.765 | 0.35 | 28.32 |
| 20/06/2017 22:49 | 0.03 | 73 | 0.830 | 0.31 | 26.08 |
| 20/06/2017 23:49 | 0.02 | 62 | 0.738 | 1.01 | 18.67 |
| 21/06/2017 00:49 | 0.00 | 59 | 0.599 | 0.35 | 16.07 |
| 21/06/2017 01:49 | 0.00 | 54 | 0.608 | 0.10 | 14.94 |
| 21/06/2017 02:49 | 0.00 | 27 | 0.741 | 0.31 | 28.11 |
| 21/06/2017 03:49 | 0.00 | 42 | 0.526 | 0.12 | 15.63 |
| 21/06/2017 04:49 | 0.03 | 37 | 0.499 | 0.15 | 17.92 |
| 21/06/2017 05:49 | 0.30 | 51 | 0.515 | 0.92 | 14.02 |
| 21/06/2017 06:49 | 0.37 | 53 | 0.579 | 0.69 | 15.62 |
| 21/06/2017 07:49 | 0.39 | 56 | 0.677 | 1.49 | 18.65 |
| 21/06/2017 08:49 | | 60 | 0.737 | 2.07 | 19.46 |
| 21/06/2017 09:49 | | 74 | 0.723 | 1.64 | 16.26 |
| 21/06/2017 10:49 | 0.88 | 90 | 0.610 | 1.14 | 12.05 |
| 21/06/2017 11:49 | 1.14 | 105 | 0.645 | 0.65 | 10.88 |
| 21/06/2017 12:52 | 0.76 | 111 | 0.738 | 0.65 | 14.18 |
| 21/06/2017 13:52 | 0.59 | 116 | 0.645 | 0.47 | 10.35 |
| 21/06/2017 14:52 | 0.84 | 122 | 0.577 | 0.31 | 9.55 |
| 21/06/2017 15:52 | 0.39 | 122 | 0.574 | 0.30 | 9.02 |
| 21/06/2017 16:52 | 0.40 | 123 | 0.691 | 0.21 | 9.72 |
| 21/06/2017 17:52 | 0.36 | 122 | 0.631 | 0.21 | 15.44 |
| 21/06/2017 18:52 | 0.04 | 104 | 0.581 | 0.66 | 14.13 |
| 21/06/2017 19:52 | 0.09 | 59 | 0.763 | 0.15 | 31.26 |
| 21/06/2017 20:52 | 0.03 | 82 | 0.460 | 0.76 | 9.05 |
| 21/06/2017 21:52 | 0.02 | 75 | 0.280 | 0.52 | 6.96 |
| 21/06/2017 22:52 | 0.00 | 72 | 0.303 | 0.26 | 8.58 |
| 21/06/2017 23:52 | 0.03 | 69 | 0.275 | 0.33 | 6.93 |
| 22/06/2017 00:52 | 0.00 | 64 | 0.279 | 0.12 | 9.09 |
| 22/06/2017 01:52 | | 60 | 0.275 | 0.22 | 8.98 |
| 22/06/2017 02:52 | 0.00 | 54 | 0.296 | 0.12 | 9.99 |
| 22/06/2017 03:52 | 0.02 | 54 | 0.313 | 0.17 | 8.88 |
| 22/06/2017 04:52 | 0.02 | 47 | 0.318 | 0.72 | 11.68 |
| 22/06/2017 05:52 | 0.05 | 31 | 0.349 | 0.50 | 21.88 |
| 22/06/2017 06:52 | 0.11 | 34 | 0.342 | 1.60 | 23.94 |
| 22/06/2017 07:52 | 0.24 | 40 | 0.391 | 2.10 | 24.58 |
| 22/06/2017 08:52 | 0.26 | 42 | 0.404 | 0.93 | 23.82 |
| 22/06/2017 09:52 | 0.28 | 48 | 0.442 | 0.95 | 19.78 |
| 22/06/2017 10:50 | 0.29 | 52 | 0.401 | 0.74 | 17.01 |
| 22/06/2017 11:50 | 0.17 | 65 | 0.431 | 0.52 | 10.61 |
| 22/06/2017 12:50 | 0.23 | 59 | 0.496 | 0.38 | 13.39 |
| 22/06/2017 13:27 | 0.23 | 59 | 0.466 | 0.60 | 12.54 |
| 22/06/2017 14:27 | 0.41 | 60 | 0.530 | 0.47 | 12.70 |
| 22/06/2017 15:27 | 0.21 | 57 | 0.543 | 0.38 | 12.80 |
| 22/06/2017 16:44 | 0.41 | 40 | 0.714 | 0.31 | 20.77 |

| | | | | | |
|---|---|---|---|---|---|
| 22/06/2017 17:44 | | 39 | 0.965 | 0.32 | 26.32 |
| 22/06/2017 18:44 | | 48 | 0.892 | 0.15 | 20.33 |
| 22/06/2017 19:44 | 0.11 | 44 | 0.965 | 0.12 | 19.31 |
| 22/06/2017 20:44 | 0.10 | 43 | 1.108 | 0.14 | 18.74 |
| 22/06/2017 21:44 | 0.07 | 40 | 1.191 | 0.14 | 16.56 |
| 22/06/2017 22:44 | 0.04 | 53 | 0.716 | 0.13 | 11.19 |
| 22/06/2017 23:44 | 0.03 | 54 | 0.589 | 0.11 | 9.43 |
| 23/06/2017 00:44 | 0.02 | 48 | 0.566 | 0.12 | 10.17 |
| 23/06/2017 01:44 | 0.03 | 56 | 0.512 | 0.12 | 5.16 |
| 23/06/2017 02:44 | 0.03 | 56 | 0.422 | 0.13 | 4.31 |
| 23/06/2017 03:44 | 0.01 | 58 | 0.343 | 0.10 | 4.07 |
| 23/06/2017 04:44 | 0.00 | 57 | 0.338 | 0.10 | 3.82 |
| 23/06/2017 05:44 | 0.02 | 55 | 0.307 | 0.10 | 5.71 |
| 23/06/2017 06:44 | 0.04 | 47 | 0.383 | 0.13 | 11.26 |
| 23/06/2017 07:44 | 0.05 | 41 | 0.446 | 0.19 | 16.59 |
| 23/06/2017 08:44 | 0.17 | 43 | 0.403 | 0.98 | 13.25 |
| 23/06/2017 09:44 | 0.25 | 43 | 0.373 | 1.06 | 11.30 |
| 23/06/2017 10:44 | | 47 | 0.384 | 0.62 | 7.06 |
| 23/06/2017 11:44 | 0.28 | 43 | 0.476 | 1.08 | 11.51 |
| 23/06/2017 12:44 | 7.50 | 43 | 0.583 | 1.19 | 11.53 |
| 23/06/2017 13:44 | 7.29 | 37 | 0.649 | 0.96 | 15.50 |
| 23/06/2017 14:57 | 0.23 | 34 | 0.494 | 0.74 | 12.36 |

**Table S3: Chemical data plotted in Fig. 4.**

| Hour of Day (h) | Isoprene (ppb) | O₃ (ppb) | OH (cm⁻³) | NO (ppb) | NO₃ (ppt) | (4-OH, 3-ONO₂)-IHN (ppt) | E-(1-ONO₂, 4-CO)-ICN (ppt) | Propanone nitrate (ppt) |
|---|---|---|---|---|---|---|---|---|
| 0 | 0.33 | 47 | 4.72E+05 | 3.1 | 10.7 | 3.2 | 7.3 | 47 |
| 1 | 0.47 | 40 | 3.82E+05 | 4.9 | 10.5 | 2.4 | 6.2 | 45 |
| 2 | 0.47 | 34 | 3.58E+05 | 4.8 | 8.3 | 2.1 | 4.8 | 43 |
| 3 | 0.62 | 28 | 3.65E+05 | 9.8 | 5.8 | 1.7 | 4.4 | 45 |
| 4 | 0.70 | 28 | 3.92E+05 | 8.4 | 4.8 | 1.5 | 3.7 | 51 |
| 5 | 0.39 | 22 | 5.93E+05 | 15.3 | 1.9 | 1.5 | 3.6 | 36 |
| 6 | 0.72 | 21 | 1.16E+06 | 11.7 | 0.3 | 1.4 | 3.4 | 36 |
| 7 | 0.42 | 23 | 2.65E+06 | 12.3 | 0.2 | 1.7 | 3.3 | 35 |
| 8 | 0.88 | 27 | 4.05E+06 | 12.5 | 0.2 | 3.0 | 2.4 | 39 |
| 9 | 0.93 | 36 | 5.82E+06 | 6.7 | 0.3 | 4.7 | 1.9 | 45 |
| 10 | 0.88 | 44 | 7.24E+06 | 5.0 | 0.5 | 6.7 | 1.2 | 45 |
| 11 | 1.03 | 59 | 8.34E+06 | 3.7 | 0.8 | 9.4 | 1.5 | 55 |
| 12 | 1.17 | 73 | 8.90E+06 | 2.9 | 1.2 | 10.2 | 1.5 | 47 |
| 13 | 1.15 | 89 | 8.91E+06 | 1.5 | 1.7 | 10.1 | 1.5 | 44 |
| 14 | 0.83 | 91 | 8.55E+06 | 1.2 | 2.0 | 8.5 | 1.6 | 44 |
| 15 | 1.07 | 95 | 7.78E+06 | 0.8 | 2.2 | 8.3 | 2.4 | 37 |
| 16 | 0.89 | 102 | 6.04E+06 | 0.9 | 2.3 | 7.5 | 1.4 | 43 |
| 17 | 0.88 | 98 | 3.92E+06 | 0.7 | 2.1 | 7.2 | 1.6 | 39 |
| 18 | 1.07 | 94 | 1.83E+06 | 0.5 | 2.2 | 7.9 | 1.7 | 31 |
| 19 | 0.52 | 87 | 1.04E+06 | 0.7 | 3.6 | 6.3 | 2.3 | 32 |
| 20 | 0.19 | 76 | 1.14E+06 | 0.7 | 8.2 | 5.9 | 3.6 | 35 |
| 21 | 0.65 | 69 | 1.09E+06 | 1.7 | 12.2 | 5.2 | 8.1 | 40 |
| 22 | 0.09 | 63 | 8.81E+05 | 1.0 | 13.1 | 3.5 | 7.9 | 43 |
| 23 | 0.16 | 54 | 5.94E+05 | 1.5 | 12.1 | 3.5 | 8.2 | 48 |
| 24 | 0.33 | 47 | 4.72E+05 | 3.1 | 10.7 | 3.2 | 7.3 | 47 |

**Table S4: Mixing layer height data plotted in Fig. 4.**

| Hour of Day (h) | Mixed Layer Height (m*10) |
|---|---|
| 0.125 | 30.7 |
| 0.375 | 28.7 |
| 0.625 | 28.6 |
| 0.875 | 29.0 |
| 1.125 | 29.0 |
| 1.375 | 28.2 |
| 1.625 | 27.9 |
| 1.875 | 28.0 |
| 2.125 | 28.3 |
| 2.375 | 28.3 |
| 2.625 | 28.3 |
| 2.875 | 28.3 |
| 3.125 | 28.1 |
| 3.375 | 28.3 |
| 3.625 | 28.8 |
| 3.875 | 28.7 |
| 4.125 | 28.6 |
| 4.375 | 28.9 |
| 4.625 | 29.1 |
| 4.875 | 29.0 |
| 5.125 | 29.8 |
| 5.375 | 30.9 |
| 5.625 | 31.4 |
| 5.875 | 31.9 |
| 6.125 | 32.7 |
| 6.373 | 33.3 |
| 6.625 | 33.9 |
| 6.875 | 35.0 |
| 7.125 | 37.1 |
| 7.375 | 39.3 |
| 7.625 | 40.6 |
| 7.875 | 42.9 |
| 8.125 | 45.7 |
| 8.375 | 48.1 |
| 8.625 | 50.8 |
| 8.875 | 53.8 |
| 9.125 | 56.8 |
| 9.375 | 59.4 |
| 9.625 | 62.9 |
| 9.875 | 66.9 |
| 10.125 | 69.9 |
| 10.375 | 73.1 |

| | |
|---|---|
| 10.625 | 76.6 |
| 10.875 | 80.5 |
| 11.125 | 84.7 |
| 11.375 | 87.5 |
| 11.625 | 89.5 |
| 11.877 | 91.8 |
| 12.125 | 94.1 |
| 12.375 | 96.2 |
| 12.625 | 98.4 |
| 12.875 | 100.8 |
| 13.125 | 102.5 |
| 13.375 | 103.4 |
| 13.625 | 103.9 |
| 13.875 | 105.2 |
| 14.125 | 107.0 |
| 14.375 | 106.8 |
| 14.625 | 105.4 |
| 14.875 | 104.5 |
| 15.125 | 104.9 |
| 15.375 | 105.3 |
| 15.625 | 105.3 |
| 15.875 | 105.9 |
| 16.125 | 106.3 |
| 16.375 | 104.6 |
| 16.625 | 102.6 |
| 16.875 | 100.6 |
| 17.125 | 97.1 |
| 17.375 | 94.1 |
| 17.625 | 93.0 |
| 17.875 | 92.8 |
| 18.125 | 91.6 |
| 18.375 | 90.1 |
| 18.625 | 87.7 |
| 18.875 | 83.9 |
| 19.125 | 79.4 |
| 19.375 | 75.3 |
| 19.625 | 70.2 |
| 19.875 | 64.8 |
| 20.125 | 61.2 |
| 20.375 | 57.7 |
| 20.625 | 54.3 |
| 20.875 | 49.9 |
| 21.125 | 45.0 |
| 21.375 | 42.1 |
| 21.625 | 40.4 |

| | |
|---|---|
| 21.875 | 39.1 |
| 22.125 | 38.7 |
| 22.375 | 38.0 |
| 22.625 | 36.7 |
| 22.875 | 36.0 |
| 23.125 | 35.8 |
| 23.375 | 35.3 |
| 23.625 | 34.3 |
| 23.750 | 33.7 |

**Table S5: Data plotted in Fig. 5.**

| Hour of Day (h) | Modelled (4-OH, 3-ONO$_2$)-IHN (ppt) | Modelled (1-OH, 2-ONO$_2$)-IHN (ppt) | Observed mean (4-OH, 3-ONO$_2$)-IHN (ppt) | Observed mean (1-OH, 2-ONO$_2$)-IHN (ppt) |
|---|---|---|---|---|
| 0 | 0.9 | 0.7 | 3.2 | 16.5 |
| 1 | 1.3 | 3.9 | 2.4 | 6.6 |
| 2 | 1.5 | 7.4 | 2.1 | 11.0 |
| 3 | 1.4 | 5.0 | 1.7 | 7.6 |
| 4 | 1.4 | 2.9 | 1.5 | |
| 5 | 1.5 | 2.0 | 1.5 | 6.9 |
| 6 | 4.3 | 8.7 | 1.4 | 5.3 |
| 7 | 3.2 | 5.2 | 1.7 | 12.3 |
| 8 | 4.3 | 8.5 | 3.0 | 7.8 |
| 9 | 9.0 | 17.2 | 4.7 | 14.7 |
| 10 | 12.7 | 28.3 | 6.7 | 30.3 |
| 11 | 16.6 | 38.0 | 9.4 | 26.5 |
| 12 | 18.6 | 48.2 | 10.2 | 46.4 |
| 13 | 19.2 | 44.4 | 10.1 | 35.8 |
| 14 | 17.9 | 38.0 | 8.5 | 28.4 |
| 15 | 16.4 | 37.7 | 8.3 | 38.2 |
| 16 | 13.2 | 32.7 | 7.5 | 30.0 |
| 17 | 9.4 | 24.7 | 7.2 | 29.2 |
| 18 | 5.6 | 12.3 | 7.9 | 48.9 |
| 19 | 2.5 | 3.1 | 6.3 | 28.0 |
| 20 | 2.1 | 0.4 | 5.9 | 28.4 |
| 21 | 1.2 | 0.2 | 5.2 | 22.4 |
| 22 | 0.5 | 0.2 | 3.5 | |
| 23 | 0.7 | 0.2 | 3.5 | 20.3 |
| 24 | 0.9 | 0.7 | 3.2 | 16.5 |

**Table S6: Observed and Modelled data plotted in Fig. 6.**

| Hour of Day (h) | (1-OH, 2-ONO$_2$)-IHN / (4-OH, 3-ONO$_2$)-IHN ratio | |
| --- | --- | --- |
| | Observed | Modelled |
| 0 | 3.02 | 2.36 |
| 1 | 1.45 | 2.29 |
| 2 | 4.38 | 2.23 |
| 3 | 3.25 | 2.21 |
| 4 | | 2.11 |
| 5 | 2.03 | 1.98 |
| 6 | 2.19 | 1.96 |
| 7 | 3.77 | 1.97 |
| 8 | 2.15 | 2.00 |
| 9 | 4.94 | 2.05 |
| 10 | 5.82 | 2.14 |
| 11 | 3.03 | 2.26 |
| 12 | 2.85 | 2.47 |
| 13 | 3.13 | 2.78 |
| 14 | 2.95 | 2.63 |
| 15 | 3.29 | 2.57 |
| 16 | 3.18 | 2.51 |
| 17 | 3.81 | 2.53 |
| 18 | 3.93 | 2.57 |
| 19 | 3.09 | 2.57 |
| 20 | 3.47 | 2.57 |
| 21 | 3.29 | 2.51 |
| 22 | | 2.51 |
| 23 | 4.14 | 2.45 |
| 24 | 3.02 | 2.36 |

**Table S7: Observed data plotted in Figs. 8 and 9.**

| Hour of Day (h) | Observed ICN Total (ppt) | Observed Propanone Nitrate (ppt) | Modelled ICN Total (ppt) | Modelled Propanone Nitrate (ppt) | Modelled (1-OH, 4-ONO$_2$)-IHN (ppt) | Modelled (4-OH, 1-ONO$_2$)-IHN (ppt) |
|---|---|---|---|---|---|---|
| 0 | 18.9 | 47 | 105.1 | 14.2 | 0.06 | 1.30 |
| 1 | 15.1 | 45 | 108.6 | 16.2 | 0.08 | 1.36 |
| 2 | 13.1 | 43 | 93.5 | 15.2 | 0.11 | 0.93 |
| 3 | 11.3 | 45 | 88.8 | 13.3 | 0.18 | 0.75 |
| 4 | 7.7 | 51 | 87.6 | 12.0 | 0.22 | 0.62 |
| 5 | 7.7 | 36 | 62.2 | 10.5 | 0.29 | 0.46 |
| 6 | 7.1 | 36 | 25.2 | 9.9 | 0.83 | 0.77 |
| 7 | 6.1 | 35 | 3.3 | 2.1 | 0.49 | 0.46 |
| 8 | 4.4 | 39 | 1.5 | 0.3 | 0.48 | 0.48 |
| 9 | 2.9 | 45 | 3.1 | 0.8 | 0.61 | 0.65 |
| 10 | 2.0 | 45 | 5.2 | 1.7 | 0.68 | 0.71 |
| 11 | 2.5 | 55 | 9.3 | 3.6 | 0.76 | 0.76 |
| 12 | 2.5 | 47 | 14.3 | 6.2 | 0.76 | 0.73 |
| 13 | 2.4 | 44 | 19.6 | 9.1 | 0.74 | 0.69 |
| 14 | 1.7 | 44 | 22.5 | 10.7 | 0.69 | 0.62 |
| 15 | 2.9 | 37 | 27.4 | 12.9 | 0.62 | 0.55 |
| 16 | 2.0 | 43 | 30.3 | 14.5 | 0.50 | 0.44 |
| 17 | 2.7 | 39 | 28.7 | 11.0 | 0.37 | 0.33 |
| 18 | 2.6 | 31 | 31.1 | 6.9 | 0.24 | 0.23 |
| 19 | 5.1 | 32 | 30.4 | 4.5 | 0.11 | 0.14 |
| 20 | 11.0 | 35 | 48.9 | 5.9 | 0.09 | 0.41 |
| 21 | 22.8 | 40 | 69.4 | 7.0 | 0.05 | 0.76 |
| 22 | 21.0 | 43 | 52.6 | 5.5 | 0.02 | 0.78 |
| 23 | 18.8 | 48 | 82.2 | 9.9 | 0.04 | 0.89 |
| 24 | 18.9 | 47 | 105.1 | 14.2 | 0.06 | 1.30 |

---

## Author Response (AR2)

**Author response are in blue.**

**Referee 1**

This paper remains a rambling unfocussed document. As I said in my previous review it lacks the organization that would let any but the most determined reader identify what, if anything has been learned. I recommend it be rejected. It started too far from acceptable to have had a proper open review. I don't see how it could become acceptable in one or two more revisions.

We have reorganised the paper substantially. There no longer separate sections on the observations and model results. Instead, they have been combined with much of the description of the observations removed such that the text is far more focussed on the interpretation and so on what is learnt.

I recommend the paper create a new figure or figures with the specific set of rate constants (or ratios of rate constants) the paper is testing. That the figure should guide the reader to thinking about what is already known and what is new in this paper.

This paper is based on field observations, some of which are the first of their kind, and we compare them with a model using a chemical mechanism that is widely used around the world to help evaluate understanding of the isoprene nitrate chemistry. The mechanism is complex and involves many reactions that contribute together to the model outputs and is well-documented elsewhere (Jenkin et al., 2015). We have added a diagram (Fig. 5) specifically to illustrate some of the key reactions in the formation of the IHN to help the reader particularly when thinking about the ratios. In the introduction we have made more reference to the existing Fig. 1 to help the reader understand the aspects of the chemistry that we are examining.

Some other specific comments follow. I have not taken the time to be complete in my suggestions. There are just too many places where this paper is in need of editing and focus.

We can only address the comments that the referee has made.

1) The abstract is 4 paragraphs. We do not consider this to be a problem, but in the rewrite this has been reduced to 3. The last one is inappropriate in an abstract. We are not sure why it is inappropriate, but it has changed in the rewrite. One in the middle includes the phrase, "although the correlation is weak due to their being only a few data points." That paragraph should be removed. It sets the tome for the rest of the paper as a rambling collection of facts and not an attempt to distill what has been learned into some key conclusions. This paragraph has been rewritten and hopefully now reflects the rest of the paper that is more focussed on what has been learnt.

2) All of the discussion in the paper that refers to their being too few data points to draw a strong conclusion should be deleted.

Our field observations are limited by the number of data points we have, but we believe that the relationships that they show, even if not strong, are important as there are virtually no other data of this kind available. They also point to where further research can improve understanding. Referee 3 seems to agree with our view stating, "The measurements made were difficult, and, with the uncertainties added in the revision, represent useful ambient data.".

3) The introduction reads like it was written years ago. I recommend it start with W2018 and articulate key current uncertainties. Going back to papers that quote 4%IN yields that no one following the literature believes any more is not helpful to the reader.

The introduction has been shortened to make it more focused and reference to much of this older research has been removed.

4) The paper would be better if it started with the model predictions and then articulated which aspects of the model predictions the observations are able to test.

We have adopted this approach.

5) If the winter data is unusable, remove discussion of that from the paper.

We have removed all comments about the winter campaign.

6) Section 4.2.1 mixes up a description of what was measured with a comparison to models and other locations. The paper would be clearer if these were completely separated.

This section has been removed in the reorganisation and tightening of focus. We have also reduced the comparison with previous research where comparison is of less relevance.

7) Line 243. The opposite of not surprising, is expected. I recommend the authors describe their results as what was expected. This has been removed when addressing point 6.

8) Line 251, starting the discussion with non-detection of chemicals (and reporting that non-detection is expected) is not a good beginning. The discussion should start with observations that can teach us something. I believe the referee means lines 241. I agree that this is not the most positive of starts. Non-detection can still teach us something and discussion of this is now presented later in the paper.

9) Lines 260-270 there are so many differences in chemical concentrations between these observations and those of SOAS that a direct comparison is not meaningful. The reviewer's point 11 below seems at odds with what they say here. We have however, removed much of this in tightening the focus of the paper.

10) Line 311-313. In what way do these comments about OH advance the theme of this paper. The fact that OH was observed at night is important as it shows that OH initiated oxidation of isoprene can be a source of isoprene nitrates at night, i.e. not just $NO_3$ initiated oxidation. However, in tightening up the paper this has been removed.

11) Line 315-325 I recommend moving all of the comparison to SOAS to a single location in the paper. We have not done this, largely because much of this comparison has been removed in response to point 9 above.

12) Lines 429-435 and similar paragraphs should be deleted. There is no justification for removing this paragraph, unless this relates to comment 2 above. I have responded to that already. Furthermore, the mention of "few data points" is just one aspect to this paragraph and to say that this paragraph and all like it should be deleted is unwarranted.

13) Line 490-496 lacks a clear focus This was rather descriptive. Much of this text has now been removed in the rewrite.

**Referee 3**

We appreciate the constructive comments that this referee has provided. They are useful in improving the manuscript.

The paper by Reeves et al. has been significantly improved. The authors have carefully addressed the main issues raised, in my judgement. I feel that the revised paper is a significant contribution to the literature, given the speciated organic nitrate measurements reported here, from isoprene chemistry. The measurements made were difficult, and, with the uncertainties added in the revision, represent useful ambient data.

The paper could be further improved, if there is one more round of edits by again paying less attention to the tedious details (e.g. telling the reader about all the information that is in W2018) and focusing on what is new.

We have made extensive changes to the organisation of the paper to make it more focussed and removing many of the tedious details. References to W2018 in the main text are rather few. However, we have included a summary of the isoprene nitrate chemistry in the supplementary information (SI), much of which is based on W2018. Perhaps this is what the referee refers to. We have included it in the SI to help the more general reader who might not be determined enough to find this information in W2018, which is an extensive paper. We also note that the referee makes minor comments below (19 to 22), which include additions to this section. It is therefore not clear whether they want this removed or not. As it is in the supplementary information, we don't see any need to delete it.

In particular, the Conclusions just repeat what is said in the paper, implying that it wasn't said well enough in the paper. The Conclusions section is an opportunity to paint the big picture as to – what did we learn that is new? A list of the points made in the paper is relatively dull. It would be better to focus on what new understanding was obtained, and at least equally importantly, what needs to be done next (which is there to some extent). Are there suggestions for improved measurements? Better time resolution? Are there laboratory studies that need to be done? What could be done to better understand the fate of nitrooxy-peroxy radicals, etc. I note that with all those standards, it would be trivial to measure all the hydrolysis rates, and information about hydrolysis lifetimes vs aerosol pH and structure are badly needed.

We have rewritten the conclusions to make them more of a synthesis rather than a repetition of points made earlier in the paper. The emphasis is on what was learnt, and we have added more comments on further areas of research required.

Minor edits I would recommend are listed below, in the order they arose in the paper.

1. Line 123 – We used a chemical…
   Typo eliminated in the rewrite.
2. Section 4.1 is not needed, it can be incorporated into the results and discussion, as needed.
   The text about the winter campaign has been deleted in response to comment 5 by reviewer 1. The sentence describing the air quality conditions during the summer campaign has been incorporated into section 3.1 where a brief overview of the campaign is provided.
3. Line 244 – during the Southern Oxidant…
4. Typo eliminated in the rewrite.
5. Line 261 – why would that be? Please speculate. Is O3 highest in the high NO2/NO condition, and is OH highest at that time? Perhaps you can examine the product [OH][isoprene]xGamma.
   We took a look into this, but with nothing conclusive and in tightening the focus of the paper, this comment has been removed.
6. Line 402 – are you assuming that all species are mixing with clean air? This is an assumption that cannot be justified for all species equivalently. Some longer-lived species such as propanone nitrate will likely have significant concentrations in the residual layer, and mix down as the BL height grows, leading to overestimation of the dilution in such a case, if you are treating all species the same. This should be discussed wherever you discuss the mixing issue (e.g. line 579, 592).
   The mixing is dealt with using a loss rate constant and does not make any specific assumptions about the concentrations of the species in the entrained air. However, we do assume the loss rate constant is the same for all species, which in reality will change for the reasons stated by the referee. I thought we had addressed this comment in the last revision with the inclusion of the comments on lines 575 to 578 and again in lines 590-593. We have now added a further comment on this at the end of section 4 and at the beginning of section 5.3.
7. Lines 424 and 425 – this implies that for some days, model and data compare better? If so, say so

We have added this (section 5.1).

8. Lines 426 and 427 – could there be greater evening production or slower loss?
   Yes. Added to section 5.1.

9. Line 432 – than observed…
   Typo corrected.

10. Line 434 – delete comma after that
    Typo corrected.

11. Line 485- this assumes that transport to the surface is not the rate limiting step for deposition of "sticky" species. If you have reason to believe that, say why.
    We do not have a strong reason for saying this so have deleted the paragraph. We have added a comment to the conclusions regarding this being an area that requires further study.

12. Line 525 (paragraph) – I think it would help the discussion to plot the average measured [NO3][isoprene] in Figure 8.
    The model is constrained by observed concentrations of $NO_3$ and isoprene, so the model diagnostic for the production term of δ-nitrooxy peroxy radicals ($INO_2$) is $[NO_3]$[isoprene]k. We discuss this modelled production term in the text (section 5.2) and do not feel there is any need to include $[NO_3]$[isoprene] to the figure.

13. Line 624 – specify the ratio being discussed.
    Done.

14. Line 638 – comma after 7
    Added.

15. Line 665 – "Our interpretation is limited by the uncertainties in our measurements" – how so?
    This was a rather general comment, but we have added a comment on the need for a more accurate calibration of (1-OH, 2-$ONO_2$)-IHN. We also note the need for more measurements, which, although not specifically a measurement uncertainty, contributes to the standard deviation in hour of day means.

16. Figure 3 – The two colors in panel c are hard to distinguish. Could you plot radiation on those plots? It would help the reader with the discussion of daytime v. nighttime sources of propanone nitrate.
    We have changed the colours. The vertical gridlines are at each midnight, which we now point out in the figure caption. Note this figure is now Fig. 8.

17. Figure 4 – I think you need legends in the figure – the colors won't work well in the published version.
    The figure caption has been expanded explain the colour coding. Note this figure has now been moved to the supplementary information (Fig. S4).

18. Figure 6 – use a color other than yellow?
    This has been changed. Note this figure is now Fig. 4. We have also changed the colours of Fig. 3 to make it consistent with the colours used in Fig 4.

19. Figure 7 – Why not add a line to panel F that represents the best fit to the data in panel E, for ease of comparison of measured and modelled data (esp. since the scales are different)?
    Lines based on the mean β-IHN ratios for bins of NO have been added to both panel E and F to ease comparison. Note this figure is now Fig. 6.

20. S1.1 – state which values for the branching ratio you used for each ISOPOO.
    This is not very straight forward because, as discussed in this section of the supplementary information and in the main paper, the reactions of these adducts with $O_2$ to form ISOPOO are reversible and so lead to a redistribution of the ISOPOO which will change as a function of NO. However, we have added the branching ratios that we used for the initial formation of the adducts (Jenkin et al, 2015) for comparison with those recommended by W2018.

21. S1.2 – first line – it should be nitrooxy, I think.
    Typo corrected.

22. S1.3.1 line 6 and line 13 – you should mention production of the organic hydroperoxides when you mention HO2 as a reactant.

   Added

23. In this section you refer to "reaction rates" throughout, when it should say rate constants.

   We have corrected that in this section and throughout the manuscript.

**Editor**

Given the concerns of the reviewers I recommend you do a careful and extensive revision of the document. Please take all comments into careful consideration including carefully considering a reorganization of the paper as suggested by one of the reviewers. If the paper is hard to follow for experts (and myself) it will be hard for general readers to benefit from the interesting findings.

In our response above, we hope we have demonstrated that we have carefully considered all of the reviewers' comments.

**Editor Response**

I have read the manuscript carefully and thought about the issue. Here are my thoughts.

1. The measured data is a valuable addition to existing datasets of isoprene oxidation products.

We agree.

2. The manuscript could indeed benefit from some restructuring as one of the reviewers has suggested. You mention in the title the implications for isoprene chemistry. However, these are not spelled out clearly in the abstract or the conclusion. I think the reason is that the manuscript talks extensively about the individual measurements of an array of isoprene oxidation products. It also presents the model results. Both of these are important but it is hard for a reader, at least for me, to follow and understand where the manuscript is going with this. I think it would be useful to focus on the underlying kinetics/processes that, in my humble opinion, are the ones tied to the implications of your work. In other words, what do your measurements tell us about the underlying processes.

We have restructured the manuscript as described above. To focus on the implications, we have shortened the manuscript, removing some of descriptions of the observations and structuring the paper around the comparison of the observations with the model. We also believe that the implications for the chemistry are now more clearly articulated in the Abstract and Conclusions.

3. I thought about how this could be implemented. I think it would be useful, early on in the manuscript, to lay out the processes that for example lay out the ratios of reaction products, be it the 1,2 vs 3,4 position or others. It is my understanding that the current state of understanding of the processes is reflected largely in the model. This could then be followed by the observations and a comparison with the model results. Perhaps some model results could already be used to detail what processes you are discussing, but I am not sure whether that would be putting the cart before the horse, if that is the correct expression.

Within the results sections, we have used the model results to lay out current understanding of the key processes, often putting this first and then using the observations to test this.

4. The conclusion could then focus on the actual implications of your work. What processes/kinetics show statistically significant discrepancies and what could the causes be, e.g., model inputs such as some constraint that may be erroneous, or some other process be it production or loss that may be

misrepresented. I think all of this is already in the document except perhaps what your findings tell us about causes for discrepancies but is somehow hard to get a good synthesis sense of this.

We have now provided more of a synthesis within the Conclusions.

In summary, the manuscript contains valuable data and analysis, but to make it more comprehensible the focus could be shifted more to the scientific understanding rather than listing observations and model results. I admit that as this is a complex chemistry it is difficult to not have the extensive discussion of observations and model results. It may be useful to add some figures that show the processes determining the ratios of e.g., the IHNs and others. as I think it is very hard to grasp this just from text. I do think having a figure(s) that shows what you are trying to understand and then discussing this could be helpful although I admit the complexity of the chemistry makes this challenging…

By removing descriptive text, restructuring the sections and focussing on the discrepancies between the model and the observations, we believe the paper is more focused on the scientific findings and more comprehensible. We have added a diagram (Fig. 5) specifically to illustrate some of the key reactions in the formation of the IHN to help the reader particularly when thinking about the ratios.

Please, do not understand my suggestions as trying to "tell you what to do". I am trying to help to find a pathway to address one of the reviewers concerns and I think shifting to starting with something like a concrete hypothesis that you then test, discuss, and finally suggest where issues my lie could do this.

Thank you for taking the time to provide these useful suggestions, many of which we have followed.

---

## Author Response (AR3)

**Response to Editor's comments (acp-2019-964)**

Response in blue

I only have one minor point. You mention on line 210-211 that the loss rate was adjusted to have glyoxal match observations and mention Figure S4 for this. However, unless I am mistaken, I did not see glyoxal in the figure.

The reference to the Fig. S4 was for the mixing layer height. The loss rate is set proportional to the mixed layer height.

1. As this loss rate is quite important, could you please add glyoxal to Figure S4.

We have added a new figure for glyoxal. Fig. S4 was already very busy, with 4 panels and 9 parameters plotted. Also, the figure contained only measurements and no model results. If we were to show the glyoxal data, it made sense to show both the measured and model data. We have therefore created a new Fig. S4 which contains a panel with the measured mixed layer height and a panel with the measured and modelled glyoxal mixing ratio. The glyoxal data plotted is provided in a new Table S8.  The old Fig. S4 became Fig. S5 (and so on). The mixed layer height has been removed from Fig. S5 panel (d) as it is now in Fig. S4 and the NO data plotted in panel (b) has been moved to panel (d) so that panel (b) is less busy.

The names and references to all figures in main manuscript have been updated to take account of the new figure.

2. It has been suggested by various authors that glyoxal can have substantial aerosol loss but there is not necessarily agreement on this (Volkamer, GRL 2007 https://doi.org/10.1029/2007GL030752; Washenfelder JGR 2011 DOI: 10.1029/2011JD016314, Li et al JGR 2016 https://doi.org/10.1002/2016JD025331). It would be helpful to have a statement on the impact this has or does not have on the loss rate.

We have added a couple of sentences to address this:

"We do not consider uptake of glyoxal onto aerosol, the rate of which is highly uncertain (Volkamer et al., 2007; Washenfelder et al., 2011; Li et al., 2016). Including one would have led to use of a slower ventilation rate.".

Once this is addressed the manuscript is ready for publication.

We are grateful to the editor for the advice he has provided that has enabled us to get this manuscript to a state where it is acceptable for publication.